# Misclassification in memory modification in *App^NL-G-F* knock-in mouse model of Alzheimer's disease

**Mei-Lun Huang[1], Yusuke Suzuki[1,2,3]\*, Hiroki Sasaguri[4,5], Takashi Saito[6], Takaomi C Saido[4], Itaru Imayoshi[1,2,3]\***

[1]Laboratory of Brain Development and Regeneration, Division of Systemic Life Science, Graduate School of Biostudies, Kyoto University, Kyoto, Japan; [2]Center for Living Systems Information Science, Graduate School of Biostudies, Kyoto University, Kyoto, Japan; [3]Laboratory of Deconstruction of Stem Cells, Institute for Life and Medical Sciences, Kyoto University, Kyoto, Japan; [4]Laboratory for Proteolytic Neuroscience, RIKEN Center for Brain Science, Saitama, Japan; [5]Dementia Pathophysiology Collaboration Unit, RIKEN Center for Brain Science, Saitama, Japan; [6]Department of Neurocognitive Science, Institute of Brain Science, Graduate School of Medical Sciences, Nagoya City University, Aichi, Japan

**\*For correspondence:**
suzuki.yusuke.7n@kyoto-u.ac.jp (YS);
imayoshi.itaru.2n@kyoto-u.ac.jp (II)

**Competing interest:** The authors declare that no competing interests exist.

## eLife Assessment

This **valuable** study employs a formalized computational model of learning to assess memory deficits in Alzheimer's Disease with the goal of developing an early diagnosis tool. Using an established mouse model of the disease, the authors studied multiple behavioral tasks and ages with the goal of showing similarities in behavioral deficits across tasks. Using the model, the authors indicate specific deficits in memory (overgeneralization and over differentiation) in mice with the transgene for the disease. The evidence presented is **solid**, yet certain concerns remain regarding the interpretation of the results of the modeling.

**Abstract** Alzheimer's disease (AD), the leading cause of dementia, could potentially be mitigated through early detection and interventions. However, it remains challenging to assess subtle cognitive changes in the early AD continuum. Computational modeling is a promising approach to explain a generative process underlying subtle behavioral changes with a number of putative variables. Nonetheless, internal models of the patient remain underexplored in AD. Determining the states of an internal model between measurable pathological states and behavioral phenotypes would advance explanations about the generative process in earlier disease stages beyond assessing behavior alone. Previously, Gershman et al., 2017b proposed the latent cause model, which provides a normative account of memory modification phenomena in Pavlovian fear conditioning. Here, we assumed the latent cause model as an internal model and estimated internal states defined by the model parameters being in conjunction with measurable behavioral phenotypes. The 6- and 12-month-old *App^NL-G-F* knock-in AD model mice and the age-matched control mice underwent memory modification learning, which consisted of classical fear conditioning, extinction, and reinstatement. The results showed that *App^NL-G-F* mice exhibited a lower extent of reinstatement of fear memory. Computational modeling revealed that the deficit in the *App^NL-G-F* mice would be due to their internal states being biased toward overgeneralization or overdifferentiation of their observations, and consequently, the competing memories were not retained. This deficit was replicated in another type of memory modification learning in the reversal Barnes maze task. Following reversal

learning, $App^{NL-G-F}$ mice, given spatial cues, failed to infer coexisting memories for two goal locations during the trial. We concluded that the altered internal states of $App^{NL-G-F}$ mice illustrated their misclassification in the memory modification process. This novel approach highlights the potential of investigating internal states to precisely assess cognitive changes in early AD and multidimensionally evaluate how early interventions may work.

## Introduction

Alzheimer's disease (AD) is the leading cause of dementia, which severely disturbs daily lives in the aging population and causes a tremendous social burden (*Nandi et al., 2022*). Neuropathological hallmarks, including amyloid-β (Aβ) plaques and neurofibrillary tangles (NFT), define AD biologically, and advances in biomarkers have enabled early diagnosis (*Jack et al., 2024*; *Jack et al., 2018*). Abnormal biomarkers can appear years before symptoms arise, a stage referred to as preclinical AD (*Sperling et al., 2011*; *Sperling et al., 2014*). As the disease progresses, the cognitive symptom gets severe, and their degree defines mild cognitive impairment (MCI) and the onset of AD dementia. Cognitive decline has been reported in a wide range of domains, including memory, language, visuospatial function, and executive function, leading to behavioral change as early as a decade before dementia (*Amieva et al., 2008*; *Knopman et al., 2021*). The early stage of AD provides a time window to intervene in disease progression, yet there is an explanatory gap and inconclusive consensus on what particular behavioral phenotype can be directly linked to neuropathology in early AD. Given the present circumstance, the subtle and heterogeneous cognitive change before the onset of dementia poses a challenge to assess clinical outcomes with traditional measurement (*Jutten et al., 2023*).

To precisely assess clinical symptoms and effects of early intervention for AD pathology, determining universal behavioral phenotypes as well as understanding how such behaviors change from asymptomatic to symptomatic is needed. Memory impairment is the predominant first symptom of AD (*Barnes et al., 2015*). The symptoms are commonly identified as a deficit in recalling previously learned or recognized items (*Grober et al., 2018*). Previous studies suggested that the performance of prioritized recall, in which memories are recalled in order of assigned value, is a sensitive measure in the early stages of AD (for review, see *Knowlton and Castel, 2022*). It remains elusive how memory impairments may arise from biased information processing that could affect the encoding, maintaining, or retrieval process of memory function. Using genetic mouse models is a common approach to dissecting the effects of pathological changes on cognitive symptoms measured in behavioral tasks while controlling for potential confounding factors (*Webster et al., 2014*; *Zhong et al., 2024*). However, the symptoms at the early stage of AD are relatively mild in traditional measurements, and phenotypes may be inconsistent among studies even when using the same AD mouse model (*Jankowsky and Zheng, 2017*). As behavioral phenotypes in conjunction with brain pathology states may not be captured by current assessments robustly, it is crucial to seek more sensitive and comprehensive approaches.

Recently proposed computational models provide generative processes underlying observed behavior, potentially bridging the gap between behavioral phenotype and AD pathology. Indeed, such models not only replicated the memory task performance of AD patients in different disease stages (*Lee et al., 2020a*; *Lee and Stark, 2023*; *Pooley et al., 2011*) but also were valuable in detecting cognitive decline in preclinical AD patients (*Bock et al., 2021*; *Vanderlip et al., 2024*). However, previous approaches did not suppose an internal model of the world to predict the future from current observations given prior knowledge. Internal models with sufficient explanatory power can replicate various behavioral phenotypes in specific tasks through a common generative process. Estimating an internal model in the early stages of AD patients would elucidate how AD alters their internal states and thereby their behaviors along with disease progression, which is informative for precise diagnosis and assessment of early intervention effects. To our knowledge, only a few studies tried to estimate internal states in amnestic MCI and AD patients and reported that a certain retention rate underlying motor learning (*Sutter et al., 2024*), but not learning rate in associative learning (*Wessa et al., 2016*) differed from healthy controls. This circumstance demands us to investigate internal models illustrating not only learning but also its interaction with memory with higher explanatory power.

In this study, we assumed the latent cause model as an internal model explaining the memory modification process to solve a problem such that a novel observation should be classified into previously acquired memory or novel memory (*Gershman et al., 2017b*). We hypothesized that the internal state of the latent cause model is altered along with disease progression from the preclinical stage of AD. To test this, we used the *App*<sup>NL-G-F</sup> knock-in AD mouse model (*Saito et al., 2014*). This model of mice carries the Swedish KM670/671 NL, Iberian I716F, and Arctic E693G mutations on the gene of amyloid-β precursor protein (APP) and displays Aβ plaque accumulation and neuroinflammation across age, reproducing neuropathological change in preclinical AD without artifacts caused by APP overexpression (*Saito et al., 2014*; *Sasaguri et al., 2017*). In this study, the 6- and 12-month-old *App*<sup>NL-G-F</sup> knock-in mice and the age-matched control mice were engaged in a memory modification task consisting of Pavlovian fear conditioning, extinction, and reinstatement. Internal states across trials were estimated for each mouse, simultaneously with determining a set of model parameters such that the simulated behavior under the latent cause model given the parameters fits enough to the observed behavior. Because we found a deficit in the *App*<sup>NL-G-F</sup> mice as overgeneralization or over-differentiation of their observations into certain memories indicated by parameters in the latent cause model, we further confirmed whether this deficit could be replicated in another kind of memory modification task, the reversal Barnes maze paradigm. In the latent cause framework, we discussed how the internal states in the *App*<sup>NL-G-F</sup> mice diverged from those in the control mice during the memory modification process.

## Results

### The latent cause model, an internal model, explains memory modification processes in the classical fear conditioning paradigm

The latent cause model proposed by *Gershman et al., 2010* is an internal model such that an agent infers a latent cause as a source of the co-occurrence of events in the environment. It was later adapted to model the memory modification process, a problem to infer previously acquired latent causes or a novel latent cause for every observation in the Pavlovian fear conditioning paradigm in rodents and humans (*Gershman et al., 2017b*). In the latent cause model, an agent learning an association between a conditioned stimulus (CS) and an unconditioned stimulus (US) has an internal model such that stimuli $\mathbf{x}$ (i.e., $d$-dimensional vector involving the CS or context) and outcome $r$ (i.e., US) at time $t$ are generated at an associative strength $\mathbf{w}$ (i.e., $d$-dimensional vector) from $k$-th latent cause $z_{tk}$ of total $K$ latent causes by a likelihood $p(\mathbf{x}|z_{tk})$ and $p(r|z_{tk})$, respectively (*Figure 1A*). $p(\mathbf{x}|z_{tk})$ and $p(r|z_{tk})$ follow Gaussian distributions in which each mean is calculated from previous observations, while each variance ($\sigma_x^2$ and $\sigma_r^2$) was *a priori* defined as a hyperparameter. Since latent causes are unobservable, the agent given an observation infers which latent cause is likely to generate current observation by Bayes' rule. The prior of $z$ follows the Chinese restaurant process, that is

$$P(z_t = k|\mathbf{z}_{1:t-1}) \propto \begin{cases} \sum_{t' < t} \mathcal{K}(\tau(t) - \tau(t'))\mathbb{I}[z_{t'} = k] & \text{if } k \leq K \text{ (i.e., } k \text{ is an old cause),} \\ \alpha & \text{otherwise (i.e., } k \text{ is a new cause).} \end{cases}$$

Prior of $z_{tk}$ refers to the frequency that $z_k$ was inferred so far, where $\mathbb{I}[\cdot] = 1$ if its argument is true; otherwise, 0. This prior is biased by a temporal kernel , as an analogy of the power law of the forgetting curve,

$$\mathcal{K}\left(\tau\left(t\right) - \tau\left(t'\right)\right) = \left(\tau\left(t\right) - \tau\left(t'\right)\right)^{-g},$$

to exponentially decay the probability that $z_k$ inferred at previous trial $t'$ is inferred again at current trial $t$ by a temporal scaling factor $g$ if the time interval between trial $t'$ and $t$ increases. A new latent cause is inferred at a probability governed by the concentration parameter $\alpha$. Higher $\alpha$ or lower likelihood of the current observation biases the agent to infer a new latent cause given an observation (i.e. differentiation) rather than to infer an old one (i.e., generalization) (*Figure 1B*). Although the latent cause model allows for having an infinite number of latent causes, agents with appropriate $\alpha$ and $g$ favor to infer a smaller number of latent causes for observations instead of inferring a unique latent cause for every observation.

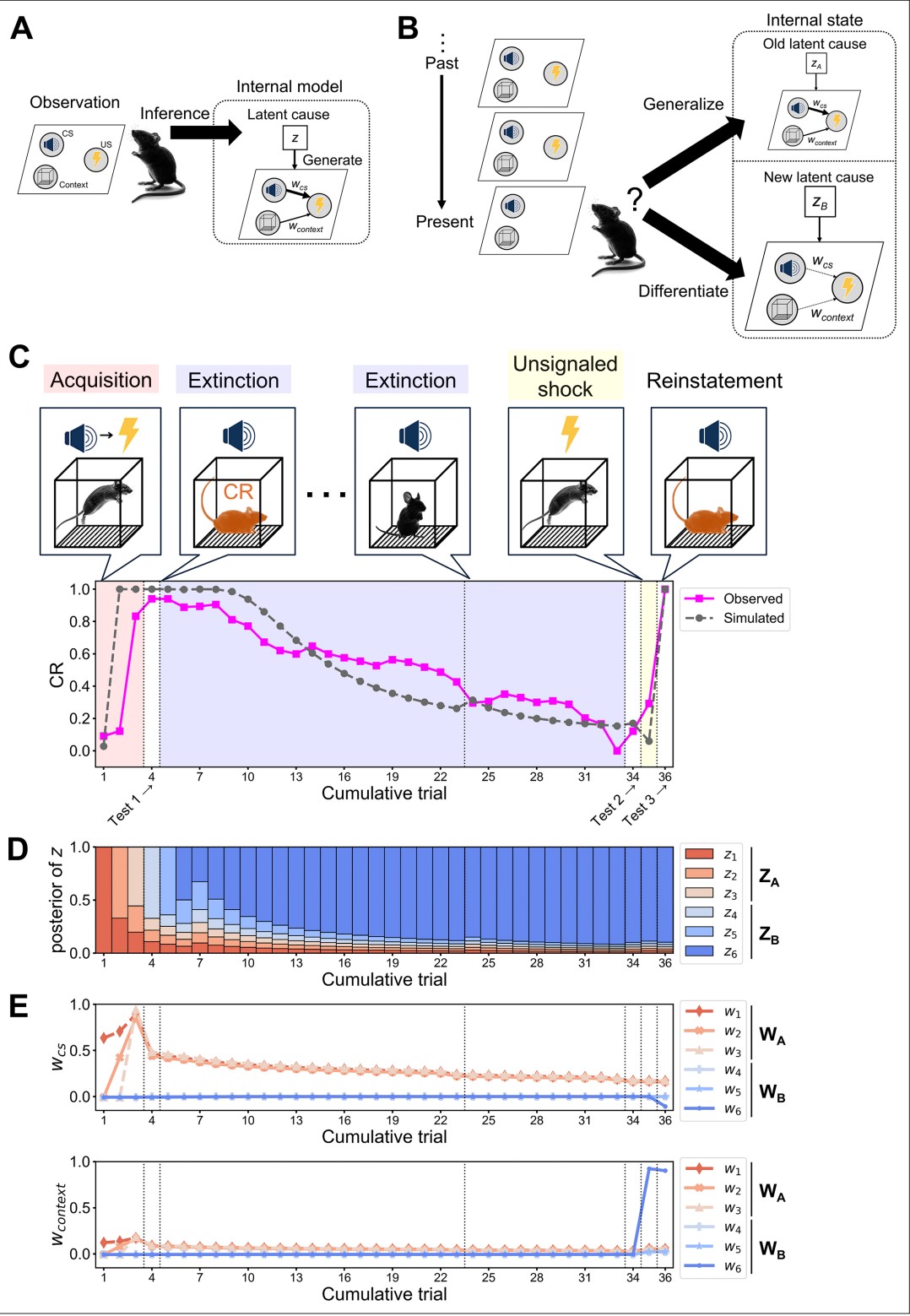

**Figure 1.** Internal state in the latent cause model in the Pavlovian fear conditioning, extinction, and reinstatement. (**A**) Schematic diagram of latent cause framework in a simple Pavlovian fear conditioning paradigm. An experimental mouse observes a tone, a context, and an electrical shock, which are set down in a parallelogram. Among the stimuli, the tone is designed as the conditioned stimulus (CS), and the electrical shock is designed as the unconditioned stimulus (US) with additional consideration of context effect. The mouse has an internal model such that a latent cause $z$ generates the observation where the tone and the context induce the shock

*Figure 1 continued on next page*

*Figure 1 continued*

at an associative weight $w_{cs}$ and $w_{context}$, respectively. The latent cause is a latent variable represented by a white square. The posterior probability of $z$ is represented by a black arrow from $z$ to the observation. The stimuli (i.e., tone, context, and shock) are observed variables represented in shaded circles, and the associative weight $w$ is represented by a black arrow between the two. Since latent causes are unobservable, the mouse infers which latent cause is more likely from posterior distribution over latent causes given the observation following Bayes' rule. (**B**) Schematic diagram of internal state posited in the latent cause model demonstrating the memory modification process in the fear conditioning paradigm. The mouse has acquired $z_A$ through two observations of the CS accompanied by the US in the same context. Now, the mouse observes the tone alone, then infers a new latent cause $z_B$ generating the CS and context without US as $w_{cs}$ and $w_{context}$ equal to 0. The internal state at the time consists of $z_A$ with largely decreased posterior probability and thereby slightly decreased $w_{cs}$ and $w_{context}$, in addition to $z_B$. The mouse in this panel is likely to differentiate the current observation from $z_A$, as it assigns a higher probability for $z_B$. The posterior probability of $z_A$ and $z_B$ is represented by the relative size of their diagram. If the mouse generalizes $z_A$ to the current observation without inferring $z_B$, it would largely decrease $w_{cs}$ and $w_{context}$ of $z_A$ to minimize prediction error. (**C**) Observed and simulated conditioned response (CR) during reinstatement in fear conditioning. In the acquisition phase, a CS was accompanied by a US, inducing associative fear memory in the animal, as it exhibits a higher CR to the CS. In the extinction phase, the CS was presented without the US, inducing extinction memory with decreased CR. After an unsignaled shock without the preceding CS, the CR increases again, and this is called reinstatement. The observed CR was sampled from a 2-month-old male wild-type C57BL/6J mouse (magenta line plot with square markers), where the vertical and horizontal axis indicate CR and cumulative trial, respectively. The latent cause model parameters were estimated so as to minimize the prediction error between the observed and the simulated CR given the parameter (gray line with square markers). The estimated parameter value: $\alpha = 2.4$, $g = 0.316$, $\eta = 0.64$, max. no. of iteration = 9, $w_0 = -0.006$, $\sigma_r^2 = 4.50$, $\sigma_x^2 = 4.39$, $\theta = 0.03$, $\lambda = 0.02$, $K = 6$. Trials with red, blue, and yellow backgrounds correspond to the acquisition, extinction, and unsignaled shock, where the first and second extinction consists of 19 and 10 trials, respectively (see Materials and methods). Three trials with white backgrounds are tests 1, 2, and 3 to evaluate if conditioning, extinction, and reinstatement are established. The vertical dashed lines separate the phases. (**D**) The evolution of the posterior probability and (**E**) the associative weight of each latent cause across the reinstatement experiment, where $w_{cs}$ (top panel) and $w_{context}$ (bottom panel) were computed. (**D, E**) Each latent cause is indicated by a unique subscript number and color, $w_{cs}$, and $w_{context}$. The latent causes acquired during the acquisition phase and their associative weight are termed $\mathbf{z}_A$ and $\mathbf{w}_A$, respectively. Meanwhile, latent causes acquired during the extinction phase and their associative weight are termed $\mathbf{z}_B$ and $\mathbf{w}_B$, respectively.

The online version of this article includes the following source data for figure 1:

**Source data 1.** Simulation data and experimental data shown in *Figure 1C–E*.

According to the latent cause model, the agent solves two computational problems during Pavlovian fear conditioning learning. One is to compute the posterior probability of each latent cause given its associative weight between stimuli **x** and shock $r$ with a history of observations, and the other is to compute the associative weight of each latent cause given its posterior. Both computations are achieved through the expectation-maximization (EM) algorithm, as the former and the latter computation correspond to E-step and M-step, respectively. In the E-step, the posterior of each latent cause is computed from its prior and the likelihood of current observation given the latent cause following Bayes' rule, consequently determining the most likely latent cause at the current trial as the one with the maximal posterior, then moves to the M-step. In the M-step, the agent predicts shock by a linear combination of stimuli and associative weight in each latent cause. The weight is updated based on the Rescorla-Wagner rule (*Rescorla and Wagner, 1972*) to minimize the prediction error between observed and predicted shock. The amount of the weight change depends on learning rate $\eta$ and prediction error, which is biased by the posterior probability of the latent causes. The amount of weight change will be smaller even in larger prediction errors if the latent cause is unlikely. The agent estimates weights for all latent causes and then returns to E-step. This procedure is iterated a preset number of times, and thereby, the posterior probability over latent causes and their associative weights are determined. Finally, the agent concludes the expected shock in the current trial by a linear combination of observed stimuli and an expectation of weight, which is weighted by the posterior distribution over latent causes. The conditioned response (CR) for the expected shock is determined as an integral of Gaussian distribution (mean is the expected shock, variance is $\lambda$) above threshold $\theta$, where $\lambda$ and $\theta$ are hyperparameters. Thus, these parameters of the latent cause model determine an internal state of the agent leading to particular behavioral outcomes given an observation.

Reinstatement exemplifies the memory modification process and indeed is well explained by the latent cause model. Reinstatement occurs when agents who have learned and then extinguished an association between a CS and a US exhibit CR again after being exposed to the US alone following the extinction. The latent cause model explains the reinstatement of fear memory with the changes of agent's internal state as follows. First, an agent repeatedly observed a CS accompanied with a US in a Pavlovian fear conditioning procedure (*Figure 1C*, 'Acquisition'). Because these observations are novel for the agent, it inferred a set of latent causes $\mathbf{z}_A$ ($z_1$, $z_2$, and $z_3$ in *Figure 1D*) where the CS is associated with the US by associative weight $\mathbf{w}_A$ ($w_1$, $w_2$, and $w_3$ in *Figure 1E*) during the acquisition. Considering the effect of the context, $\mathbf{w}$ contains both $w_{cs}$ and $w_{context}$ (*Figure 1E*). After the acquisition, the agent inferred $\mathbf{z}_A$ and then predicted the upcoming US, leading to increased CR when it observed the CS (*Figure 1C*, 'Test 1' after 'Acquisition'). Second, the agent repeatedly observed the CS alone in an extinction procedure. Early in the extinction, the agent would infer $\mathbf{z}_A$ and then exhibit higher CR to the CS, while the US was absent (*Figure 1C*, left 'Extinction'). This state increases prediction error, then prompts the agent either to decrease $\mathbf{w}_A$ toward 0 or to infer a novel set of latent causes $\mathbf{z}_B$ where the CS is not accompanied with the US, that is, associative strength $\mathbf{w}_B = 0$ (*Figure 1B*); both processes are run in the mouse in *Figure 1CDE*. Note that the sets A and B were introduced arbitrarily for explanation here. Later in the extinction, $\mathbf{z}_B$ rather than $\mathbf{z}_A$ is inferred, leading to the decreased CR (*Figure 1C*, right 'Extinction' or 'Test 2'). Third, when the agent observes the US alone after the extinction (*Figure 1C*, 'Unsignaled shock'), the probability of $\mathbf{z}_A$ or associative weight of some latent causes increases, hence it predicts the US and thereby the CR increases again, that is reinstatement (*Figure 1C*, 'Reinstatement' at 'Test 3'); in the mouse in *Figure 1CDE*, the unsignaled shock increases not only the posterior probability of $\mathbf{z}_A$ that has maintained higher $w_{cs}$ but also $w_{context}$ in $\mathbf{w}_6$ interpreted as a modification in the association between context and US in $z_6$. Thus, it is rational to expect that certain parameters of the latent cause model are in conjunction with particular behaviors in the reinstatement paradigm (see Appendix 2 for the relationship between parameters, CR, and internal states). As demonstrated in *Gershman and Hartley, 2015*, the likelihood that $\alpha > 0$ in human participants correlates with the degree of their spontaneous recovery, that is, a return of the CR after a long hiatus following an extinction procedure. Furthermore, a recent study has shown that the latent cause model explains maladaptive decision-making in post-traumatic stress disorder patients as a perseverated inference of previously acquired latent causes even under situations where they should infer novel latent causes (*Norbury et al., 2022*). By estimating such parameters from the behaviors of each experimental mouse, how the internal state of the AD group diverges from that of the control group could be determined.

## *App$^{NL-G-F}$* mice showed unimpaired associative fear and extinction learning but a lower extent of reinstatement

To examine the age and *App$^{NL-G-F}$* knock-in effects on the reinstatement, all cohorts of control and homozygous *App$^{NL-G-F}$* mice were tested at the age of either 6 months or 12 months in the reinstatement paradigm based on auditory-cued fear conditioning (*Figure 2A* and *Supplementary file 1a*). According to previous studies, the *App$^{NL-G-F}$* mice displayed an age-dependent Aβ deposition from 4 months old and memory impairments could be detected at 6 months of age (*Masuda et al., 2016*; *Saito et al., 2014*; *Sakakibara et al., 2018*). The widespread of Aβ accumulation was confirmed in this study, and results were provided in *Figure 2—figure supplement 1*. Given that age affects the Aβ accumulation caused by *App$^{NL-G-F}$* knock-in, a one-way ANCOVA was used to evaluate the *App$^{NL-G-F}$* knock-in effects on behavioral measures controlling for age.

In the acquisition phase of fear conditioning, the freezing rate gradually increased with the number of CS-US pairings (*Figure 2B*, leftmost red panel). In the test 1 phase, 24 hr after the acquisition, a significant genotype effect was observed in freezing rates during CS presentation [$F(1,95) = 9.679$, $p = 0.002$], where the *App$^{NL-G-F}$* mice showed a higher freezing rate than in the control (*Figure 2C* and *Supplementary file 1b*), indicating the associative fear memory was established in *App$^{NL-G-F}$* mice. As the increasing trials of CS alone presented, both groups showed a gradually decreased freezing rate in the extinction phase (*Figure 2B*, blue panel). In the test 2 phase, 24 hr after the last trial of the extinction, there was a significant age effect in freezing rates during CS presentation [$F(1,95) = 4.698$, $p = 0.033$], where 12-month-old mice showed a higher freezing rate than in the 6-month-old mice (*Figure 2D* and *Supplementary file 1b*). In the test 3 phase, 24 hr after given an unsignaled shock, mice showed

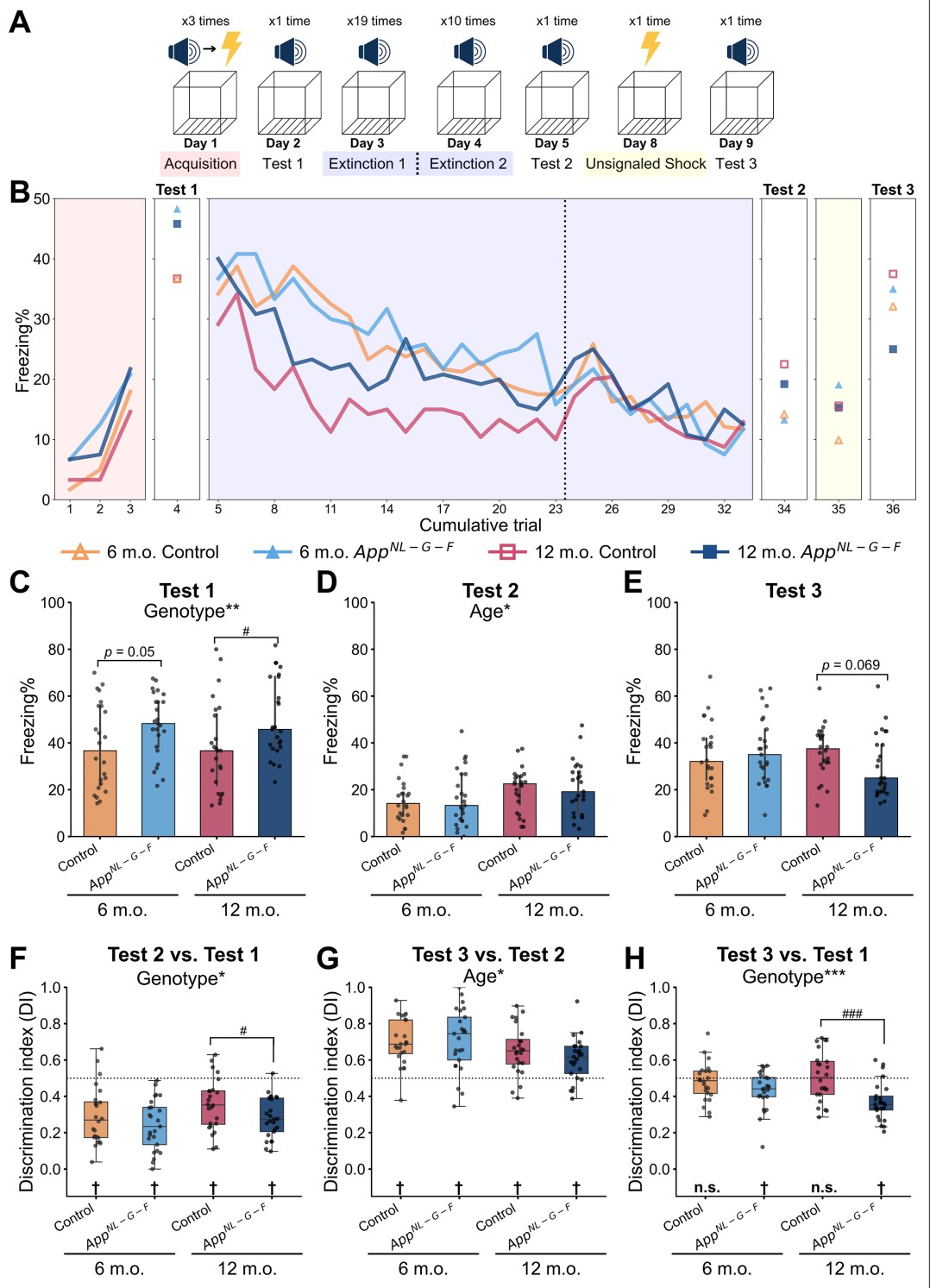

**Figure 2.** $App^{NL-G-F}$ mice exhibited successful associative learning and extinction but a lower extent of reinstatement. (**A**) Schematic diagram of reinstatement paradigm in this study. In the acquisition phase, the conditioned stimulus (CS) was accompanied with the unconditioned stimulus (US) three times. In the following phase, either the CS or the US was presented at certain times. The context was the same throughout the experiment. (**B**) Freezing rate during CS presentation across reinstatement paradigm. The markers and lines show the median freezing rate of each group in each trial. Red, blue, and yellow backgrounds represent acquisition, extinction, and unsignaled shock in (**A**). The dashed vertical line separates the extinction 1 and extinction 2 phases. Freezing rate during CS presentation in test 1 (**C**), test 2 (**D**), and test 3 (**E**). Discrimination index (DI)

*Figure 2 continued on next page*

*Figure 2 continued*

between test 2 and test 1 (**F**), between test 3 and test 2 (**G**), and between test 3 and test 1 (**H**) calculated from freezing rate during CS presentation. (**C–E**) The data are shown as median with interquartile range. $*p < 0.05$, $**p < 0.01$, and $***p < 0.001$ by one-way ANCOVA with age as covariate; $^{\#}p < 0.05$, $^{\#\#}p < 0.01$, and $^{\#\#\#}p < 0.001$ by Student's *t*-test comparing control and $App^{NL-G-F}$ mice within the same age. Detailed statistical results are provided in **Supplementary file 1b**. (**F–H**) The dashed horizontal line indicates DI = 0.5, which means no discrimination between the two phases. $^{\dagger}p < 0.05$ by one-sample Student's *t*-test, and the alternative hypothesis specifies that the mean differs from 0.5; $*p < 0.05$, $**p < 0.01$, and $***p < 0.001$ by one-way ANCOVA with age as a covariate; $^{\#}p < 0.05$, $^{\#\#}p < 0.01$, and $^{\#\#\#}p < 0.001$ by Student's *t*-test comparing control and $App^{NL-G-F}$ mice within the same age. Detailed statistical results are provided in **Supplementary file 1c and d**. (**B–H**) Colors indicate different groups: orange represents 6-month-old control (*n* = 24), light blue represents 6-month-old $App^{NL-G-F}$ mice (*n* = 25), pink represents 12-month-old control (*n* = 24), and dark blue represents 12-month-old $App^{NL-G-F}$ mice (*n* = 25). Each black dot represents one animal.

The online version of this article includes the following source data and figure supplement(s) for figure 2:

**Source data 1.** Freezing % and Aβ plaque quantification data shown in *Figure 2* and *Figure 2—figure supplements 1 and 2*.

**Figure supplement 1.** Representative image of immunofluorescence staining using Iba1 (blue), GFAP (green), and Aβ (red) antibodies on anterior (**A**) and posterior (**B**) coronal brain sections.

**Figure supplement 2.** Freezing rate before tone presentation in the reinstatement paradigm.

an elevated freezing rate, and no significant genotype or age effect was detected (*Figure 2B and E* and *Supplementary file 1b*). To evaluate the effect of the context on the CR, we measured the preCS freezing rate during 1 min before the tone presentation (*Figure 2—figure supplement 2*). Generally, the preCS freezing rate was lower than CS freezing rate and gradually reduced but kept positive across trials after acquisition (*Figure 2—figure supplement 2A and B*), suggesting the context has a certain effect on CR. When we compared the preCS freezing rate (*Figure 2—figure supplement 2E*) and the CS freezing rate (*Figure 2E*) in test 3 within group by the paired samples *t*-tests, the CS freezing rate was significantly higher than the preCS freezing rate in all groups: 6-month-old control, $t(23) = -6.344$, $p < 0.001$, $d = -1.295$; 6-month-old $App^{NL-G-F}$, $t(24) = -4.679$, $p < 0.001$, $d = -0.936$; 12-month-old control, $t(23) = -4.512$, $p < 0.001$, $d = -0.921$; 12-month-old $App^{NL-G-F}$, $t(24) = -2.408$, $p = 0.024$, $d = -0.482$. Although the associative strength between the context and the US would be increased by the unsignaled shock, these results suggest that reinstatement was not solely explained by the CR for context, but also reactivation acquisition memory.

To evaluate the magnitude of how each mouse can discriminate between different phases, we calculated the discrimination index (DI) using freezing rates during the CS presentation. If the mouse shows the same freezing rate at the two test phases, the value of DI will be 0.5, while if the mouse shows a higher freezing rate in one phase compared to the other, DI will be far from 0.5. The results showed that the DIs between test 2 and test 1 were significantly lower than 0.5 in all groups (*Figure 2F* and *Supplementary file 1c*), indicating the successful establishment of extinction memory. The genotype effect was detected [$F(1,95) = 5.013$, $p = 0.027$] (*Figure 2F* and *Supplementary file 1d*), where $App^{NL-G-F}$ mice showed lower values due to higher freezing in the test 1 phase. These results suggested that there was no markable fear and extinction learning deficit in $App^{NL-G-F}$ mice at the behavioral level. The DIs between the test 3 and 2 phases in four groups were significantly higher than 0.5, indicating that the reinstatement was successfully induced after the unsignaled shock (*Figure 2G* and *Supplementary file 1c*). Six-month-old mice showed higher DIs, as a significant age effect was detected [$F(1,95) = 7.480$, $p = 0.007$] (*Figure 2G* and *Supplementary file 1d*). The DIs between the test 3 and 1 phases did not deviate from 0.5 for both 6- and 12-month-old control mice, suggesting that they displayed comparable freezing rates in tests 1 and 3 (*Figure 2H* and *Supplementary file 1c*). In contrast, the DIs between test 3 and 1 phases were significantly lower than 0.5 in 6-month-old $App^{NL-G-F}$ mice [$t(24) = -3.245$, $p = 0.003$] and 12-month-old $App^{NL-G-F}$ mice [$t(24) = -6.165$, $p < 0.001$] (*Figure 2H* and *Supplementary file 1c*). Moreover, a significant genotype effect was detected [$F(1,95) = 15.393$, $p < 0.001$] (*Figure 2H* and *Supplementary file 1d*), indicating $App^{NL-G-F}$ mice exhibited a lower extent of reinstatement. Thus, these results suggest that mice retrieved fear memory in the test 1 phase and retrieved extinction memory in the test 2 phase, respectively. The fear memory and the extinction memory compete with each other: the US is present in the former, while it is absent in

the latter. In the test 3 phase, after they observed the unsignaled shock, mice had to decide which memory was more relevant given the CS. The same level of freezing rate in tests 1 and 3 observed in control mice suggests that they could retain both fear and extinction memory and thereby infer the fear memory more preferentially. In contrast, $App^{NL-G-F}$ mice displayed a lower freezing rate in test 3 compared to test 1, suggesting that they might still infer extinction memory even after the unsignaled shock or a completely new memory, while the initial fear memory might be suppressed or eliminated.

## The internal state underlying reinstatement simulated with the latent cause model differs between the ages and *APP* genotype

To seek the internal state of each mouse generating the behaviors in the reinstatement paradigm, we estimated the parameters of the latent cause model, minimizing prediction errors between the observed CR and simulated CR in the latent cause model given a set of parameters (see also Materials and methods). *Figure 3* demonstrates the traces of observed CR, simulated CR, and the changes in the internal state in each group. We initially confirmed that the DIs in the simulated CRs in each group well replicated those in the observed CRs, except that the DI between test 3 and test 1 in the 6-month-old $App^{NL-G-F}$ mice did not significantly deviate from 0.5 in simulated results (c.f. *Figure 2F–H*, *Figure 3—figure supplement 1A–C*). To check if the latent cause model has a certain bias to simulate CR in either group, we compared the residual (i.e., observed CR minus simulated CR) between groups. The fit was similar between control and $App^{NL-G-F}$ mice groups in the test trials, except test 3 in the 12-month-old group (*Figure 3—figure supplement 1D and E*). In test 3, the residual was significantly higher in the 12-month-old control mice than $App^{NL-G-F}$ mice, indicating the model underestimated the reinstatement in the control mice. These results suggest that the latent cause model fits our data with little systematic bias, such as an overestimation of CR for the control group in the reinstatement, supporting the validity of the comparisons in estimated parameters between groups.

In the 6-month-old group, two latent causes ($z_1$ and $z_2$) were generated in the acquisition phase with increasing associative weight (rows 2–4 in *Figure 3A and B*). During the extinction phase, the same latent causes were inferred, and the corresponding associative weights declined, indicating the devaluation of previously acquired fear memory (rows 2–4 in *Figure 3A and B*). Notably, the $w_{cs}$ in $\mathbf{w}_2$ in control mice remained at a higher value compared to that in $App^{NL-G-F}$ mice. At trial 35, given an unsignaled shock, $w_{context}$ in $\mathbf{w}_1$ was elevated in control mice (row 4 in *Figure 3A*), while both $w_{context}$ in $\mathbf{w}_1$ and $\mathbf{w}_2$ were elevated in $App^{NL-G-F}$ mice (row 4 in *Figure 3B*). At trial 36 (test 3 phase), in control mice, higher values in $w_{context}$ in $\mathbf{w}_1$ and $w_{cs}$ in $\mathbf{w}_2$ contributed to increased CR (*Figure 3A*), whereas $w_{context}$ in $\mathbf{w}_1$ and $\mathbf{w}_2$ were the main components for reinstatement in $App^{NL-G-F}$ mice (*Figure 3B*). This discrepancy suggests that the expectation of the US is different between 6-month-old control and $App^{NL-G-F}$ mice.

The transition of internal state in the 12-month-old group was similar to that in the 6-month-old group until the test 1 phase. Unlike the 6-month-old group, new latent causes ($z_3$ and $z_4$) were inferred during the extinction phase in the 12-month-old group (rows 2–4 in *Figure 3C and D*). At trial 35 given an unsignaled shock, the control mice inferred the latent causes $z_3$ and $z_4$ acquired in the extinction phase so that their weights $w_{context}$ in $\mathbf{w}_3$ and $\mathbf{w}_4$ were updated (row 4 in *Figure 3C*). In contrast, there was virtually no update of weights in $App^{NL-G-F}$ mice (row 4 in *Figure 3D*). At trial 36 (test 3 phase), the elevated CR in control mice was attributed to a successful update of associative weight after the unsignaled shock (*Figure 3C*), whereas the CR in $App^{NL-G-F}$ mice did not increase due to a redundant latent cause $z_5$, which was newly inferred and reduced the posterior probability of latent causes having higher associative weight (*Figure 3D*).

## The unsuccessful state inference at reinstatement was caused by misclassification in *App^{NL-G-F}* mice

Due to the different evolution of latent causes between ages (*Figure 3*), we separately discuss the contribution of differences in estimated parameters within the same age and their correlations with DI between test 3 (after unsignaled shock) and test 1 (after acquisition) (*Figure 2H*). We first investigated the individual internal state differences in the 12-month-old group, where the impairment was apparent at the behavioral level (*Figure 2H*). $App^{NL-G-F}$ mice had significantly lower $\alpha$ (Mann-Whitney $U = 100$, $p = 0.002$) and lower $\sigma_x^2$ (Mann-Whitney $U = 88.5$, $p = 0.008$) than control mice (*Figure 4A*). $\alpha$ and $\sigma_x^2$ were significantly correlated each other (*Figure 4—figure supplement 1*). Both were also

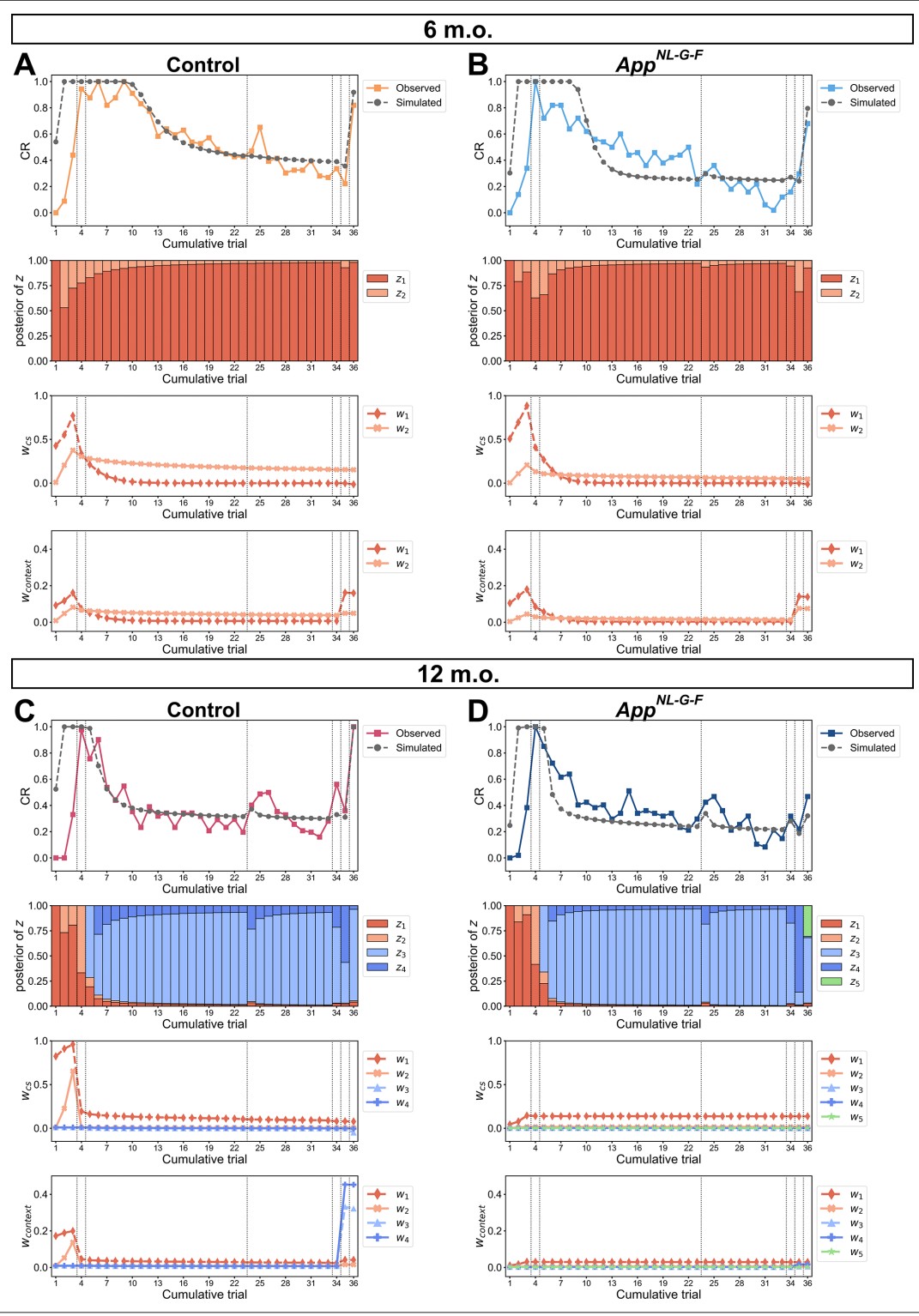

**Figure 3.** The divergence of internal states between control and $App^{NL-G-F}$ mice differed in age. (**A**) Simulation of reinstatement in the 6-month-old control mice given a set of estimated parameters of the latent cause model. The estimated parameter value: $\alpha$ = 2.4, $g$ = 0.05, $\eta$ = 0.42, max. no. of iteration = 2, $w_0$ = 0.009, $\sigma_r^2$ = 2.90, $\sigma_x^2$ = 0.49, $\theta$ = 0.008, $\lambda$ = 0.018, $K$ = 10. (**B**) Simulation of reinstatement in 6-month-old $App^{NL-G-F}$ mice. The estimated parameter value: $\alpha$ = 1.1, $g$ = 0.61, $\eta$ = 0.51, max. no. of iteration = 2, $w_0$ = 0.004, $\sigma_r^2$ = 1.51, $\sigma_x^2$ = 0.32, $\theta$ = 0.015, $\lambda$ = 0.019, $K$ = 30. (**C**) Simulation in 12-month-old control mice. The estimated parameter value: $\alpha$ = 1.1, $g$ = 0.92,

*Figure 3 continued on next page*

*Figure 3 continued*

$\eta$ = 0.82, max. no. of iteration = 5, $w_0$ = 0.009, $\sigma_r^2$ = 1.88, $\sigma_x^2$ = 0.51, $\theta$ = 0.010, $\lambda$ = 0.017, $K$ = 4. (**D**) Simulation in 12-month-old *App^NL-G-F* mice. The estimated parameter value for: $\alpha$ = 1.0, $g$ = 1.10, $\eta$ = 0.04, max. no. of iteration = 2, $w_0$ = 0.0025, $\sigma_r^2$ = 1.51, $\sigma_x^2$ = 0.18, $\theta$ = 0.011, $\lambda$ = 0.011, $K$ = 36. (**A–D**) The first row shows the trace of observed conditioned response (CR) and simulated CR. The observed CR is the median freezing rate during the conditioned stimulus (CS) presentation over the mice within each group; the observed CR of each group was divided by its maximum over all trials. The second row shows the posterior probability of each latent cause in each trial. The third and fourth rows show the associative weight of tone to shock ($w_{cs}$) and that of context to shock ($w_{context}$) in each trial. Each marker and color corresponds to a latent cause up to 5. Each latent cause is represented by the same color as that in the second row and contains $w_{cs}$ and $w_{context}$. The vertical dashed lines indicate the boundaries of phases.

The online version of this article includes the following source data and figure supplement(s) for figure 3:

**Source data 1.** Simulation data and median of experimental data shown in *Figure 3*.

**Figure supplement 1.** Replication of DI in the test phases and prediction error across trials in the latent cause model.

**Figure supplement 1—source data 1.** Simulation data shown in *Figure 3—figure supplement 1*.

---

positively correlated with the DI (*Figure 4B*), and the trend was preserved after adjusting for the genotype effect (*Figure 4—figure supplement 2*). While the maximal number of inferable latent causes ($K$) was significantly higher in *App^NL-G-F* mice than in control mice (*Supplementary file 1g*), the total number of acquired latent causes was comparable between the groups ($K_{total}$ in *Supplementary file 1i*). $K$ was negatively correlated with $\alpha$ and $\sigma_x^2$ (*Figure 4—figure supplement 1*).

We assume that the effect of $K$ on the memory modification process together with $\alpha$ and $\sigma_x^2$ could come to the surface in extremely artificial conditions, but not in natural conditions as in the empirical data (see *Appendix 2—table 6*). Statistical results for remaining parameters are shown in *Supplementary file 1g and h*. It should be noted that these estimated parameters and internal state might not be unique but one of the possible solutions.

We further characterized how the latent causes acquired in different phases contribute to the distinct internal states between control and *App^NL-G-F* mice. Lower $\alpha$ and lower $\sigma_x^2$ would bias *App^NL-G-F* mice toward overgeneralization, as they favor inferring a small number of latent causes when facing the same cue. Indeed, the number of latent causes initially inferred at the acquisition phase was significantly lower in *App^NL-G-F* mice, where most of them inferred two latent causes, and more than half of the control mice inferred three latent causes (*Figure 4C* and *Supplementary file 1i*). Furthermore, when they initially observed the CS in the absence of the US in test 1, the posterior probabilities of acquisition latent causes in test 1 were significantly higher in the *App^NL-G-F* mice than in the control mice, leading to the higher CRs in test 1 in the simulation (*Figure 4—figure supplement 3*), consistent with the observed CR (*Figure 2C*).

According to the latent cause model, lower $\sigma_x^2$ would bias *App^NL-G-F* mice toward overdifferentiation unless the cue exactly matches to previously observed ones. In other words, they prefer to infer new potential causes when similar observations are presented instead of interpreting them generated from the same distribution. The likelihood of current stimuli given latent causes was calculated from a Gaussian distribution with the mean of stimulus values under the latent cause so far and fixed variance $\sigma_x^2$. Repeated observation of the CS alone during the extinction would sufficiently increase the likelihood of stimuli given the latent causes inferred during the extinction in both *App^NL-G-F* and control mice. This attenuates the need to infer new latent causes and the number of latent causes acquired at the extinction phase was comparable between groups (*Figure 4D* and *Supplementary file 1i*). However, the CS was absent in the unsignaled shock and then presented again in test 3. These observations would decrease the likelihood of stimuli given past latent causes, and consequently increase the probability to infer the new latent causes. This volatility would be more obvious in *App^NL-G-F* mice with lower $\sigma_x^2$. As a result, the number of latent causes acquired after the extinction (i.e., the unsignaled shock and test 3) as well as the sum of posterior probabilities of them were greater in *App^NL-G-F* mice than those in the control mice, despite that the mice received either the US or the CS alone that were the same with those they observed so far (*Figure 4E* and *Supplementary file 1i*).

Such internal states in *App^NL-G-F* mice would diverge the update of associative weight from those in the control mice after extinction (*Figure 4F and G* and *Supplementary file 1i*). Both *App^NL-G-F* and control mice would still infer the extinction latent causes at the unsignaled shock with higher posterior

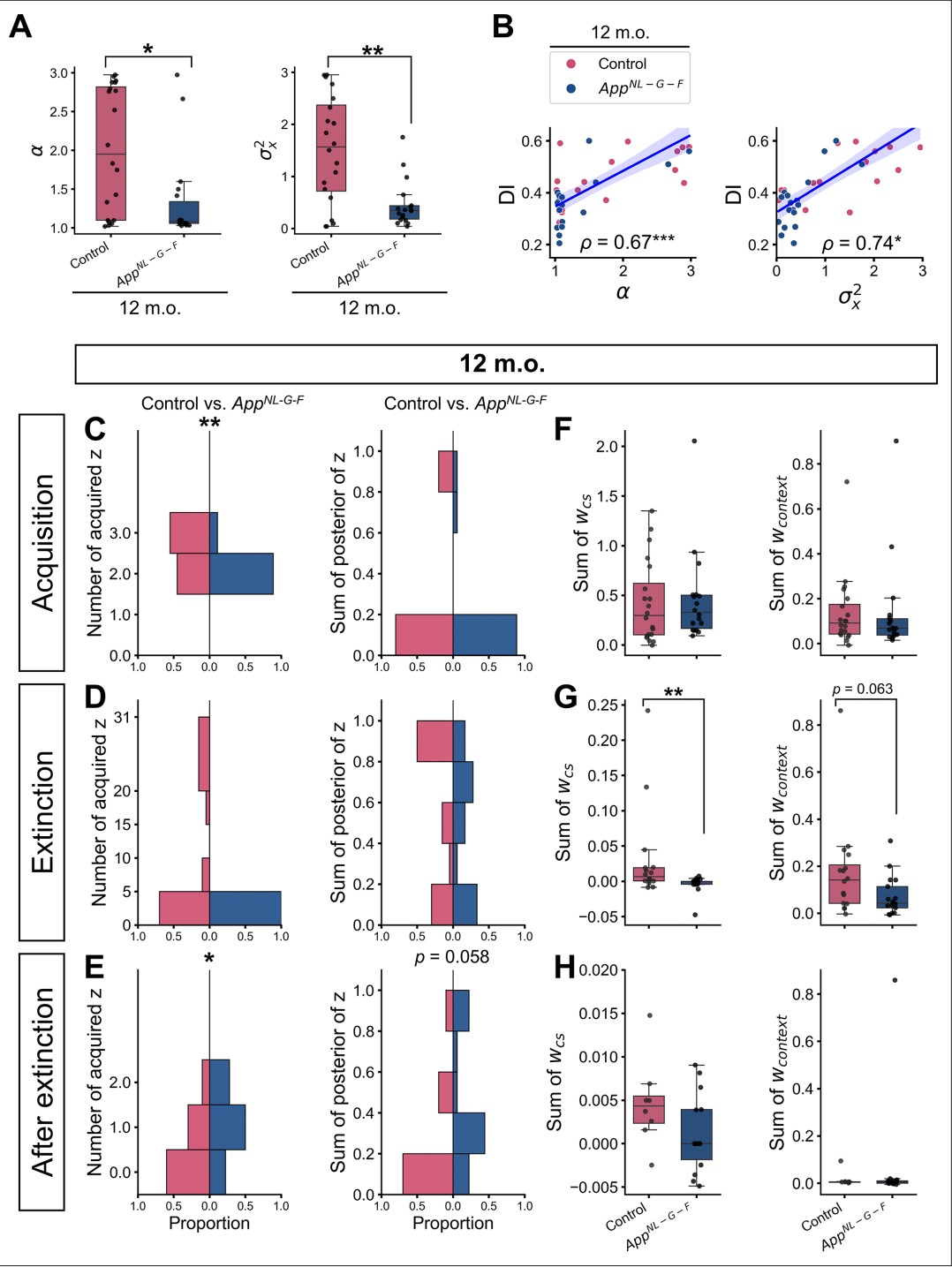

**Figure 4.** Individual parameter estimation and internal state in the 12-month-old group. (**A**) The estimated latent cause model parameter. (**B**) Correlation between discrimination index (DI) and estimated parameter values in the 12-month-old group. The count of latent causes initially inferred during the acquisition trials (**C**, left), extinction trials (i.e., test 1, extinction, test 2) (**D**, left), and trials after extinction (i.e., the unsignaled shock and test 3) (**E**, left), with the sum of posterior probabilities (**C, D, E**, right), and the sum of associative weights at test 3 in these latent causes (**F, G, H**). (**A**) Each black dot represents one animal. $*p < 0.05$, and $**p < 0.01$ by the Mann-Whitney $U$ test. (**B**) The y-axis shows the DI between test 3 and test 1. Spearman's correlation coefficient ($\rho$) was labeled with significance, where $*p < 0.05$ and $***p < 0.001$. The blue line represents the linear regression model fit, and the shaded area indicates the confidence interval. Each dot represents one animal. (**C–E**) In the first column, the histogram of the number of latent causes (z) acquired in each phase is shown. The maximum number of latent

*Figure 4 continued on next page*

*Figure 4 continued*

causes that can be inferred is 3, 31, and 2 in panels C, D, and E. In the second column, the histogram of the sum of the posterior probabilities of the latent causes is shown. The horizontal axis indicates the proportion of the value in each group. (**F–H**). In the first and second columns, the sum of $w_{cs}$ and $w_{context}$ of latent causes are shown in the boxplot, respectively. Note that the initial value of associative weight could take non-zero values, though those were comparable between groups (**Supplementary file 1g**). Each black dot represents one animal. *$p < 0.05$, and **$p < 0.01$ by the Mann-Whitney $U$ test comparing control and $App^{NL-G-F}$ mice, and $p$-values greater than 0.1 were not labeled on the plot. (**A–H**) Pink represents 12-month-old control ($n = 20$), and dark blue represents 12-month-old $App^{NL-G-F}$ mice ($n = 18$). Detailed statistical results are provided in **Supplementary file 1g, h, and i**.

The online version of this article includes the following source data and figure supplement(s) for figure 4:

**Source data 1.** Estimated parameter values of the 12-month-old group.

**Source data 2.** Simulation data shown in *Figure 4C–H* and *Figure 4—figure supplement 3*.

**Figure supplement 1.** Correlation matrix of estimated parameters in the 12-month-old group.

**Figure supplement 2.** Correlations between (DI) and estimated parameters within the 12-month-old control group (**A**) and $App^{NL-G-F}$ group (**B**).

**Figure supplement 3.** Simulated conditioned response (CR) (**A**), and posterior probabilities (**B**), associative weights (**C**) of latent causes initially inferred during the acquisition trials in the 6-month-old group in test 1 phase.

probabilities. Since the update of the associative weight of each latent cause is modulated by both prediction error and posterior probability of the latent cause, the associative weights in the extinction latent causes were increased by the unsignaled shock in the control mice (*Figures 3C and 4G* and *Supplementary file 1i*). In $App^{NL-G-F}$ mice, however, they inferred new latent causes with low associative weight after the extinction (*Figure 4E and H* and *Supplementary file 1i*), which decreased the posterior probabilities of other latent causes and impeded update of their weight (*Figures 3D and 4G*). As the initial weight of new latent causes showed a trend toward significance that control mice showed a higher value (*Supplementary file 1g*), this would explain the significantly higher in $w_{CS}$ in extinction latent cause (*Figure 4G*). This distinct internal state, therefore, led to the lower CR and DI in test 3 in $App^{NL-G-F}$ mice (*Figure 2E and H*). These results suggest that the $\alpha$ and $\sigma_x^2$ would differentiate the internal state and thereby behavioral phenotypes between $App^{NL-G-F}$ and control mice. In the Chinese restaurant process, the probability that a new latent cause is inferred decreases with trials, preventing that too many latent causes are inferred (*Gershman and Blei, 2012*). This could be one of the reasons why the overgeneralization by lower $\alpha$ was less effective than the overdifferentiation by lower $\sigma_x^2$ in test 3 in $App^{NL-G-F}$ mice.

As shown previously in *Figure 2*, *Figure 2—figure supplement 1*, Aβ aggregation has already widely spread in the brain of 6-month-old $App^{NL-G-F}$ mice, but no clear behavioral impairments were detected by the conventional analysis. Similar to the 12-month-old group, we investigated the individual internal state differences in the 6-month-old group and tested whether the discrepancy in $\alpha$ and $\sigma_x^2$ emerged earlier without behavioral impairment. We found that $App^{NL-G-F}$ mice had significantly lower $\alpha$ (Mann-Whitney $U = 38$, $p = 0.012$), and it was significantly correlated with DI (*Figure 5A and B* and *Supplementary file 1j, k*). The $\sigma_x^2$ was comparable between control and $App^{NL-G-F}$ mice, but its significant correlation with DI was found (*Figure 5A and B* and *Supplementary file 1j, k*). Significant positive correlation between $\alpha$ and $\sigma_x^2$ was also found in the 6-month-old group (*Figure 5—figure supplement 1*). Unlike in the 12-month-old group, the significant correlation between $\alpha$ and DI, as well as that between $\sigma_x^2$ and DI was not preserved within subgroups (*Figure 5—figure supplement 2*), suggesting genotype would be confounder for $\alpha$, $\sigma_x^2$, and DI. The hyperparameter $\theta$, a threshold to emit CR for stimulus inputs, was significantly higher in $App^{NL-G-F}$ mice than in control mice. Since the $\lambda$ was comparable between control and $App^{NL-G-F}$ mice (*Supplementary file 1j*), the CR in $App^{NL-G-F}$ mice could rapidly decrease with expected US during the extinction phase, even if their CR at the test 1 was higher than those in the control. The effect of higher $\theta$ was subtle when the expected US was far from its value, and therefore had little contribution to the DI between test 3 and test 1, which is in line with the correlation result in *Supplementary file 1k*. Statistical results for the remaining parameters are shown in *Supplementary file 1j, k*.

Similar to the 12-month-old group, the influence of $\alpha$ was observed in the acquisition phase, where most $App^{NL-G-F}$ mice inferred two latent causes, while more than half of control mice inferred three latent causes (*Figure 5C–H*). In addition, lower $\alpha$ in $App^{NL-G-F}$ mice affect the posterior probabilities

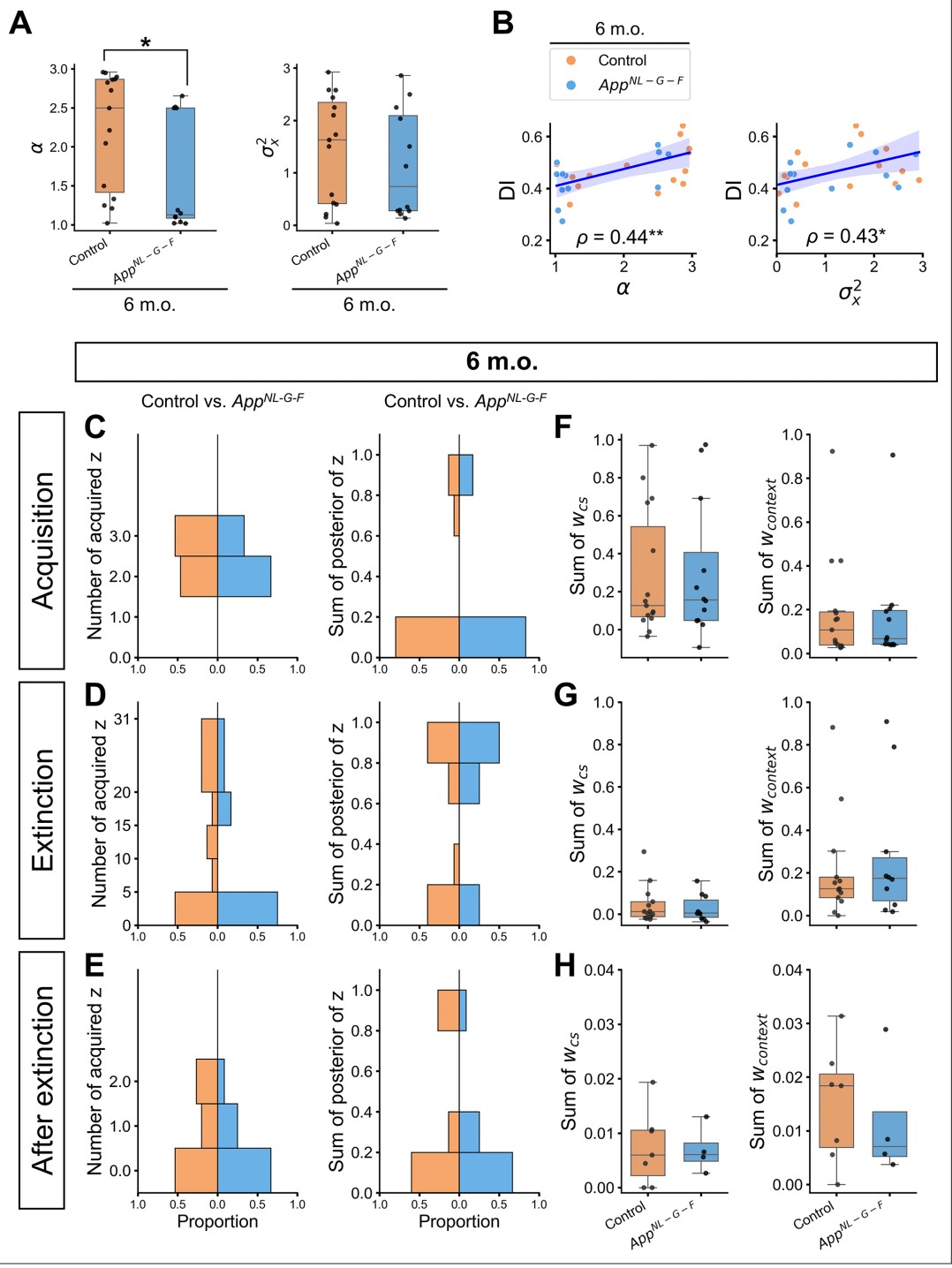

**Figure 5.** Individual parameter estimation and internal state in the 6-month-old group. (**A**) The estimated parameters of the latent cause model. (**B**) Correlation between discrimination index (DI) and estimated parameter values in the 6-month-old group. The count of latent causes initially inferred during the acquisition trials (**C**, left), extinction trials (i.e., test 1, extinction, test 2) (**D**, left), and trials after extinction (i.e., the unsignaled shock and test 3) (**E**, left), with the sum of posterior probabilities (**C, D, E**, right), and the sum of associative weights at test 3 in these latent causes (**F, G, H**). (**A**) Each black dot represents one animal. *$p < 0.05$ by the Mann-Whitney $U$ test. (**B**) The y-axis shows the DI between test 3 and test 1. Spearman's correlation coefficient ($\rho$) was labeled with significance, where *$p < 0.05$ and **$p < 0.01$. The blue line represents the linear regression model fit, and the shaded area indicates the confidence interval. Each dot represents one animal. (**C–H**) The configuration of the figure is the same as in *Figure 4*. All the $p$-values of the Mann-Whitney $U$ test comparing control and $App^{NL\text{-}G\text{-}F}$

*Figure 5 continued on next page*

*Figure 5 continued*

mice were greater than 0.05 and were not labeled on the plot. (**A–H**) Orange represents 6-month-old control (*n* = 15), and light blue represents 6-month-old *App*^NL-G-F mice (*n* = 12). Detailed statistical results are provided in *Supplementary file 1j, k, and l*.

The online version of this article includes the following source data and figure supplement(s) for figure 5:

**Source data 1.** Estimated parameter values of the 6-month-old group.

**Source data 2.** Simulation data shown in *Figure 5C–H* and *Figure 5—figure supplement 3*.

**Figure supplement 1.** Correlation matrix of estimated parameters in the 6-month-old group.

**Figure supplement 2.** Correlation between DI and selected parameters in 6-month-old control (**A**) and *App*^NL-G-F mice (**B**).

**Figure supplement 3.** Simulated conditioned response (CR) (**A**), and posterior probabilities (**B**), associative weights (**C**) of latent causes initially inferred during the acquisition trials in the 6-month-old group in test 1 phase.

of acquisition latent cause and CR in test 1 phase (*Figure 5—figure supplement 3*). No significant difference in the internal states was found between control and *App*^NL-G-F mice (*Figure 5* and *Supplementary file 1l*).

In summary, $\alpha$ and $\sigma_x^2$ are the main contributors to differential internal states and memory modification processes between control and *App*^NL-G-F mice. Regardless of age, *App*^NL-G-F mice have a defect in forming a new memory even if the same cue comes with a new outcome as in test 1, suggesting overgeneralization associated with lower $\alpha$ (*Figures 4A and 5A*). With a more severe phenotype, 12-month-old *App*^NL-G-F mice were biased to classify similar cues into different causes, which is associated with lower variance of the stimulus, $\sigma_x^2$ (*Figure 4A*). This overdifferentiation would eventuate the lower reinstatement of fear memory (*Figure 2H*). Thus, these results suggest that *App*^NL-G-F mice failed to retain competing memories because of the misclassification of observation into memories, even though each memory accounts for past and present observations.

## *App*^NL-G-F mice failed to infer coexisting memories in the reversal Barnes maze task

To explore the explanatory power of the latent cause framework formalizing mechanisms of associative learning and memory modification, cohorts 4, 5, 6, and 7 were subjected to a reversal Barnes maze task (*Barnes, 1979*) two weeks after the reinstatement experiment (*Supplementary file 1*). The Barnes maze is a commonly used behavioral task to examine spatial learning and memory in Alzheimer's disease model mice (*Webster et al., 2014*). Associative learning is fundamental in spatial learning (*Leising and Blaisdell, 2009*; *Pearce, 2009*). Although we did not make any specific assumptions of what kind of associations were learned in the Barnes maze, the trial-and-error updates of these associations involving sensory preconditioning or secondary conditioning could underlie the learning performance. The first stage contains a 6 day training and 1 day probe test to establish the first spatial memory of the target hole (*Figure 6A*). During the training phase, the mice explored a circular field and were reinforced to find an escape box under a target hole as any one of 12 holes equally spaced around the perimeter of the field (*Figure 6A*). In the 12-month-old group, the number of errors, latency, and travel distance from start to goal decreased across days in the training phase, suggesting successful initial learning (*Figure 6—figure supplement 1A*, and *Supplementary file 1m*). A significant interaction between day and genotype [$F(5, 170) = 2.447$, $p = 0.036$] was observed in the latency where control mice took a longer time to reach the target hole on training days 4, 5, and 6 (*Figure 6—figure supplement 1A*, and *Supplementary file 1m*). In order to estimate what kind of strategies mice used to solve the task relying on their memory, we performed algorithm-based strategy analysis (*Suzuki and Imayoshi, 2017*; *Tachiki et al., 2023*). A moving trajectory in each trial was categorized into one of five spatial strategies following the definition (*Figure 6—figure supplement 2*). As in the conventional analysis above, the usages of the strategies were virtually comparable between *App*^NL-G-F and control mice (*Figure 6B* and *Supplementary file 1n*).

In the probe test phase, the mice explored the field for 5 min without the escape box, which generated a certain prediction error (*Figure 6A*). While the mice retrieved the spatial memory of the target hole in spite of the novel observation, they would stay around the target hole. Consequently, *App*^NL-G-F mice spent significantly less time around the target hole than control mice did in the probe test 1,

as a significant interaction between genotype and holes [$F(11, 374) = 4.22$, $p < 0.001$] was detected (*Figure 6C* and *Supplementary file 1o*). These results suggest that $App^{NL-G-F}$ mice could successfully form a spatial memory of the target hole, while they did not retrieve the spatial memory of the target hole as strongly as control mice when they observed the absence of the escape box during the probe test.

In the first reversal learning phase, the position of the target hole with the escape box was moved to the opposite from day 9–11, then returned to the original position from day 12 to day 14 (*Figure 6A*). Through the first reversal learning, we expected that the second spatial memory of the target hole is formed that competes with the spatial memory of the first target hole. In other words, this reversal training phase allowed the mice to acknowledge that both two holes were possible to be the target hole. The improved performance across reversal training days in the conventional analysis was similar to that of the initial training phase, suggesting intact behavioral flexibility (*Brown and Tait, 2010*; *Gawel et al., 2019*; *Figure 6—figure supplement 1B* and *Supplementary file 1m*). Unexpectedly, 12-month-old $App^{NL-G-F}$ mice reached the goal significantly faster than the control mice [Genotype effect from day 9–11, $F(1,68) = 10.53$, $p = 0.003$; Genotype effect from day 12–14, $F(1,68) = 10.27$, $p = 0.003$] (*Figure 6—figure supplement 1B*, and *Supplementary file 1m*). This would be due to the difference in the strategies primarily taken in each group of mice. During the first reversal training (days 9–11), more than half of $App^{NL-G-F}$ mice used the perimeter strategy, which is significantly higher than control mice (*Figure 6B* and *Supplementary file 1n*). The mice taking this strategy could reach the novel target hole by exploring holes sequentially without spending time around the previous target hole (*Figure 6—figure supplement 2*). On the other hand, the use of perimeter strategy decreased, and spatial strategy increased in control mice from days 9–11 (*Figure 6B*). Moreover, the use of the confirmatory strategy was around 30% in control mice, which is significantly higher than $App^{NL-G-F}$ mice (*Figure 6B* and *Supplementary file 1n*). This strategy is the one such that the mice visit the first and second target holes as if they relied on the two memories of the target holes where the escape box should exist (*Figure 6—figure supplement 2*). During the second reversal training (days 12–14), $App^{NL-G-F}$ mice constantly used the perimeter strategy more than the control mice. Meanwhile, the control mice used the confirmatory strategy more than $App^{NL-G-F}$ mice, as in the first reversal learning phase (*Figure 6B* and *Supplementary file 1n*). These results suggest that the search strategy diverged between $App^{NL-G-F}$ mice and control mice, since the reversal learning. The control mice could retrieve two competing spatial memories of the target hole during each trial. In contrast, $App^{NL-G-F}$ mice were less likely to retrieve these two memories within a trial even though the use of spatial strategy was comparable with control mice on the same day.

As expected, the $App^{NL-G-F}$ mice explored significantly shorter time for the second target hole as well as the first target hole, compared to the control mice (*Figure 6D* and *Supplementary file 1o*). A significant interaction between genotype and hole was detected [Genotype × Hole, $F(11, 374) = 3.24$, $p < 0.001$]. The exploration time was uniform over holes in $App^{NL-G-F}$ mice, while control mice showed hole preferences around two target holes as if the $App^{NL-G-F}$ mice and the control mice took the perimeter and the confirmatory strategy, respectively. At last, we calculated the discrimination index (DI) from the exploration time for the first and second target holes in probe test 2 and test 1. As in the reinstatement experiment, we expected that if the mice did not retrieve a spatial memory of the second target hole during probe test 2, the exploration time of this hole would be the same between probe tests 1 and 2, resulting in DI around 0.5. The DI for the second target hole was significantly higher than 0.5 in control mice, indicating that the mice successfully modified the spatial memories from probe test 1 to probe test 2 at a discriminable level (*Figure 6E* and *Supplementary file 1p*). On the contrary, the DI in $App^{NL-G-F}$ mice was comparable to 0.5 and significantly lower than that of control mice, indicating that their spatial memory for the second target hole was not modified after the reversal learning (*Figure 6E* and *Supplementary file 1p*). We confirmed that the DIs for the first target hole were close to 0.5 and comparable between groups, suggesting that the exploration time for the first target hole remained similarly long in both probe tests (*Figure 6—figure supplement 1C*, and *Supplementary file 1p*).

The distinct learning process observed in the 12-month-old group was also present in the 6-month-old group. The performance shown in the conventional analysis was comparable between the 6-month-old control and $App^{NL-G-F}$ mice (*Figure 6—figure supplement 2A and B*, and *Supplementary file 1q*). The use of perimeter strategy was over 30% throughout two reversal training phases

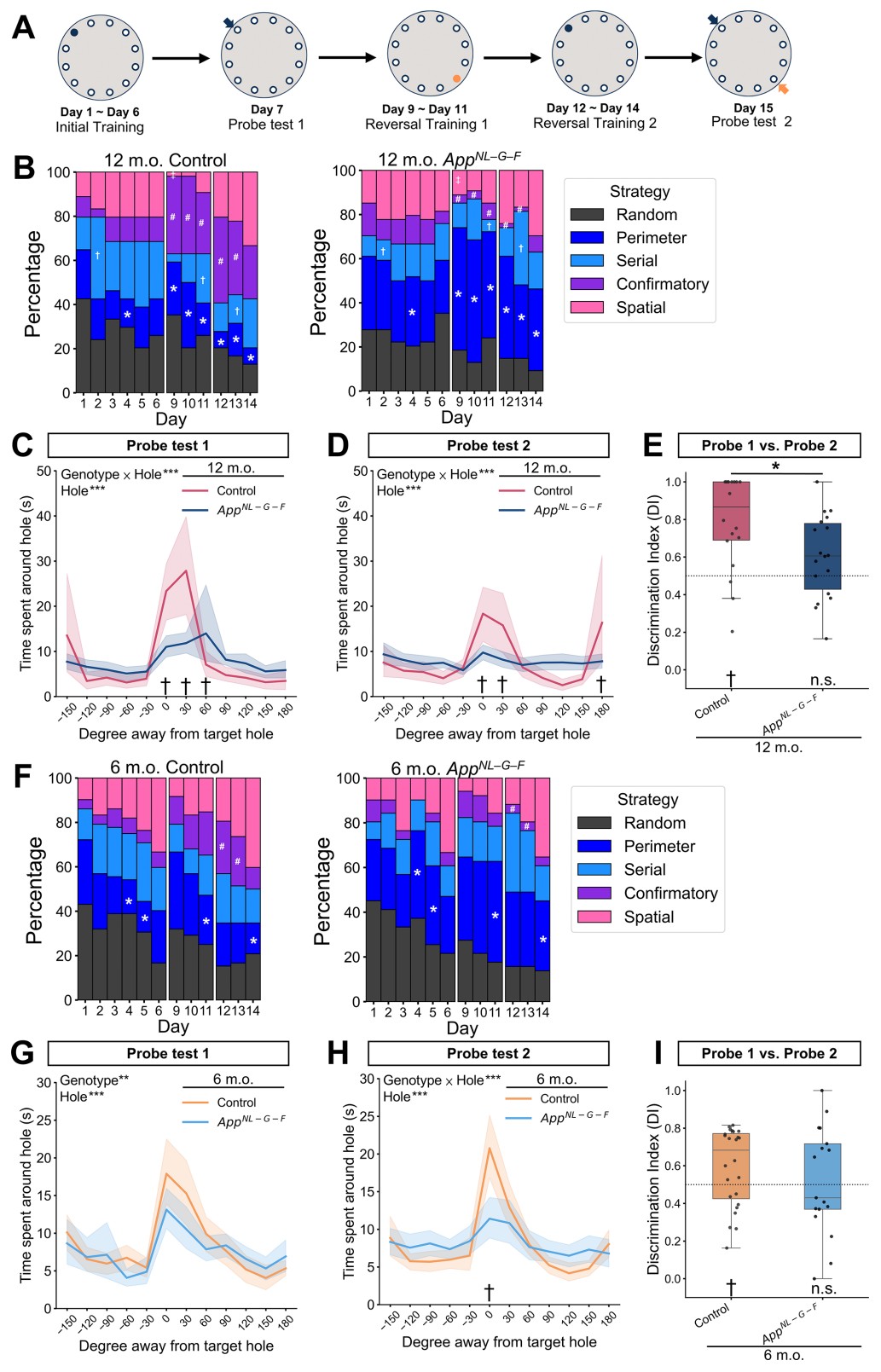

**Figure 6.** $App^{NL-G-F}$ mice failed to infer coexisting spatial memories in the reversal Barnes maze task. (**A**) Schematic diagram of the reversal Barnes maze task. The largest circle represents the Barnes maze field. The filled and open small circles represent the target hole with the escape box and the remaining holes in the field, respectively. The small arrow pointing to the hole indicated the position of the target hole without an escape box in the probe test.

*Figure 6 continued on next page*

*Figure 6 continued*

The same color of the filled circle and arrow indicates the same position. (**B**) Strategy usage in initial training (days 1–6), the first reversal training (days 9–11), and the second reversal training (days 12–14) in the 12-month-old group. Time spent around each hole at probe test 1 (**C**) and probe test 2 (**D**) in the 12-month-old group. (**E**) Discrimination index (DI) of the second target hole between probe test 2 and probe test 1 in the 12-month-old group. (**F**) Strategy usage in initial training (days 1–6), the first reversal training (days 9–11), and the second reversal training (days 12–14) in the 6-month-old group. Time spent around each hole at probe test 1 (**G**) and probe test 2 (**H**) in the 6-month-old group. (**I**) Discrimination index (DI) of the second target hole between probe test 2 and probe test 1 in the 6-month-old group. (**B, F**) *$p < 0.05$ of perimeter strategy, #$p < 0.05$ of confirmatory strategy, †$p < 0.05$ of serial strategy, ‡$p < 0.05$ of spatial strategy by the Wilcoxon rank-sum test comparing control and $App^{NL-G-F}$ mice at the same age. (**C, D, G, and H**) The data are shown as mean with 95% confidence interval. *$p < 0.05$, **$p < 0.01$, and ***$p < 0.001$ by mixed-design two-way [Genotype (control, $App^{NL-G-F}$) × Hole (1-12)] ANOVA; †$p < 0.05$ by Tukey's HSD test at the specific hole to compare control and $App^{NL-G-F}$ mice at the same age. (**E, I**) The dashed horizontal line indicates DI = 0.5, which means no discrimination between the two phases. †$p < 0.05$ by the Wilcoxon signed-rank test, and the alternative hypothesis specifies that the mean differs from 0.5; *$p < 0.05$ by Mann-Whitney $U$ test comparing control and $App^{NL-G-F}$ mice. (**C–E, G–I**) Each black dot represents one animal. Colors indicate the different groups: pink represents 12-month-old control ($n = 18$), dark blue represents 12-month-old $App^{NL-G-F}$ mice ($n = 18$), orange represents 6-month-old control ($n = 24$), and light blue represents 6-month-old $App^{NL-G-F}$ mice ($n = 17$). Detailed statistical results are provided in **Supplementary file 1n, o, and p** for the 12-month-old group results and **Supplementary file 1r, s, and t** for the 6-month-old group results.

The online version of this article includes the following source data and figure supplement(s) for figure 6:

**Source data 1.** Behavioral data of the reversal Barnes maze experiment shown in **Figure 6** and **Figure 6—figure supplements 1 and 3**.

**Figure supplement 1.** Conventional analysis for initial (**A**) and reversal (**B**) training day in the Barnes maze task and discrimination index of the first target hole between probe test 2 and probe test 1 (**C**) in the 12-month-old group.

**Figure supplement 2.** Strategy analysis in the Barnes maze task.

**Figure supplement 3.** Conventional analysis for initial (**A**) and reversal (**B**) training day in the Barnes maze task and discrimination index of the first target hole between probe test 2 and probe test 1 (**C**) in the 6-month-old group.

---

in 6-month-old $App^{NL-G-F}$ mice, while control mice used significantly higher confirmatory strategy in the second reversal training (**Figure 6F**, and **Supplementary file 1r**). During the probe tests, the difference in exploration time around the target hole was less prominent in the 6-month-old group, while $App^{NL-G-F}$ mice uniformly explored the holes as in the 12-month-old $App^{NL-G-F}$ mice, especially in the probe test 2 (**Figure 6G and H**, and **Supplementary file 1s**). Unlike 12-month-old control mice, the 6-month-old control mice did not show a strong preference for the second target hole in probe test 2 (**Figure 6H**). However, their DIs for the first and second target holes were significantly higher than 0.5, suggesting an increased confidence of spatial memory for each target hole (**Figures 3C**, and **Supplementary file 1t**). In $App^{NL-G-F}$ mice, the DIs for the first and second target hole were comparable to 0.5, as those in 12-month-old $App^{NL-G-F}$ mice, suggesting few modifications of the spatial memories (**Figure 6I**, **Figure 6—figure supplement 3C**, and **Supplementary file 1t**).

In summary, these results suggest that the control mice retained competing spatial memories for the first and the second target holes following reversal learning. In contrast, the $App^{NL-G-F}$ mice retained at most one spatial memory during a trial. In other words, the $App^{NL-G-F}$ mice, given the spatial cues, struggled to infer the spatial memory of either the first or second target hole, leading to the rare usage of the confirmatory strategy. Thus, we confirmed the misclassification in the $App^{NL-G-F}$ mice in the reversal Barnes maze paradigm. Although both DIs in the reinstatement (**Figure 2H**) and the Barnes maze experiment (**Figure 6E and I**) suggest that inferred memories in each time point would diverge between the $App^{NL-G-F}$ mice and the control mice, no significant correlations were detected (12-month-old group: Spearman's correlation coefficient = 0.062, $p = 0.738$, N = 32; 6-month-old group: Spearman's correlation coefficient = –0.029, $p = 0.862$, N = 39), potentially due to differential effects of Aβ accumulation on associative and instrumental learning.

## Discussion

Recent advances in disease-modifying intervention have shed light on AD treatment and prevention. Identifying the earliest manifestation of cognitive decline is critical for intervention. However,

the behavioral phenotypes and altering trajectories are heterogeneous in predementia AD patients (*Duara and Barker, 2022*; *Jutten et al., 2023*), yet standard neuropsychological tests have been optimized for detecting stereotypical symptoms near or after the onset of dementia (*Snyder et al., 2014*). Recently proposed generative models successfully contributed to predicting nature-nurture factors conjugate with Aβ and behavioral outcomes from the preclinical to severe stage of AD (*Hwang et al., 2023*; *Petrella et al., 2019*; *Yada and Naoki, 2023*), while only a few studies explicitly assume an internal model of the patients (*Sutter et al., 2024*). Assuming an internal model would broaden and deepen interpretations of various cognitive deficits in terms of common computational account (*Kocagoncu et al., 2021*).

In line with this, our perspective is that states of an internal model would be one candidate of factors to generate a wide variety of behavioral phenotypes in AD, and hence probing the internal state would contribute not only to detecting early signs of AD but also to multidimensionally evaluating how the disease gets worse and how early intervention works. In this study, we assumed the latent cause model proposed by *Gershman et al., 2017b* as an internal model and unveiled the internal state in 6- and 12-month-old $App^{NL-G-F}$ mice, a preclinical mouse model of AD, from their behavioral data by estimating the parameters of the latent cause model well replicating the behaviors.

## The misclassification in the memory modification process underlies the impairment of $App^{NL-G-F}$ mice in the reinstatement of conditioned fear memory

Initially, we confirmed a significant accumulation of Aβ increased along aging in the $App^{NL-G-F}$ mice (*Figure 2—figure supplement 1*). We then tested whether the memory modification process was behaviorally impaired in the $App^{NL-G-F}$ mice across the auditory-cued fear conditioning, extinction, and reinstatement. First, the mice learned an association between a tone as a CS and an electrical shock as a US, as they exhibited increased CR in test 1 (*Figure 2B and C*). Next, they observed the CS alone repeatedly through the extinction, and then they exhibited decreased CR in test 2 (*Figure 2B and D*). After they observed the US alone, the mice exhibited increased CR again in test 3, suggesting that the reinstatement was established (*Figure 2B and E*). However, the $App^{NL-G-F}$ mice exhibited a lower extent of reinstatement compared to the control (*Figure 2H*). The DI between test 1 and 3 in the $App^{NL-G-F}$ mice was significantly lower than 0.5, while that in the control was comparable with 0.5 (*Figure 2H*), suggesting that the $App^{NL-G-F}$ mice would have lower expectations of upcoming shock in the reinstatement than immediately after the acquisition. These results were the first to report the deficits in the reinstatement in the $App^{NL-G-F}$ mice, suggesting that Aβ accumulation induces the defect in the memory modification process after the extinction.

Although our results support that the Aβ accumulation in $App^{NL-G-F}$ mice would have little effect on associative fear learning and extinction, the results have been mixed in studies using fear conditioning tasks. Two studies found normal contextual or auditory-cued fear conditioning responses up to 18 months old (*Kundu et al., 2021*; *Sakakibara et al., 2018*), which aligns with our results. On the contrary, another two studies reported that $App^{NL-G-F}$ mice have defects in both contextual and cued fear conditioning from the age of 5 months (*Emre et al., 2022*; *Mehla et al., 2019*). The inconsistency suggests the impact of Aβ may be complicated and cannot be simply detected by common behavioral tasks, highlighting the need for novel approaches to probe in-depth behavioral changes.

Recent advances in computational psychiatry offer a wealth of internal models to interpret a patient's internal states (*Hauser et al., 2022*; *Huys et al., 2016*; *Montague et al., 2012*). For instance, *Norbury et al., 2022* simulated a memory modification process in severe post-traumatic stress disorder (PTSD) patients by the latent cause model and found that they were biased to infer old memories rather than to acquire new memories for novel observations, as they have a lower concentration parameter $\alpha$. In this study, we found that $App^{NL-G-F}$ mice exhibited lower $\alpha$ both at 6 months and 12 months of age (*Figures 4A and 5A*). As the Aβ accumulation increased with age (*Figure 2—figure supplement 1*), lower $\sigma_x^2$ was observed in 12-month-old $App^{NL-G-F}$ mice (*Figure 4A*). As $\alpha$ and $\sigma_x^2$ significantly correlated with DI between test 3 and 1 (*Figures 4B and 5B*), we considered both as critical parameters that shaped the diverged internal states underlying reinstatement between control and $App^{NL-G-F}$ mice. Although we have considered alternative internal models, the latent cause model offered better explanatory power to our data (see Appendix 1), therefore, we stood on the latent cause model to explain reinstatement.

$App^{NL-G-F}$ mice with lower $\alpha$ favored to generalize the observation as if it arose from the old cause instead of differentiating it as a new memory. Indeed, their posterior probabilities of acquisition latent causes were higher than those of the control mice when a novel observation was given in test 1 (*Figure 4—figure supplement 3B*), contributing to the higher CR in $App^{NL-G-F}$ mice (*Figure 2C*, *Figure 4—figure supplement 3A*). Such overgeneralization associated with lower $\alpha$ may, in part, explain the deficits in discriminating highly similar visual stimuli reported in AD mouse model (*Ding et al., 2023*; *Saifullah et al., 2020*; *Zhu et al., 2017*), preclinical AD patients (*Leal et al., 2019*; *Lee et al., 2020b*), and MCI patients (*Ally et al., 2013*; *Belliart-Guérin and Planche, 2023*; *Laczó et al., 2021*; *Parizkova et al., 2020*; *Wesnes et al., 2014*; *Yassa et al., 2010*), as they classify the lure as previous learned object.

Following the Chinese restaurant process, the impact of $\alpha$ on internal states decreases with trials, and meanwhile, that of $\sigma_x^2$ relatively increases in our reinstatement paradigm. $App^{NL-G-F}$ mice in 12-month-old with lower $\sigma_x^2$ favored to differentiate similar observations into unique memories, especially in later phases, due to the decreased posterior probabilities of existing latent causes. Indeed, $App^{NL-G-F}$ mice had a larger number of latent causes acquired after the extinction with posterior probabilities (*Figure 4E*), resulting in a limited update of the associative weight of the existing latent causes by unsignaled shock (*Figure 4D*), and thereby lower CR in test 3 (*Figure 2E and H*). These results may explain the impaired ability to transfer previously learned association rules could be due to misclassified them as separated knowledge in preclinical autosomal dominant AD mutation carriers and mild AD patients (*Bódi et al., 2009*; *Petok et al., 2018*). Such overdifferentiation associated with lower $\sigma_x^2$ might underlie delusions, a common neuropsychiatric symptom reported in AD dementia (*Kumfor et al., 2022*), as an extended latent cause model could simulate the emergence of delusional thinking (*Erdmann and Mathys, 2022*). Thus, the deficit in reinstatement of conditioned fear memory in the 12-month-old $App^{NL-G-F}$ mice could be due to their overgeneralization or overdifferentiation of observations into memories, and consequently, the competing memories were not retained through the memory modification process (*Figures 4 and 5*).

Our study demonstrated that estimating internal states with the parameters of the latent cause model provided additional explanations for the cognitive differences between control and $App^{NL-G-F}$ mice. It is undoubted that conventional behavioral tests for AD patients or AD mouse models have provided evidence of cognitive decline and are useful for stratification in clinical trials. Nonetheless, it is also possible that certain deficits have antecedently emerged in internal states, even if behavioral deficits are not obvious as in 6-month-old $App^{NL-G-F}$ mice. The computational phenotype was proposed by *Montague et al., 2012* and defined as parameters of a computational model being in conjunction with measurable behavioral or biological phenotypes. In this sense, $\alpha$ and $\sigma_x^2$ in the latent cause model might satisfy this definition in this study. Computational phenotype was established to fill explanatory gaps between biological and psychological evidence in psychiatry, which would be informative likewise in AD studies.

## The misclassification prevents $App^{NL-G-F}$ mice from retaining competing memories

We further tested whether the deficit in the $App^{NL-G-F}$ mice observed in the reinstatement experiments can be replicated in the reversal Barnes maze paradigm, where the reversal training phases could resemble the extinction phase in the reinstatement paradigm, in that 'counterexample' of the previously established memory is observed. The $App^{NL-G-F}$ and control mice exhibited similar learning curves in the initial training phase regardless of age (*Figure 6—figure supplements 1A and 3A*). These data align with previous studies that did not find a significant spatial learning deficit in $App^{NL-G-F}$ mice up to 10 months old by using the Morris water maze, a common spatial learning task relevant to the Barnes maze (*Latif-Hernandez et al., 2019*; *Saifullah et al., 2020*; *Whyte et al., 2018*). Although $App^{NL-G-F}$ mice were previously reported to have longer latency and make more errors during training in the Barnes maze (*Broadbelt et al., 2022*; *Sakakibara et al., 2018*), these studies haven't described the mice's navigation strategies to explain such deficits.

During the reversal training, $App^{NL-G-F}$ mice displayed behavioral flexibility as control mice did based on the conventional analysis (*Figure 6—figure supplements 2B and 3B*), similar to findings in *Sakakibara et al., 2018*. However, our strategy analysis revealed that this comparable performance was achieved in a sub-optimal way. The usage of confirmatory strategy, such that mice selectively

explored the first and the second target hole was significantly higher in the control mice during the reversal training phase, especially in the 12-month-old (*Figure 6B and F*). After the reversal learning, the confirmatory strategy would become the optimal strategy at the first trial in each day to know today's goal location. In contrast, the $App^{NL-G-F}$ mice primarily took the perimeter search strategy such that mice explored the target hole in a 'brute-force' manner (*Figure 6B and F*), consistent with deficits in navigation ability found in preclinical AD patients (*Allison et al., 2016*; *Coughlan et al., 2018*). These results suggest that the control mice solved the task by inferring the two competing spatial memories representing the two possible target holes, whereas the $App^{NL-G-F}$ mice solved the task not relying on selective spatial memories but the perimeter strategy, while both mice were given the same spatial cues. Thus, our hypothesis that the $App^{NL-G-F}$ mice have a deficit in classifying observed cues into two or more competing memories in the same context would be supported by the results of the reversal Barnes maze experiment.

In the perspective from the latent cause framework, we posit that the mice would learn a larger number of association rules between stimuli in the maze compared to those in the reinstatement of the fear conditioning, and either infer new memories or modify existing memories for the unexpected observations in the Barnes maze (e.g. changed location or absence of escape box) as in the reinstatement paradigm. In the reversal Barnes maze paradigm, the animals would infer that a latent cause generates the stimuli in the maze at certain associative weights in each trial and would adjust behavior by retaining competing memories. Both overgeneralization and overdifferentiation could explain the lower exploration time of the target hole in the $App^{NL-G-F}$ mice. In the case of overgeneralization, the mice would overwrite the existing spatial memory of the target hole with a memory that the escape box is absent. In the case of overdifferentiation, the mice would infer a new memory such that the goal does not exist in the novel field, in addition to the old memory where the goal exists in the previous field. In both cases, the $App^{NL-G-F}$ mice would not infer that the location of the goal is fixed at a particular point and failed to retain competing spatial memories of the goal, leading to relying on a less precise, non-spatial strategy to solve the task.

As the latent cause model was built on the fear conditioning paradigm and does not explicitly assume the contribution of other cognitive functions such as attention or working memory, this study provides no direct evidence as to which model parameters are in conjunction with the behavioral outcomes in the reversal Barnes maze and which parameters involve other the cognitive functions that are presumably affected by AD pathology (*Finke et al., 2013*; *Kirova et al., 2015*; *Malhotra, 2019*). To evaluate them, further studies would be required to integrate computational models encompassing different cognitive functions, such as multiple successor representations (*Madarasz and Behrens, 2019*) or hidden state inference (*Sanders et al., 2020*) for spatial learning, upon more fundamental principles, such as the predictive coding in a hierarchical neural network (*Kocagoncu et al., 2021*). We believe that such models would not only explain a wide variety of symptoms in dementia and AD in a common framework but also predict disease progression or effects of interventions for various cognitive functions.

## Putative brain regions involving the altered internal states in the $App^{NL-G-F}$ mice

This study did not explicitly provide evidence about neural mechanisms underlying the defect in memory classification in $App^{NL-G-F}$ mice. Nonetheless, we speculated that possible neural circuits are presumably involved in the results reported here, according to the findings of the previous studies.

It is well known that the amygdala, the hippocampus, and the medial prefrontal cortex are critical in regulating fear memory and its extinction (*Maren et al., 2013*). In addition, the ventral tegmental area (VTA) plays a modulatory role with its wide projection to these brain regions (*Beier et al., 2015*; *Cai and Tong, 2022*). The dysfunction in the abovementioned brain regions may cause the defect of reinstatement. Indeed, previous studies have shown that the inhibition of dopaminergic signal from VTA to the infralimbic cortex (*Hitora-Imamura et al., 2015*) or inhibition of glutamatergic signal from VTA to the dorsal hippocampus (*Han et al., 2020*) during unsignaled shock reduced the reinstatement of fear memory in mice.

Although the neural implementation of the latent cause model has not been demonstrated yet, contributions of certain neuronal populations are expected. It is well established that prediction error in classical or operant conditioning learning is computed in the VTA (*Lerner et al., 2021*). The

prediction error is signaled from the VTA to the hippocampal CA1 via the dopaminergic projections, while the VTA receives a feedback signal from the CA1 (*Lisman and Grace, 2005*). The pattern separation and completion were implemented in the dentate gyrus (DG) and the CA3 in the hippocampus, respectively (*Neunuebel and Knierim, 2014*). As *Sanders et al., 2020* pointed out, one might find an analogy between the process of pattern completion/separation and the inference of old/new latent cause, in the sense that both consist of a generalization/differentiation process to assign a current observation to either the same class with the previous observations or a novel class. The dysfunction in this circuit has long been implicated in memory problems in aging and AD (*Palmer and Good, 2011*; *Wilson et al., 2006*). *Gershman et al., 2017b* predicted that activities of newborn neurons in the DG and neurons in the CA3 correlate with the prior probability over latent causes, while the dopaminergic projection from the VTA to the hippocampal CA1 represents the posterior probability over latent causes. In other words, prediction error signals from the VTA to the CA1 might serve to compute the likelihood of the observation given the latent causes. The feedback from the CA1 to the VTA might signal posterior probability over latent causes and then modulate the extent of associative weight update.

Different lines of evidence suggest that hippocampus functions were affected in $App^{NL-G-F}$ mice, including lower neuronal firing rates in the CA1 region (*Inayat et al., 2023*), degradation of gamma oscillation power in the CA3 region (*Arroyo-García et al., 2021*), and disrupted place cell remapping in CA1 region (*Jun et al., 2020*). While VTA dysfunction was not reported in $App^{NL-G-F}$ mice, studies using Tg2576 mice, which overexpress mutated APP, have shown that neurodegeneration in the VTA occurred at an early age and consequent dysfunction was correlated with impairments in the hippocampus (*Nobili et al., 2017*; *Spoleti et al., 2024*). Given the evidence above, dysfunctions caused by pathological Aβ in the hippocampus and VTA are potential candidates underlying unsuccessful latent cause inference in $App^{NL-G-F}$ mice, leading to a lower extent of reinstatement. Although this study considered limited disease factors, cognitive impairments in AD patients result from a combination of neuropathological factors and cannot be ascribed solely to Aβ deposition in clinical practice. Examining whether the interaction of neuropathology can be seen in the internal state with a mouse model, including tau pathology, e.g., $App^{NL-G-F}$/MAPT double knock-in mice (*Hashimoto et al., 2019*; *Saito et al., 2019*), would improve the translational interpretation in this study.

## Conclusion

Uncovering the underdeveloped symptoms in the AD continuum is critical to early detection and intervention by filling the explanatory gap between the pathological states and the cognitive decline in AD. We expect that seeking internal state as well as parameters in an internal model in conjunction with behavioral or biological phenotypes could provide in-depth explanations about behavioral outcomes in AD beyond assessing behavior alone. Our study is the first to report that the alteration in the internal state of $App^{NL-G-F}$ mice biased their memory modification process toward overgeneralization or overdifferentiation.

## Materials and methods

**Key resources table**

| Reagent type (species) or resource | Designation | Source or reference | Identifiers | Additional information |
|---|---|---|---|---|
| Strain, strain background (mouse) | C57BL/6-App<tm3(NL-G-F)Tcs> (KI/KI) | https://doi.org/10.1038/nn.3697 | RRID:IMSR_RBRC06344 | Denoted as $App^{NL-G-F}$ mice in this study |
| Strain, strain background (mouse) | C57BL/6J-Tg(Thy1-G-CaMP7,-DsRed2);C57BL/6-App<tm3(NL-G-F)Tcs>(KI/KI) | https://doi.org/10.1523/JNEUROSCI.0208-21.2021 | N/A | Denoted as $App^{NL-G-F}$ mice in this study |
| Strain, strain background (mouse) | C57BL/6JmsSlc | SLC Japan | RRID:IMSR_JAX:000664 | Denoted as control in this study |
| Strain, strain background (mouse) | C57BL/6J-Tg(Thy1-G-CaMP7,-DsRed2) | https://doi.org/10.1523/JNEUROSCI.0208-21.2021 | N/A | Denoted as control in this study |

*Continued on next page*

| Reagent type (species) or resource | Designation | Source or reference | Identifiers | Additional information |
|---|---|---|---|---|
| Software, algorithm | MATLAB R2022a, R2023a | MathWorks Inc, MA, US | RRID:SCR_001622 | Simulation and parameter estimation; Barnes maze results analysis; http://www.mathworks.com/ |
| Software, algorithm | LabVIEW 2013 | National Instruments, TX, US | RRID:SCR_014325 | Barnes maze experiment; http://www.ni.com/labview/ |
| Software, algorithm | QuickNII | https://doi.org/10.1371/journal.pone.0216796 | RRID:SCR_016854 | IHC image analysis; https://www.nitrc.org/projects/quicknii |
| Software, algorithm | VisuAlign | N/A | RRID:SCR_017978 | IHC image analysis; https://www.nitrc.org/projects/visualign |
| Software, algorithm | Ilastik | https://doi.org/10.1038/s41592-019-0582-9 | RRID:SCR_015246 | IHC image analysis; https://www.ilastik.org/ |
| Software, algorithm | Nutil v0.8.0 | https://doi.org/10.3389/fninf.2020.00037 | RRID:SCR_017183 | IHC image analysis; https://nutil.readthedocs.io/en/latest/ |
| Software, algorithm | JASP 0.18.3.0 | JASP team, 2024 | RRID:SCR_015823 | Statistical analysis; https://jasp-stats.org/ |
| Software, algorithm | Python 3.11.5 | N/A | RRID:SCR_008394 | Visualization; https://www.python.org/ |
| Antibody | anti-β-Amyloid (mouse monoclonal) | Biolegend | Cat # 803001, RRID:AB_2564653 | 1:1000 |
| Antibody | anti-GFAP (rat monoclonal) | Invitrogen | Cat # 13–0300, RRID:AB_2532994 | 1:1000 |
| Antibody | anti-Iba1 (rabbit polyclonal) | FUJIFILM Wako Pure Chemical Corporation | Cat # 019–19741, RRID:AB_839504 | 1:1000 |
| Antibody | Alexa Fluor 488 Donkey Anti-Rat IgG (H+L) Antibody | Molecular probes | A21208, RRID:AB_2535794 | 1:500 |
| Antibody | Alexa Fluor 568 Donkey Anti-Mouse IgG (H+L) Antibody | Molecular probes | A10037, RRID:AB_11180865 | 1:500 |
| Antibody | Alexa Fluor 647 Donkey Anti-Rabbit IgG (H+L) Antibody | Molecular probes | A31573, RRID:AB_2536183 | 1:500 |

## Animals

Male $App^{NL-G-F}$ homozygous knock-in mice (abbreviated as $App^{NL-G-F}$ mice in this study) and age-matched control mice were subjected to behavioral tests at 6 and 12 months old. Cohort and geno-type information was provided in *Supplementary file 1a*. We decided to assign at least 20 mice for each group (6-month-old $App^{NL-G-F}$ mice, 6-month-old control mice, 12-month-old $App^{NL-G-F}$ mice, and 12-month-old control mice) on average when designing the reinstatement experiment (see below). Animals were housed under a 12 hr light/dark cycle in a temperature-controlled environment (23–24°C temperature and 40–50% relative humidity) with food pellets (Japan SLC) and water provided *ad libitum*. Behavioral experiments were performed during the dark cycle (8 am–8 pm). After the behavioral experiments, mice were sacrificed by cervical dislocation unless their brains were sampled. All animal procedures were performed in accordance with the Kyoto University animal care committee's regulations (approval number: Lif-K24014).

## Behavioral test

### Reinstatement paradigm in fear conditioning

The reinstatement of conditioned fear response after a reminder of unconditioned stimulus is observed in different species (*Hermans et al., 2005*; *Hitora-Imamura et al., 2015*; *Monfils et al., 2009*). In the present study, the four-chamber fear conditioning testing system (O'HARA & CO., LTD, Tokyo, Japan) was used to deliver tone and electrical foot shock, and the top view camera in the chamber recorded

mouse behavior at 2 Hz, as previously described (*Tachiki et al., 2023*). The procedure contained the following phases (*Figure 2A*): acquisition (day 1), test 1 (day 2), extinction (day 3 and day 4), test 2 (day 5), unsignaled shock (day 8), and test 3 (day 9). The context (acrylic box) was the same throughout the test. Mice were transferred to the experimental room from the breeding room and then habituated for 30 min in their homecage in a rack until the phase started. In each phase, mice were moved between their homecage and the chamber via a transporting cage filled with woodchips. In the acquisition phase, a 60 s tone (65 dB, 10 kHz) as the conditioned stimulus (CS) was presented 300 s later, since the mice were released in the chamber. The last 2 s of the CS were accompanied by an electrical foot shock (0.3 mA) as the unconditioned stimulus (US). This CS-US association was repeated three times with a 90 s interval. After the last US, the mice were left in the chamber for 140 s and then returned to the homecage. In the test phases, a 60 s tone was given 120 s after the mice entry. In the extinction phase, only a 60 s tone was given 19 times on day 3 and 10 times on day 4 with a 200 s interval. In the unsignaled shock phase, an electrical foot shock (0.7 mA) was given 10 min after the mice were placed into the chamber. After the electrical shock, the mice were left in the chamber for 5 min and then transferred back to the homecage. The chamber and grid were cleaned with 70% propanol before trials started. The freezing behavior was measured as a conditioned response (CR), and the criterion for freezing is defined as the change of detected mouse area being less than 10 pixels within 1 second. The freezing rate was calculated as the percentage of time spent freezing 1 min before (preCS) and during the tone presentation (CS). Discrimination index (DI) has been widely used as a normalized measure to evaluate the extent to which the system can distinguish between two conditions. To evaluate the magnitude of how each mouse can discriminate between different phases, DI is calculated as $CR(t) / [CR(t)+CR(t')]$, where CR indicates freezing rate during CS presentation; $t$ and $t'$ indicate any two test phases.

## Barnes maze

Barnes maze is a commonly used behavioral task to examine spatial learning and reference memory in Alzheimer's disease model mice (*Webster et al., 2014*). The apparatus and setting were the same as previously reported (*Suzuki and Imayoshi, 2017*; *Tachiki et al., 2023*) (Bio-Medica, Osaka, Japan). The maze consisted of a circular open arena (98 cm in diameter) with 12 holes equally spaced around the perimeter. A black iron escape box was placed under one of the holes as a goal during the training phase. The location of the goal stayed constant for a given mouse but was randomized across mice. Four unique visual cues were located at the outside of the arena. Behavior during the trials was recorded using a GigE Vision camera (UI-5240SE-NIR; IDS Imaging Development Systems GmbH, Obersulm, Germany) mounted on the top of the maze to estimate the mouse's position in each frame. All programs used for data acquisition, processing, saving, and synchronized device controls were written in LabVIEW 2013 (National Instruments, TX, USA). The procedure contained the following phases (*Figure 6A*): habituation (day 0), initial training (day 1 to day 6), probe test 1 (day 7), retraining (day 8), reversal training 1 (day 9 to day 11), reversal training 2 (day 12 to day 14) and probe test 2 (day 15). Mice were transferred to the experimental room from the breeding room and then waited for 30 min in their homecage in a rack by the phase started. In the habituation phase, mice were released to the center of the field and could freely explore the field for 5 min. After the trial, the mice were moved into the escape box for another 5 min, then back to the homecage. In the training phase, the mice had to explore the field until they successfully got into the escape box within 10 min. Then, they were returned to the homecage and on standby for the next trial. In the probe test, the procedure was the same as the habituation phase, except that the mouse was returned to the homecage directly after the field exploration. The field was cleaned with 70% ethanol thoroughly, and the paper bedding in the escape box was changed per trial to reduce the effect of possible olfactory cues in the maze.

The analysis of the Barnes maze was performed on the custom code based on our previous study (*Suzuki and Imayoshi, 2017*) with some modifications in strategy analysis. The code for strategy analysis is freely available online (*Suzuki, 2024a*). In brief, the conventional analysis included the number of errors (count to visit holes other than the goal), latency, and travel distance from start to goal in each trial in the training phase, and time spent around each hole in the probe test. The strategy analysis was performed to qualitatively evaluate what kind of exploratory strategies were used by mice based on their belief in each trial (*Suzuki and Imayoshi, 2017*; *Tachiki et al., 2023*). For each mouse, the trajectory from start to goal in each trial was automatically assigned to any one of the exploration

strategies following the definition described in *Figure 6—figure supplement 2*. In this study, confirmatory and perimeter strategies were assumed in addition to the strategies previously reported. The former is the one such that the mice visit two holes, the current or past goal located opposite in the field, as if they have two memories, such that the goal exists in each of the two holes. The latter is the one such that the mice find the goal simply running along with the holes, as if they do not have memories that the goal exists in a specific hole. To evaluate how the belief in the mice changed after the reversal training, the discrimination index (DI) is calculated as $ER(t_2) / [ER(t_1)+ER(t_2)]$, where ER indicates the exploration rate; $t_1$ and $t_2$ indicate probe test 1 and probe test 2. The ER was calculated as time spent around the hole divided by the maximum exploration time across all holes in each mouse. The DI is calculated for the first target hole (hole 1) and the second target hole (hole 7).

## Simulating reinstatement and estimating parameters in the latent cause model

As the latent cause model serves as a generative model, it allows us to simulate the reinstatement experiment while exploring a set of model parameters, minimizing the prediction error between traces of simulated and observed CR for each mouse. The observed trace of CR across all trials in each mouse was initially normalized. In the simulation, the reinstatement experiment was replicated as follows. The number of trials was the same with our experiment: trials 1–3 was the acquisition phase, trial 4 was the test 1 phase, trials 5–23 was the first extinction phase, trials 24–33 was the second extinction phase, trial 34 was the test 2 phase, trial 35 was the unsignaled shock phase, and trial 36 was the test 3 phase. All trials contained stimuli, while the acquisition and the unsignaled shock trials contained the US. The stimulus given a trial was represented as a two-dimensional vector, where the first and the second dimensions indicated the CS and the context, respectively. If the CS was presented at the phases except for the unsignaled shock, the value got 1; otherwise, 0. Because the context was always presented, the value of context was given 0.2 for all trials. The US was represented as a binary value, either 1 at the acquisition and the unsignaled shock or 0 otherwise. The temporal distance of 24 hr between each phase was set to be 20, the same as in the original study (*Gershman et al., 2017b*).

Ten model parameters to be estimated were concentration parameter ($\alpha$), temporal scaling parameter ($g$), learning rate ($\eta$), maximum number of iterations in the EM algorithm (max. no. of iterations), initial associative weight ($w_0$), variance of the US ($\sigma_r^2$) and the stimulus ($\sigma_x^2$), response threshold ($\theta$), response gain ($\lambda$), and maximum number of latent causes ($K$), where $g$ is not explicitly described in the original model (*Gershman et al., 2017b*) while this is implemented in their source code (*Gershman, 2017a*). They were estimated through the slice sampling algorithm in the Markov chain Monte Carlo method (*Neal, 2003*). In implementation, we used the MATLAB function, *slicesample*. The initial guess and upper and lower bounds of each parameter were listed in *Appendix 2—table 1*. The initial size of the sampling window for each parameter was the difference between the upper and lower bounds of the parameter divided by 50. In each sampling step, a set of parameters was proposed, and then a CR trace was simulated in the virtual reinstatement experiment. For each parameter, one sample was drawn for every 100 steps in 200,000 times sampling, then the first 1000 samples were discarded as burn-in, and eventually 1000 samples were left. The convergence of the sample chain was determined if the Gelman-Rubin statistic was smaller than 1.1. The expected value of each parameter was considered to be the maximum bin in the sample histogram, for which the bin width was statistically optimized by the method proposed by *Shimazaki and Shinomoto, 2007*. As max. no. of iterations and $K$ are integers, each expected value was rounded toward infinity. Given the expected parameters, the likelihood of squared errors between the simulated and the observed CR trace was evaluated on a half-normal distribution ($\mu = 0$, $\sigma = 0.3$) so that increasing error decreased the likelihood. If the likelihoods of errors were smaller than the thresholds (1 for acquisition trials; 0.25 for tests 1, 2, and 3 trials; 0.5 for remaining trials) in all trials and the Gelman-Rubin statistic < 1.1, the sampling procedure above was terminated. Otherwise, it was continued while keeping the sample chain and setting the initial guess as the last value of the sampling chain. If the procedure was not terminated within 20 iterations, the observed CR trace was classified as an anomaly. The code described above is freely available online (*Suzuki, 2024b*).

Given the estimated parameters for each mouse, their internal state and CR were simulated for every trial. The internal states consisted of latent causes, their posterior probabilities, and associative

weight. The latent causes were categorized based on which phase they were initially inferred: acquisition trials, extinction trials (i.e., test 1, extinction, test 2), and trials after extinction (i.e., the unsignaled shock and test 3).

## Immunohistochemistry

After the designated behavioral test, mice were anesthetized by intraperitoneal injection with a cocktail of 0.3 mg/kg medetomidine (Nippon Zenyaku Kogyo, Koriyama, Japan), 4.0 mg/kg midazolam (Sandoz, Minato, Japan), and 5.0 mg/kg butorphanol (Meiji Seika Pharma, Chuo, Japan) and transcardially perfused phosphate buffer saline (PBS) followed by 4% paraformaldehyde (PFA). Brain samples were preserved in 4% PFA overnight, and the solution was replaced with 30% sucrose in PBS. Then, the sample was embedded in the OCT compound (Tissue TEK, Sakura Finetek Japan, Chuo, Japan) and sectioned at 30 µm (CM1950, Leica Biosystems, Tokyo, Japan). Every six sections were collected. Ten coronal brain sections from the anterior part (AP = +2.2 ~ +1 mm from Bregma) and the posterior part (AP = -1.5–-2.5 mm from Bregma) were subjected to staining in a 12-well plate. Sections were washed in PBS with 0.3% Triton X-100 (PBST) to remove excessive OCT compound, then blocked in 5% normal donkey serum (NDS) in PBST for an hour at room temperature and incubated with primary antibodies on a shaker at 4 °C overnight. The following day, sections were washed in PBST and reacted with second antibodies and DAPI for 2 hr at room temperature. Finally, the sections were washed in PBST, mounted on slides, and coverslipped with antifade reagent (FluoroMount-G, SoutherBiotech, Cat#0100–01). Primary antibodies, including mouse anti-β-Amyloid (1:1000, Biolegend, Cat#803001, RRID:AB_2564653), rabbit anti-Iba1 (1:1000, FUJIFILM Wako Pure Chemical Corporation, Cat#019–19741, RRID:AB_839504), and rat anti-GFAP (1:1000, Invitrogen, Cat#13–0300, RRID:AB_2532994), were prepared in PBST with 1% NDS. Second antibodies against mouse Alexa Fluor-568 nm (Invitrogen, Cat#A10037, RRID:AB_11180865), rabbit Alexa Fluor-647nm (Invitrogen, Cat#A31573, RRID:AB_2536183), and rat Alexa Fluor-488 nm (Invitrogen, Cat#A21208, RRID:AB_2535794) were used in a dilution of 1:500 in PBST with 1% NDS. Fluorescent immunohistochemistry slides were imaged on a fluorescence microscope (BZ-X800, Keyence, Japan) with a 10 x objective, and the stitching function was used to cover the entire section. Images were stitched in built-in analysis software (Keyence, Japan). The Aβ plaque area of each brain was quantified under QUINT workflow (*Yates et al., 2019*). First, a series of brain section images were registered to Allen Brain Atlas CCFv3 on QuickNII (*Puchades et al., 2019*) and fine-adjusted on VisuAlign (https://www.nitrc.org/projects/visualign). Next, the Aβ staining channel images were segmented in Ilastik (*Berg et al., 2019*). Finally, the integration of registration and segmented results was performed in Nutil (*Groeneboom et al., 2020*) to quantify Aβ plaque in selected brain regions. The Aβ plaque area (%) is calculated as the percentage of positive Aβ staining area covering the whole coronal brain section.

## Statistical analysis

For the fear conditioning, extinction, and reinstatement results, samples with freezing rates in test phases exceeding the 1.5 times interquartile range of the group were treated as outliers and excluded for further statistical analysis. The one-way ANCOVA was performed to analyze genotype effects on the freezing rate and DI, with age as a covariate. The Student's *t*-test was performed to compare the freezing rate and the DI between control and $App^{NL-G-F}$ mice within the same age. To test whether the DI differs from 0.5, the one-sample *t*-test was performed within the group. To compare the estimated parameters and properties of latent causes between control and $App^{NL-G-F}$ mice within the same age, the Mann-Whitney $U$ test was performed, as the Shapiro-Wilk normality test reported a violation of normality assumption in the data. The Spearman's rank correlation coefficient between the estimated model parameters and the DIs between test 3 and test 1 phase were calculated.

For the reversal Barnes maze results, a mixed-designed two-way [Genotype (control, $App^{NL-G-F}$) × Day (1–6, 9–11, or 12–14)] ANOVA was performed for the number of errors, the latency, and the travel distance in the initial and reversal training. A mixed-designed two-way [Genotype (control, $App^{NL-G-F}$) × Hole (1-12)] ANOVA was performed for the time spent around each hole in the probe test 1 and 2. If a significant interaction between genotype and hole was detected, Tukey's HSD test at a specific hole between control and $App^{NL-G-F}$ was performed. The Wilcoxon rank sum test was performed to compare the use of each strategy on each training day between control and $App^{NL-G-F}$ within the same age. To test whether the DI between probe tests 1 and 2 differs from 0.5, the Wilcoxon signed-rank test was

performed within the group, as the Shapiro-Wilk normality test reported a violation of the normality assumption in the data. The Mann-Whitney *U* test was performed to compare the DI between control and *App*$^{NL-G-F}$ mice within the same age.

All statistical analyses were done in JASP 0.18.3.0 (JASP team, 2024) and MATLAB R2022b (Math-Works Inc, MA, USA). The significance level for all statistical tests was set at 0.05. The power analysis table was provided in *Supplementary file 1u and v* by using G*Power 3.1 (*Faul et al., 2007*) Not all of the multiple comparisons were corrected for, and the sample sizes written in the figure legends do not include outliers detected through the exclusion procedure mentioned above.

## Acknowledgements

We thank Adam T Guy, PhD and Shigeru Shinomoto, PhD for informative comments on the manuscript. We also thank all the members of the Imayoshi lab for their support. We acknowledge ChatGPT-4 and Google Gemini for their contributions, which were selectively referenced during the Python code refactoring process for visualization and drawing mice artwork in *Figure 1*. This work was supported by the Japan Society for the Promotion of Science (JSPS), Grant-in-Aid for Early-Career Scientists JSPS 19K16292 (to YS), Grant-in-Aid for Scientific Research (B) JSPS 21H02485 (to II) from the Ministry of Education, Culture, Sports, Science and Technology of Japan (MEXT); by Japan Science and Technology Agency (JST) CREST program Grants JPMJCR1921 (to II) and JPMJCR1752 (to II); and by the Program for Technological Innovation of Regenerative Medicine. Grant 21bm0704060h0001 and 24bm1123049h0001 (to II) and Brain/MINDS Grant 21dm0207090h0003 (to II) and Moonshot Research & Development Program JP22zf0127007 (to II) from the Japanese Agency for Medical research and Development (AMED).

## Additional information

### Funding

| Funder | Grant reference number | Author |
| --- | --- | --- |
| Japan Society for the Promotion of Science | 19K16292 | Yusuke Suzuki |
| Japan Society for the Promotion of Science | 21H02485 | Itaru Imayoshi |
| Japan Science and Technology Agency | 10.52926/jpmjcr1752 | Itaru Imayoshi |
| Japan Agency for Medical Research and Development | 21bm0704060h0001 | Itaru Imayoshi |
| Japan Agency for Medical Research and Development | 24bm1123049h0001 | Itaru Imayoshi |
| Japan Science and Technology Agency | 10.52926/jpmjcr1921 | Itaru Imayoshi |
| Japan Agency for Medical Research and Development | 21dm0207090h0003 | Itaru Imayoshi |
| Japan Agency for Medical Research and Development | JP22zf0127007 | Itaru Imayoshi |

The funders had no role in study design, data collection and interpretation, or the decision to submit the work for publication.

### Author contributions

Mei-Lun Huang, Conceptualization, Formal analysis, Investigation, Visualization, Methodology, Writing - original draft; Yusuke Suzuki, Conceptualization, Software, Formal analysis, Supervision,

Methodology, Writing – review and editing; Hiroki Sasaguri, Takashi Saito, Resources, Writing – review and editing; Takaomi C Saido, Resources; Itaru Imayoshi, Resources, Supervision, Funding acquisition, Writing – review and editing

### Author ORCIDs
Mei-Lun Huang (iD) https://orcid.org/0009-0000-3689-9749
Yusuke Suzuki (iD) https://orcid.org/0000-0002-5379-4030
Itaru Imayoshi (iD) https://orcid.org/0000-0001-9728-481X

### Ethics
All animal procedures were performed in accordance with the Kyoto University animal care committee's regulations (approval number: Lif-K24014).

Reviewer #1 (Public review): https://doi.org/10.7554/eLife.105347.3.sa1
Reviewer #2 (Public review): https://doi.org/10.7554/eLife.105347.3.sa2
Reviewer #3 (Public review): https://doi.org/10.7554/eLife.105347.3.sa3
Author response https://doi.org/10.7554/eLife.105347.3.sa4

## Additional files

### Supplementary files
Supplementary file 1. Tables corresponding to the statistical results.
MDAR checklist

### Data availability
All data generated or analysed during this study are included in the manuscript and supporting files; source data files have been provided for all main figures.

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

# Appendix 1

## Alternative models

In order to test whether the latent cause model (LCM) has better explanatory power in this study, we compared its fit to behavioral data with that of two alternative models: the variant of the Rescorla-Wagner (RW; *Rescorla and Wagner, 1972*) model and the latent state model (LSM; *Cochran and Cisler, 2019a*). The RW model serves as a baseline, given its known limitations in explaining fear return after extinction, while the LSM is another contemporary model capable of modeling memory modification.

## Rescorla-Wagner (RW) model

The RW model, proposed by *Rescorla and Wagner, 1972*, is an elemental theory in which an association between a compound of stimuli and an outcome consists of associations between individual stimulus and the outcome (*Soto et al., 2014*). In the present reinstatement experiment, the gross associative strength $V$ in each trial given the CS and the context is the sum of $V_{CS}$ and $V_{context}$:

$$V = V_{CS} + V_{context},$$

where the former is the associative strength between the CS and the US, while the latter is that between the context and the US. According to the original RW model, $V_{CS}$ and $V_{context}$ are updated as

$$\Delta V_{CS} = \alpha_{CS}\beta(\lambda - V),$$

$$\Delta V_{context} = \alpha_{context}\beta(\lambda - V),$$

where $\alpha$ and $\beta$ are the saliency of each stimulus and the learning rate for the US, respectively. $\lambda$ is the observed US that is the asymptote of $V$, and $(\lambda - V)$ expresses a prediction error between the observed US and $V$ as a predicted US given the stimuli. Consequently, $\Delta V_{CS}$ and $\Delta V_{context}$ would gradually increase during acquisition and decrease during extinction. However, in our reinstatement paradigm, this update rule causes an increase in $\Delta V_{CS}$ in the unsignaled shock trial, even though CS is not presented. In contrast, $\Delta V_{context}$ is underestimated due to the $V_{CS}$ in the prediction error term. To prevent this, we modified each update rule as

$$\Delta V_{CS} = \alpha_{CS}\beta(\lambda - V_{CS})x_{CS},$$

$$\Delta V_{context} = \alpha_{context}\beta(\lambda - V_{context})x_{context},$$

where $x_{CS}$ and $x_{context}$ were stimulus inputs while $\lambda$ was the US input, and these values were the same as those in the simulation in the latent cause model (see Materials and methods). Thus, $V_{CS}$ and $V_{context}$ are updated only in the presence of the CS and the context, respectively. Finally, we estimated CR as a linear combination of stimulus $\mathbf{x}$ and associative strength $\mathbf{V}$,

$$CR = x_{CS}V_{CS} + x_{context}V_{context}.$$

## Latent state model (LSM)

The LSM, proposed by *Cochran and Cisler, 2019a*, shares several concepts and processes with the latent cause model (*Gershman et al., 2017b*). In the LSM, an agent has an internal model such that $R(t)$, the outcome (e.g., US) at trial $t$, is generated according to an $n$-dimensional cue vector $\mathbf{c}(t)$ (e.g., CS), $l$-th latent state in a total of $L$ latent states acquired so far, and a latent noise. The series of latent states across trials is a Markov chain, as each latent state changes at probability $\gamma(L - 1) / L$, otherwise stays, where $\gamma$ is a parameter that governs the transition probability of latent states. $R(t)$ given $l$-th latent state is approximated by a linear combination of $\mathbf{c}$ and an associative strength vector $\mathbf{v}$ as the RW model, resulting in the prediction error $E_l(t)=R(t) - \mathbf{c}(t)'\mathbf{v}_l(t)$. $\mathbf{v}$ is iteratively updated by the 'rumination' process at the end of this trial (see below). As in the latent cause model, the purpose of its computation is to minimize the prediction error.

At the beginning of each trial, the agent observes $\mathbf{c}(t)$ and $R(t)$, then calculates the posterior probability of each latent state $p_l(t)$ by Bayes' rule,

$$p_l(t) = \frac{1}{l_0}\left((1-\gamma)p_l(t-1) + \frac{\gamma}{L}\right)\phi\left(\frac{E_l(t)}{\sigma(t)}\right),$$

where $\varphi(\cdot)$ is the likelihood of the prediction error given latent state $l$, evaluated using a standard normal distribution. $\sigma(t)$ represents the uncertainty of $R(t)$, and is updated over trials based on the prediction errors (see below). The term before the likelihood is the prior of latent state $l$, and $l_0$ is the partition function, the sum over $p_l(t)$ across latent states 1 to $L$. Next, the agent determines whether a new latent state should be inferred using the change point statistic $q(t)$,

$$q(t+1) = \max\left(q(t) + \log\left(\frac{\phi(0)}{l_0}\right) - \delta, 0\right),$$

where $\varphi(0)$ represents the ideal state with no prediction error, hence $\log(\varphi(0)/l_0)$ can be interpreted as the log likelihood ratio between this ideal state and the gross likelihood provided by $L$ latent states. $\delta$ is the penalty term. Intuitively, $q(t)$ indicates how the set of latent states acquired so far is close to the ideal state. If $q(t)$ exceeds the threshold $\eta$, $L+1$ th new state is inferred while $L < L_{max}$, where $L_{max}$ is the predefined maximum number of latent states that can be acquired.

Given $\mathbf{c}(t)$ and $E_l(t)$, associative strength $\mathbf{v}$ of latent state $l$ is updated a

$$\mathbf{v}_l(t+1) = \mathbf{v}_l(t) + \mathbf{A}_l(t)\,\mathbf{c}(t)\,E_l(t),$$

where $\mathbf{A}_l(t)$ is the associability matrix for latent state $l$. $\mathbf{A}_l(t)$ can be viewed as a learning rate as it governs how quickly associative strength changes. It depends on the probability of latent state $l$ and the history of cue observations, according to the effort matrix $\mathbf{B}_l(t)$ under latent state $l$, that is,

$$\mathbf{A}_l(t) = \alpha_0 p_l(t)\,\mathbf{B}_l(t)^{-1},$$

and $\mathbf{B}_l(t)$ is updated by

$$\mathbf{B}_l(t+1) = \mathbf{B}_l(t) + \alpha_2\left(p_l(t)\,\mathbf{c}(t)\,\mathbf{c}(t)' - \mathbf{B}_l(t)\right),$$

where $\alpha_0$ and $\alpha_2$ are the learning rate and ranging between 0 and 1. Then the agent updates uncertainty $\sigma(t)$ of $R(t)$ by

$$\sigma^2(t+1) = \sigma^2(t) + \alpha_1(\mathbb{E}(E^2(t)) - \sigma^2(t)),$$

where $\alpha_1$ is the learning rate for the uncertainty ranging between 0 and 1. $\mathbb{E}(E^2(t))$ is the squared prediction error averaged over latent states. The initial value of $\sigma^2$, denoted as $\sigma_0^2$, is predefined. By the beginning of next trial $t+1$, $p_l(t)$ is updated by

$$p_l(t)^* = (1-\gamma)^{ITI-1}p_l(t) + \left(1 - (1-\gamma)^{ITI-1}\right)\frac{1}{L},$$

expressing the temporal decay over inter-trial interval (*ITI*) and then used as the prior in the next trial. In addition, the 'rumination' process runs given the ITI: the agent iteratively updates $\mathbf{v}_l$ using the same observation of $\mathbf{c}(t)$ and $R(t)$ from the previous trial. The number of this iteration is determined by $\min(\chi, ITI-1)$, where $\chi$ is predefined.

## Evaluation of model fit

### Settings

To compare the model fit among the RW model, the LSM, and the LCM, we initially estimated the parameters of models minimizing the squared prediction errors between the observed CR and the simulated CR given the proposed parameters. To apply the same evaluation criteria for prediction error set in the LCM, we normalized both the observed CR and the simulated CR in the RW model and the LSM.

For the RW model, we estimated $\alpha_{CS}$, $\alpha_{context}$, and $\beta$ using the least squares error method in the grid search. We assumed that the value of each parameter could take any one of a uniform grid of 50 points between 0.1 and 1, therefore there were $50^3$ possible combinations of the three parameters. For each observed CR, we selected the combination of parameters with the lowest sum of squared

prediction errors across all trials. The criteria for anomaly detection were the same as those for the LCM, except that the Gelman-Rubin statistic was not applied (see Materials and methods).

For the LSM, we estimated $\alpha_0$, $\alpha_1$, $\alpha_2$, $\gamma$, $L_{max}$, $\eta$, $\chi$, and $\sigma_0$, but fixed two parameters, $\tau$ (exploration-exploitation parameter) and $\delta$ (constant involved in change point statistic), assuming that their effects would be minimal. $\tau$ and $\delta$ were set to the default value in *Cochran and Cisler, 2019a*. We simulated the reinstatement paradigm given the proposed parameters, using the codes provided by *Cochran and Cisler, 2019a* (https://github.com/cochran4/OnlineLatentStateLearning, *Cochran and Cisler, 2019b*). Note that the variable names above follow those in the code, except $L_{max}$; $L_{max}$ is noted *ncop* in the code; $\alpha_1$, $\alpha_2$, and $\eta$ in the code correspond to $\alpha_0$, $\beta_0$, and $v$ in Table 3 of *Cochran and Cisler, 2019a*. The values of the CS, context, US, *ITI*, and other experimental designs were the same as those set in parameters estimation in the LCM (see Material and Method), whereas the initial bias for the US centering was set to 0. We treated the expected outcome given cues, $\mathbf{c}(t)'\mathbf{v}(t)$, as the CR at each trial. As in the LCM, the proposed parameters were sampled while they reduced the prediction errors, through the MCMC method with the slice sampling (see Materials and methods). The step size in *slicesample* function was changed from 100 to 50 to reduce the sampling time. As only three samples in the 12-month-old group were marked as anomalies due to the failure of the convergence, not prediction error, the quality of sampled parameters should be comparable to those of the LCM, in terms of convergence. The initial guess, upper bound, and lower bound of each parameter are shown in *Appendix 1—table 1*. The criteria for anomaly detection were the same as those for the LCM (see Materials and methods).

## Simulation

To illustrate the representative fitting characteristics of each model, we initially estimated parameters from the median CR curves in each group. The RW model given the estimated parameters successfully replicated the gradual increase of CRs in the acquisition and their decrease in the extinction (*Appendix 1—figure 1A–D*). As expected, the RW model failed to replicate the reinstatement, as a single trial of context-US pairing, the unsignaled shock, only resulted in a small increase in the overall associative strength after sufficient extinction. Indeed, only the simulated CR of 12-month-old $App^{NL-G-F}$ mice groups passed the anomaly detection test. In the LSM, the simulated CR well replicated the gradual increase of CR in the acquisition in the 6-month-old group than in the 12-month group, while the gradual decrease of CR was better reproduced in the 12-month group than in the 6-month-old group (*Appendix 1—figure 1E–H*). The increased CR in the first trial in the second extinction phase (trial 24) was captured only in the 12-month-old control (*Appendix 1—figure 1G*). The magnitude of reinstatement was well replicated in all groups other than the 6-month-old control (*Appendix 1—figure 1E–H*).

The number of samples classified as anomalies in each group was highest in the RW model across all groups, and that in the LSM was twice that of in the LCM, except 6-month-old $App^{NL-G-F}$ mice, suggesting that the LCM was robust for individual differences (*Appendix 1—figure 2*). The prediction errors of all accepted samples in the RW model were obviously higher in the reinstatement at trial 36 (*Appendix 1—figure 3A*), which is aligned with the median data (*Appendix 1—figure 1A–D*). In contrast, the prediction errors of accepted samples in the LSM at trial 36 were smaller (*Appendix 1—figure 3B*), suggesting that it could replicate the reinstatement, as shown in the median data (*Appendix 1—figure 1E–H*). In both models, the prediction errors during the extinction phase were lower than in other phases (*Appendix 1—figure 3A and B*). We then statistically compared the sum of prediction errors over trials, as well as those in acquisition trials and test phase trials, between the LCM and LSM, excluding the RW model due to the lower number of accepted samples (*Appendix 1—figure 2*). The sum of prediction error over trials was significantly higher in the LCM, especially in the 12-month-old group (*Appendix 1—figure 4A and B*, left panel). This would be due to the significantly larger prediction errors during the acquisition phase in the LCM (*Appendix 1—figure 4A and B*, center panel). Notably, the prediction errors in the 3 test phases were almost comparable (*Appendix 1—figure 4A and B*, right panel).

In this study, we favored models that robustly replicate CR during the 3 test phases, rather than during the acquisition phase, from as many samples as possible. In this sense, the LSM remains a potential candidate as it successfully replicated the reinstatement in a different manner from that in the LCM. There were only one or two latent states inferred during the acquisition (*Appendix 1—figure 5A and D*, left panel). Almost all accepted samples continued to infer the acquisition latent

states during extinction, without inferring new states (*Appendix 1—figure 5B and E*, left panel). Although a new state was inferred after extinction in some of the samples, its posterior probability was low in test 3 (*Appendix 1—figure 5C and F*). On the contrary, the sum of posterior probabilities of acquisition latent states was highest in test 3 (*Appendix 1—figure 5A and D*), suggesting that the reinstatement is explained by the return of the acquisition state. However, this pattern was also observed in the 12-month-old *App*$^{NL-G-F}$ group. When we calculated the DIs between tests from simulated CR in the LSM (*Appendix 1—figure 6*), the significantly lower DI between test 3 and test 1 in the 12-month-old *App*$^{NL-G-F}$ group (*Figure 2H*) was indeed not replicated (*Appendix 1—figure 6C*), indicating accepted samples in the LSM failed to capture our behavioral features of interest.

Thus, we conclude that standing on the LCM to explain the memory modification observed in this study would be reasonable, given its robustness for individual differences and the reproducibility of the empirical data of interest. On the other hand, we did not test all possible combinations of parameters of the LSM or all possible experimental protocols to induce reinstatement. Since the conditions for simulating reinstatement and estimating parameters were initially developed in the LCM and applied *post hoc* to the LSM, we did not rule out the possibility that the LSM can demonstrate better explanatory power than the LCM in alternative conditions. A comprehensive search of such conditions may be a valuable direction for future research.

**Appendix 1—table 1.** The initial value, lower bound, and upper bound of parameters in slice sampling in the latent state model.

| Description | Parameters | Initial value | Lower bound | Upper bound |
| --- | --- | --- | --- | --- |
| learning rate for associative strength | $\alpha_0$ | 0.05 | 0.005 | 0.1 |
| learning rate for variance | $\alpha_1$ | 0.06 | 0.005 | 0.5 |
| learning rate for covariance | $\alpha_2$ | 0.04 | 0.005 | 0.5 |
| transition probability between states | $\gamma$ | 0.1 | 0.01 | 1 |
| maximum number of latent states | *ncop* | 15 | 1 | 36 |
| threshold to activate new state | $\eta$ | 2.5 | 0.01 | 3.05 |
| number of rumination updates | $\chi$ | 5 | 1 | 10 |
| initial standard deviation | $\sigma_0$ | 0.5 | 0.1 | 1 |

Note. $\tau$ and $\delta$ were fixed at 10 and 0.6, respectively, in the present simulation.

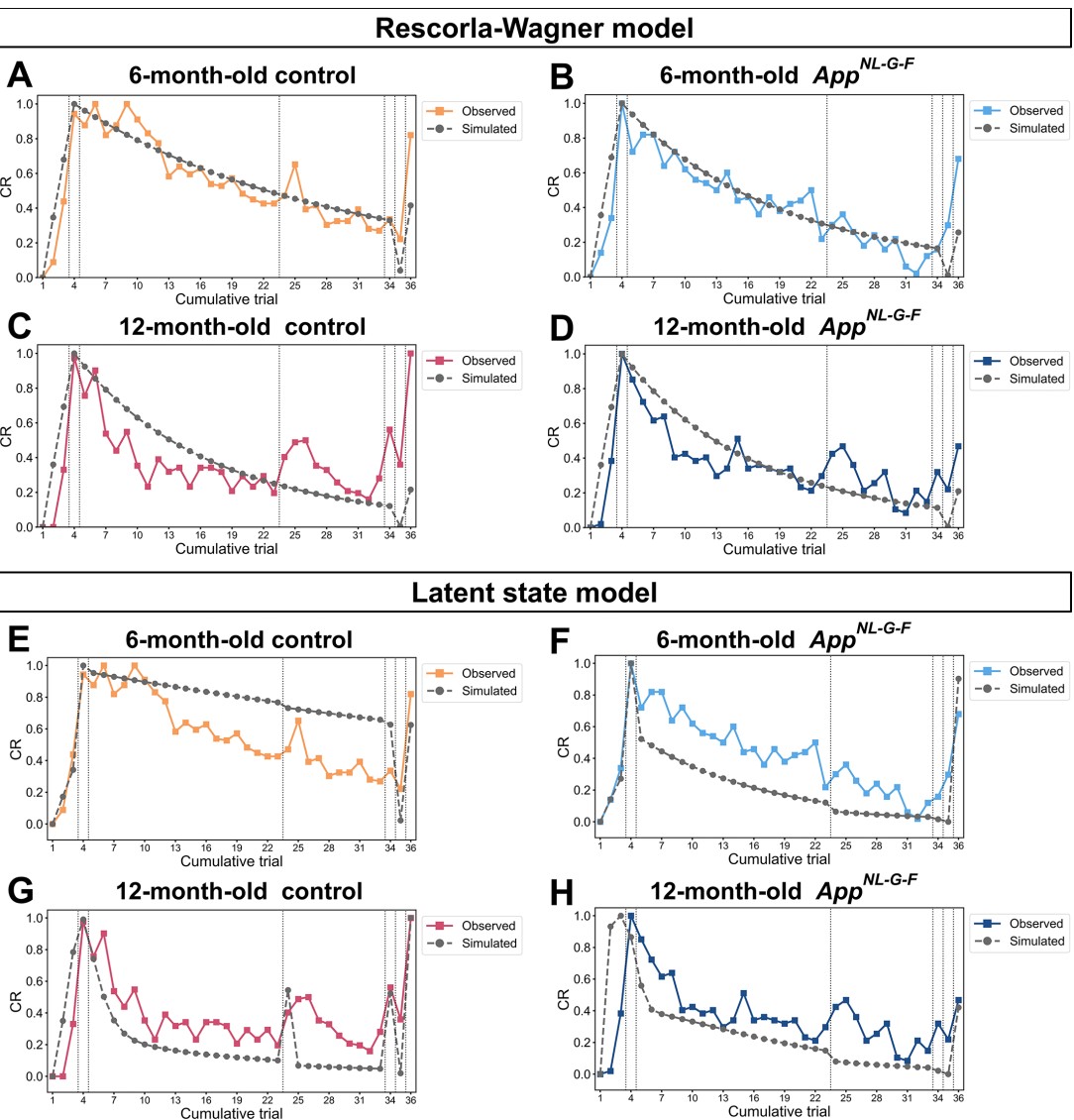

**Appendix 1—figure 1.** Model fit of the Rescorla-Wagner model and the latent state model to the behavioral data. (**A**) Simulation of reinstatement in the 6-month-old control mice in the RW model. The estimated parameter value: $\alpha_{CS}$ = 0.1, $\alpha_{context}$ = 1, $\beta$ = 0.504. (**B**) Simulation of reinstatement in the 6-month-old $App^{NL-G-F}$ mice in the RW model. The estimated parameter value: $\alpha_{CS}$ = 0.1, $\alpha_{context}$ = 1, $\beta$ = 0.302. (**C**) Simulation of reinstatement in the 12-month-old control mice in the RW model. The estimated parameter value: $\alpha_{CS}$ = 0.1, $\alpha_{context}$ = 1, $\beta$ = 0.614. (**D**) Simulation of reinstatement in the 12-month-old $App^{NL-G-F}$ mice in the RW model. The estimated parameter value: $\alpha_{CS}$ = 0.1, $\alpha_{context}$ = 1, $\beta$ = 0.596. (**E**) Simulation of reinstatement in the 6-month-old contol mice in the latent state model (LSM). The estimated (parameter value: $\alpha_0$ = 0.012, $\alpha_1$ = 0.020, $\alpha_2$ = 0.040, $\gamma$ = 0.020, ncop = 10, $\eta$ = 2.369, $\chi$ = 3, $\sigma_0$ = 0.99). (**F**) Simulation of reinstatement in the 6-month-old $App^{NL-G-F}$ mice in the LSM. The estimated parameter value: $\alpha_0$ = 0.078, $\alpha_1$ = 0.058, $\alpha_2$ = 0.418, $\gamma$ = 0.258, ncop = 10, $\eta$ = 2.290, $\chi$ = 7, $\sigma_0$ = 0.69. (**G**) Simulation of reinstatement in the 12-month-old control mice in the LSM. The estimated parameter value: $\alpha_0$ = 0.043, $\alpha_1$ = 0.191, $\alpha_2$ = 0.376, $\gamma$ = 0.099, ncop = 10, $\eta$ = 0.044, $\chi$ = 8, $\sigma_0$ = 0.58. (**H**) Simulation of reinstatement in the 12-month-old $App^{NL-G-F}$ mice in the LSM. The estimated parameter value: $\alpha_0$ = 0.069, $\alpha_1$ = 0.020, $\alpha_2$ = 0.495, $\gamma$ = 0.886, ncop = 32, $\eta$ = 0.040, $\chi$ = 9, $\sigma_0$ = 0.62. (**A–H**) The observed conditioned response (CR) is the median freezing rate during the conditioned stimulus (CS) presentation over the mice within each group; both observed and simulated CR of each group were divided by their maximum over all trials. The vertical dashed lines indicate the boundaries of phases. In the RW model, 6-month-old control (**A**), 6-month-old $App^{NL-G-F}$ (**B**), and 12-month-old control (**C**) were marked as anomalies in parameter estimation.

The online version of this article includes the following source data for appendix 1—figure 1:

**Appendix 1—figure 1—source data 1.** Data shown in *Appendix 1—figure 1*.

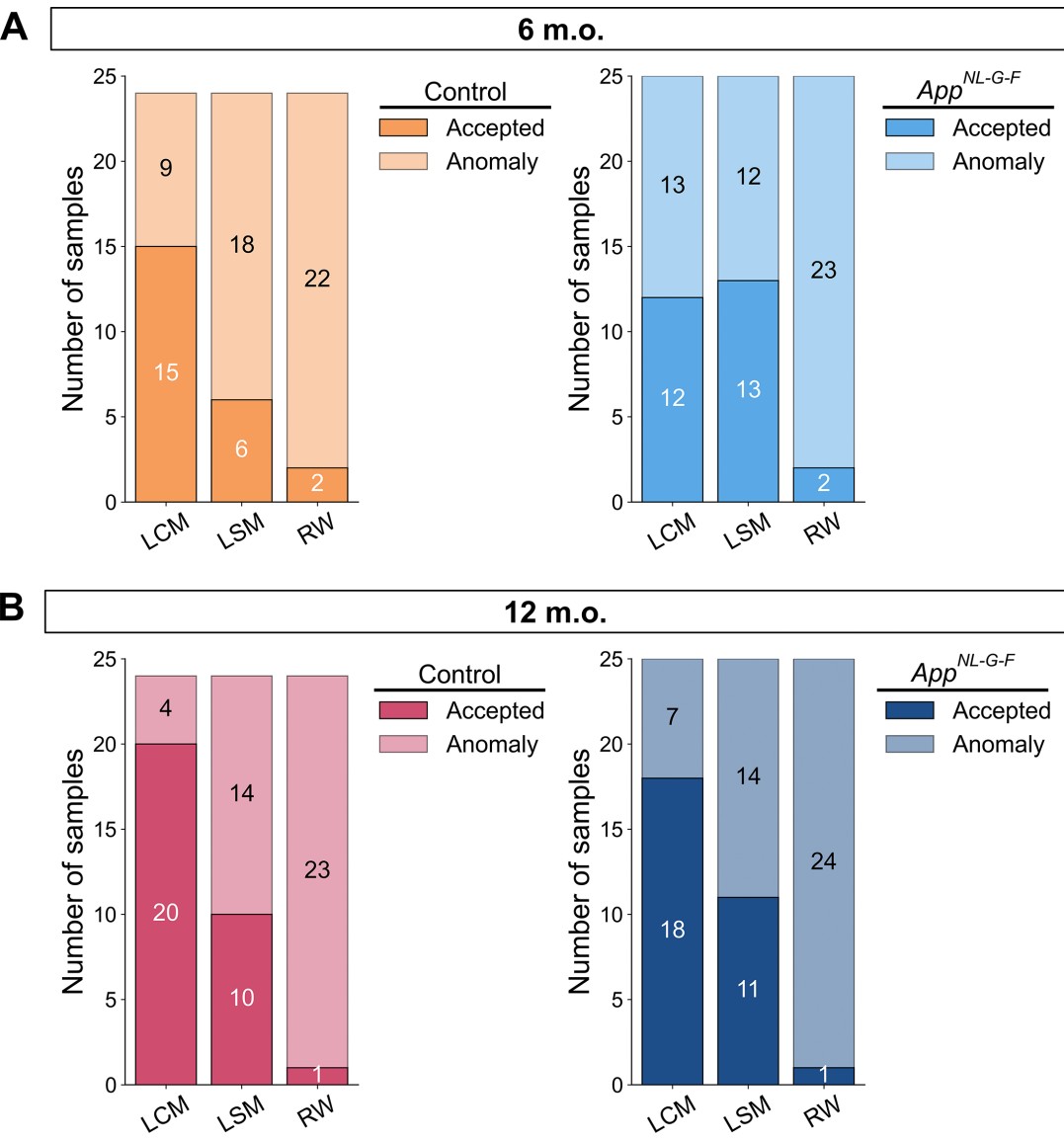

**Appendix 1—figure 2.** Number of excluded samples during parameter estimation in each model. The number of samples that did not pass the accepted criteria in the 6-month-old group (**A**) and 12-month-old group (**B**). LCM: latent cause model; LSM: latent state model; RW: Rescorla-Wagner model. Colors indicate different groups: orange represents 6-month-old control ($n = 24$), light blue represents 6-month-old $App^{NL-G-F}$ mice ($n = 25$), pink represents 12-month-old control ($n = 24$), and dark blue represents 12-month-old $App^{NL-G-F}$ mice ($n = 25$). The transparency of the color indicates accepted samples (dark) and anomalies (light).

The online version of this article includes the following source data for appendix 1—figure 2:

**Appendix 1—figure 2—source data 1.** Data shown in *Appendix 1—figure 2*.

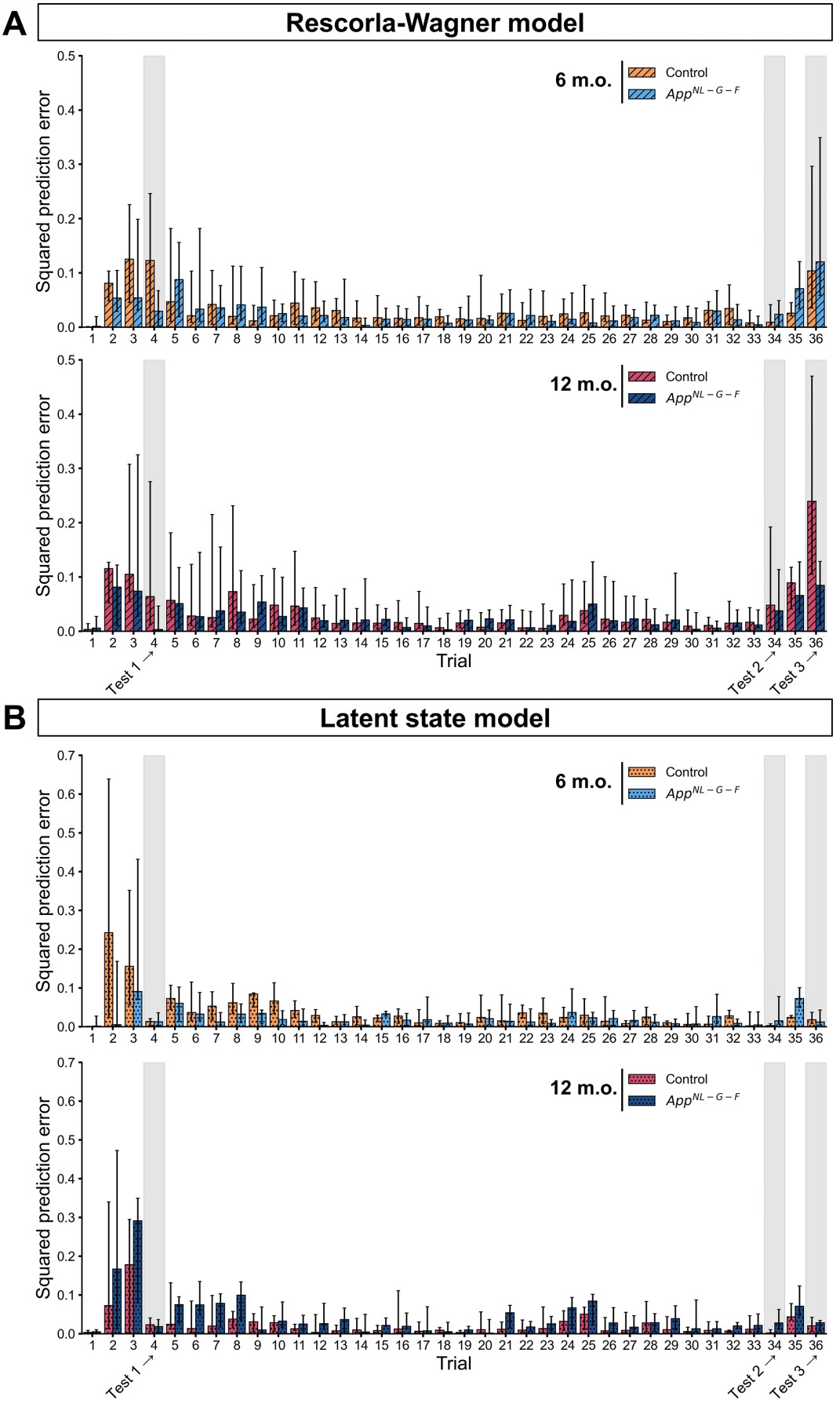

**Appendix 1—figure 3.** The squared prediction error in the Rescorla-Wagner model (**A**) and the latent state model (**B**). The squared prediction error across trials is shown as median with interquartile range. In the RW model, data from all samples are used. In the LSM, only accepted samples are shown (the exact numbers are shown in *Appendix 1—figure 2*).

The online version of this article includes the following source data for appendix 1—figure 3:

**Appendix 1—figure 3—source data 1.** Data shown in *Appendix 1—figure 3*.

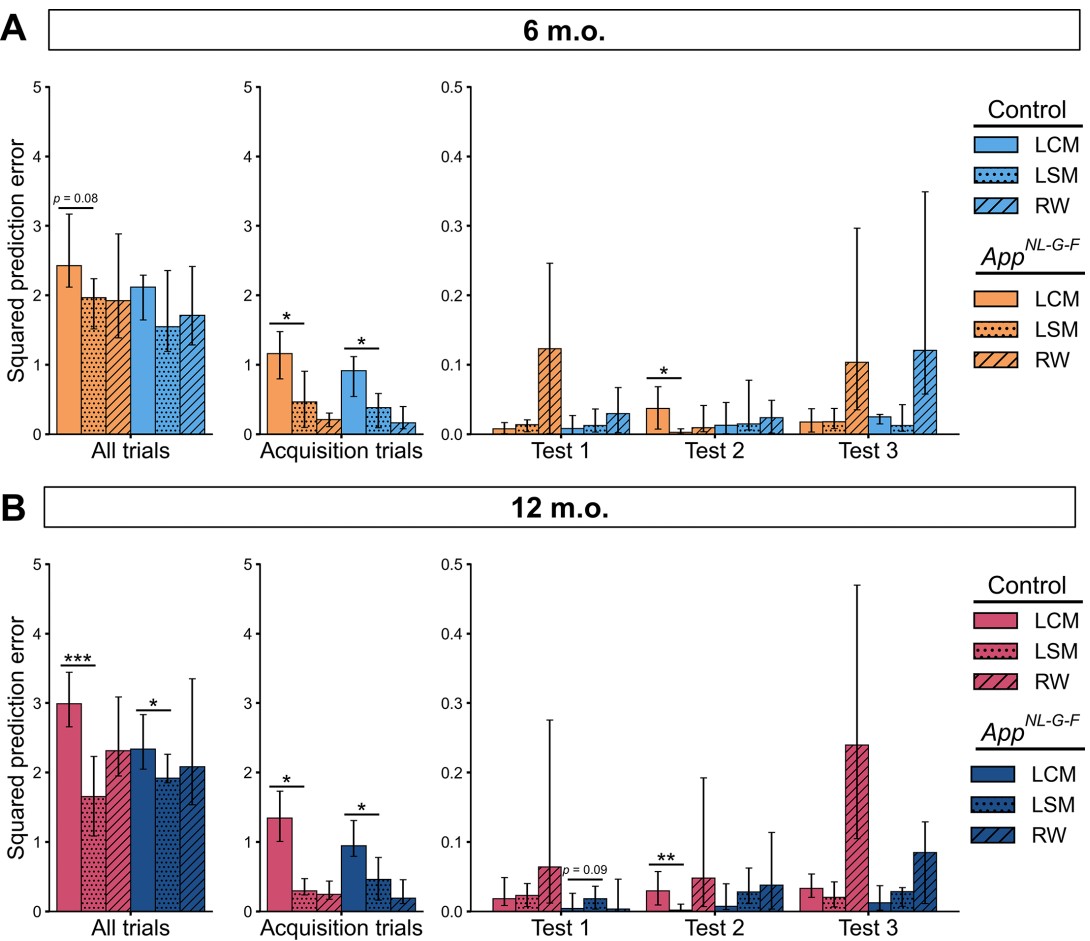

**Appendix 1—figure 4.** Comparison of squared prediction error among models in the 6-month-old group (**A**) and 12-month-old group (**B**). The data are shown as median with interquartile range. In the RW model, data from all samples are used. In the LSM and LCM, only accepted samples are shown (the exact numbers are shown in *Appendix 1—figure 2*). LCM: latent cause model; LSM: latent state model; RW: Rescorla-Wagner model. *$p <$ 0.05, **$p <$ 0.01, and ***$p <$ 0.001 by Mann-Whitney $U$ test comparing LCM and LSM. The RW model was excluded from the statistical comparison because of the low sample number.

The online version of this article includes the following source data for appendix 1—figure 4:

**Appendix 1—figure 4—source data 1.** Data shown in *Appendix 1—figure 4*.

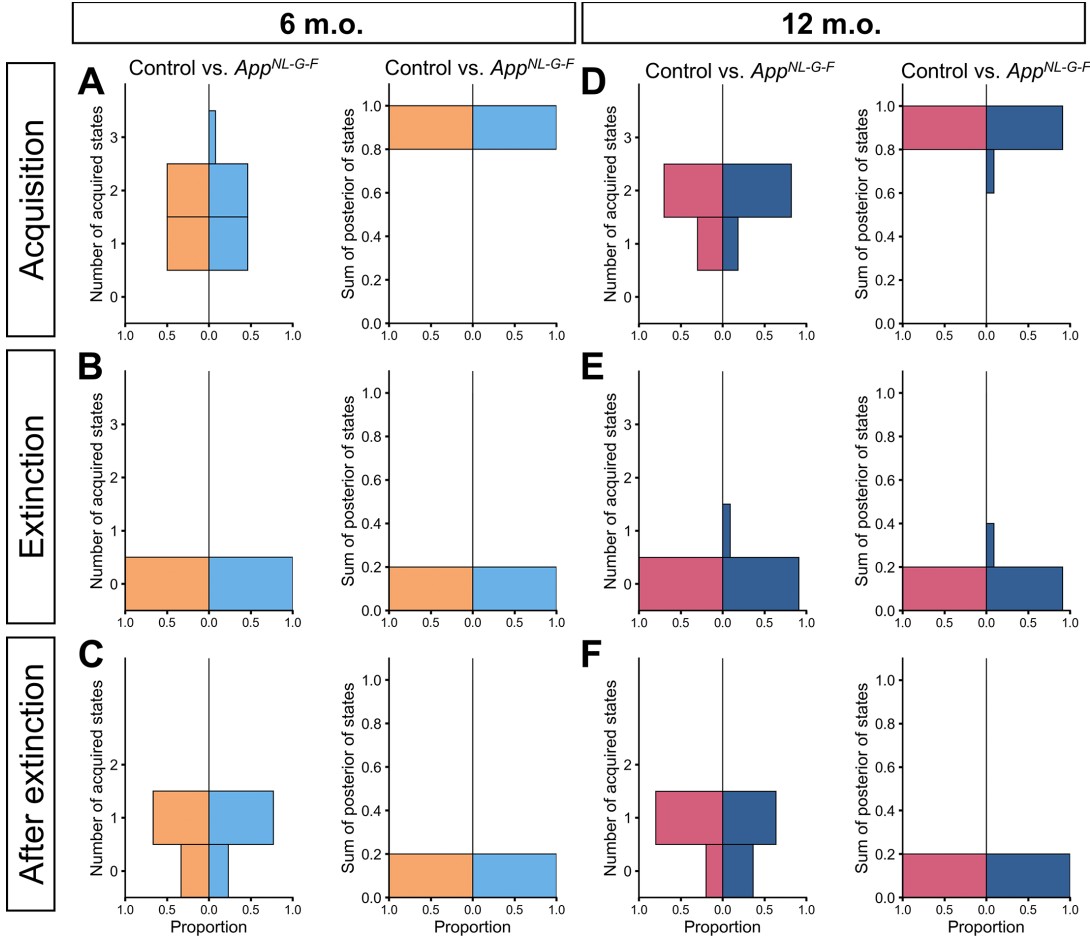

**Appendix 1—figure 5.** The number of latent states inferred in different phases and their posterior probability in test 3. (**A–E**) The figure format is the same as that in *Figures 4C–E and 5C–E* for the latent cause model. The latent states were classified based on the trial they were initially inferred: during the acquisition trials (first row), extinction trials (second row), and trials after extinction (third row). The first and third columns show the count of latent states in the 6-month-old group and 12-month-old group, respectively. The second and fourth columns show the posterior probability of latent states in test 3 in the 6-month-old group and 12-month-old group, respectively. The horizontal axis indicates the proportion of the value in each group. Colors in the histogram indicate different groups: orange represents 6-month-old control (*n* = 6), light blue represents 6-month-old *App*^NL-G-F^ mice (*n* = 13), pink represents 12-month-old control (*n* = 10), and dark blue represents 12-month-old *App*^NL-G-F^ mice (*n* = 11). Each black dot represents one animal.

The online version of this article includes the following source data for appendix 1—figure 5:

**Appendix 1—figure 5—source data 1.** Data shown in *Appendix 1—figure 5*.

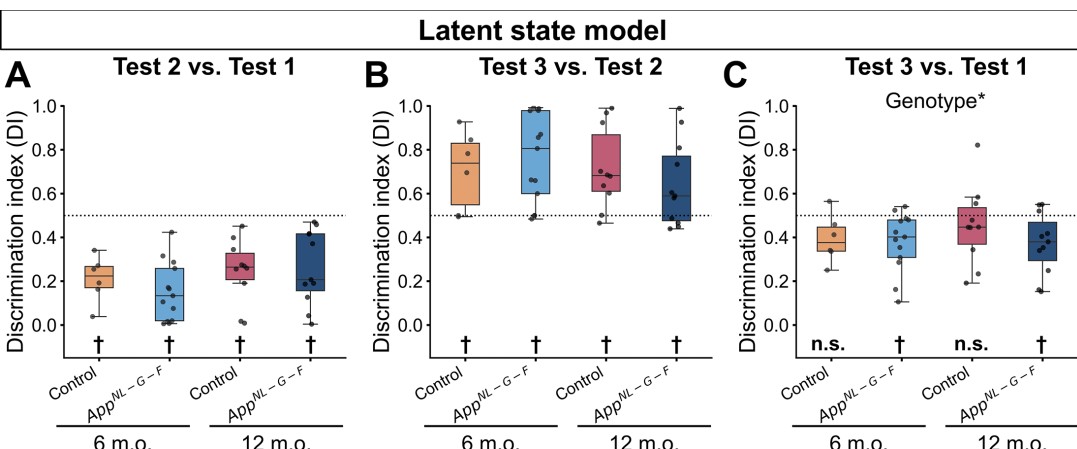

**Appendix 1—figure 6.** Discrimination index (DI) between test 2 and test 1 (**A**), between test 3 and test 2 (**B**), and between test 3 and test 1 (**C**) calculated from simulated conditioned response (CR) in the latent state model. The dashed horizontal line indicates DI = 0.5, which means no discrimination between the two phases. $^{†}p < 0.05$ by one-sample Student's *t*-test, and the alternative hypothesis specifies that the mean differs from 0.5; $^{*}p < 0.05$ by one-way ANCOVA with age as a covariate; No significant difference between control and $App^{NL-G-F}$ mice within the same age was detected by Student's *t*-test comparing. Colors in the box plot indicate different groups: orange represents 6-month-old control (*n* = 6), light blue represents 6-month-old $App^{NL-G-F}$ mice (*n* = 13), pink represents 12-month-old control (*n* = 10), and dark blue represents 12-month-old $App^{NL-G-F}$ mice (*n* = 11). Each black dot represents one animal.

The online version of this article includes the following source data for appendix 1—figure 6:

**Appendix 1—figure 6—source data 1.** Data shown in *Appendix 1—figure 6*.

# Appendix 2

## Relationship between parameters, conditioned response, and internal states

**Appendix 2—table 1.** The initial value, lower bound, and upper bound of parameters in slice sampling.

| Parameters | Initial value | Lower bound | Upper bound |
|---|---|---|---|
| $\alpha$ | 1.5 | 1 | 3 |
| $g$ | 1 | 0.01 | 2 |
| $\eta$ | 0.1 | 0.01 | 1 |
| maxIter* | 3 | 1 | 5 |
| $w_0$ | 0 | –0.01 | 0.01 |
| $\sigma_r^2$ | 0.4 | 0.01 | 3 |
| $\sigma_x^2$ | 1 | 0.01 | 3 |
| $\theta$ | 0.01 | 0.002 | 0.02 |
| $\lambda$ | 0.01 | 0.002 | 0.02 |
| $K$ | 33 | 1 | 36 |

*max. no. of iterations.

**Appendix 2—table 2.** Effect of $\alpha$ and $\eta$ on test trials conditioned response (CR), number of latent causes, and test 3 internal states.

| | | | | | | | | Test 3 | | | | | |
|---|---|---|---|---|---|---|---|---|---|---|---|---|---|
| $\alpha$ | $\eta$ | Test 1 | Test 2 | Test 3 | $K_{acq}$ | $K_{ext}$ | $K_{rem}$ | $P_{acq}$ | $P_{ext}$ | $P_{rem}$ | $w_{context,\,acq}$ | $w_{context,\,ext}$ | $w_{context,\,rem}$ |
| 1 | 0.01 | 0.78 | 0.18 | 0.19 | 2 | 2 | 1 | 0.026 | 0.477 | 0.497 | 0.010 | 0.006 | 0.000 |
| 1 | 0.5 | 1.00 | 0.64 | 0.93 | 2 | 2 | 1 | 0.017 | 0.482 | 0.502 | 0.147 | 0.288 | 0.000 |
| 3 | 0.01 | 0.19 | 0.17 | 0.18 | 3 | 30 | 0 | 0.043 | 0.957 | 0.000 | 0.010 | 0.006 | 0.000 |
| 3 | 0.5 | 1.00 | 0.49 | 1.00 | 2 | 30 | 1 | 0.049 | 0.863 | 0.089 | 0.155 | 0.049 | 0.243 |

Note. This table is used for the reply to reviewer comment #14. The fixed parameter value: $g$ = 1, max. no. of iteration = 3, $w_0$ = 0, $\sigma_r^2$ = 0.4, $\sigma_x^2$ = 1, $\theta$ = 0.01, $\lambda$ = 0.01, $K$ = 33. Test 1, Test 2, Test 3: the simulated CR in each test. $K_{acq}$, $K_{ext}$, $K_{rem}$: the number of latent causes inferred during acquisition phases, extinction phases, and the remaining trials. $P_{acq}$, $P_{ext}$, $P_{rem}$: the posterior probability of latent causes inferred during acquisition phases, extinction phases, and the remaining trials in test 3. $w_{context,\,acq}$, $w_{context,\,ext}$, $w_{context,\,rem}$: the associative weight between context and US in the latent causes inferred during acquisition phases, extinction phases, and the remaining trials in test 3.

**Appendix 2—table 3.** Effect of $w_0$, $\theta$, and $\lambda$ on conditioned response (CR) at each trial in the acquisition phase.

| | | $\lambda$ = 0.01 | | | $\lambda$ = 0.02 | | |
|---|---|---|---|---|---|---|---|
| $w_0$ | $\theta$ | Trial 1 | Trial 2 | Trial 3 | Trial 1 | Trial 2 | Trial 3 |
| –0.01 | 0.002 | 0.080757 | 0.999969 | 1 | 0.241964 | 0.977479 | 1 |
| 0 | 0.002 | 0.42074 | 1 | 1 | 0.460172 | 0.994947 | 1 |
| 0.01 | 0.002 | 0.841345 | 1 | 1 | 0.691462 | 0.999156 | 1 |
| –0.01 | 0.01 | 0.013903 | 0.999333 | 1 | 0.135666 | 0.945672 | 0.999999 |
| 0 | 0.01 | 0.158655 | 0.999993 | 1 | 0.308538 | 0.98508 | 1 |
| 0.01 | 0.01 | 0.57926 | 1 | 1 | 0.539828 | 0.996929 | 1 |

*Appendix 2—table 3 Continued on next page*

*Appendix 2—table 3 Continued*

| | | $\lambda = 0.01$ | | | $\lambda = 0.02$ | | |
|---|---|---|---|---|---|---|---|
| $w_0$ | $\theta$ | Trial 1 | Trial 2 | Trial 3 | Trial 1 | Trial 2 | Trial 3 |
| −0.01 | 0.02 | 0.000687 | 0.986396 | 1 | 0.054799 | 0.865261 | 0.999994 |
| 0 | 0.02 | 0.02275 | 0.999588 | 1 | 0.158655 | 0.952757 | 1 |
| 0.01 | 0.02 | 0.211855 | 0.999996 | 1 | 0.344578 | 0.987459 | 1 |

Note. This table is used for the reply to reviewer comment #26. The fixed parameter value: $\alpha = 1.5$, $g = 1$, $\eta = 0.1$, max. no. of iteration = 3, $\sigma_r^2 = 0.4$, $\sigma_x^2 = 1$, $K = 33$

**Appendix 2—table 4.** Effect of $\alpha$ and $g$ on the number of latent causes inferred at each phase.

| $\alpha$ | $g$ | $K_{acq}$ | $K_{ext}$ | $K_{rem}$ |
|---|---|---|---|---|
| 1 | 0.01 | 2 | 0 | 0 |
| 2 | 0.01 | 2 | 0 | 0 |
| 1 | 1 | 2 | 2 | 1 |
| 2 | 1 | 2 | 30 | 1 |

Note. This table is used for the reply to reviewer comment #29. The fixed parameter value: $\eta = 0.1$, max. no. of iteration = 3, $w_0 = 0$, $\sigma_r^2 = 0.4$, $\sigma_x^2 = 1$, $\theta = 0.01$, $\lambda = 0.01$, $K = 33$. $K_{acq}$, $K_{ext}$, $K_{rem}$: the number of latent causes inferred during acquisition phases, extinction phases, and the remaining trials.

**Appendix 2—table 5.** Variance inflation factor of each parameter.

| | Variance inflation factor | |
|---|---|---|
| Parameter | 6-month-old group | 12-month-old group |
| $\alpha$ | 3.368 | 3.420 |
| $g$ | 2.966 | 1.472 |
| $\eta$ | 3.941 | 1.370 |
| maxIter* | 1.878 | 1.373 |
| $w_0$ | 2.232 | 1.644 |
| $\sigma_r^2$ | 2.292 | 1.610 |
| $\sigma_x^2$ | 2.240 | 3.343 |
| $\theta$ | 2.579 | 1.547 |
| $\lambda$ | 1.431 | 1.923 |
| $K$ | 1.612 | 2.789 |

Note. This table is used for the reply to reviewer comment #31.
*max. no. of iterations.

**Appendix 2—table 6.** The joint effect of $K$, $\alpha$, and $\sigma_x^2$ on conditioned responses (CRs) (test 1 and 3), discrimination index (DI) (test 3 vs. test 1), and the number of latent causes.

| $K$ | $\alpha$ | $\sigma_x^2$ | Test 1 | Test 3 | DI (test 3 vs test 1) | $K_{acq}$ | $K_{ext}$ | $K_{rem}$ |
|---|---|---|---|---|---|---|---|---|
| | 1 | 0.01 | 1.00 | 0.34 | 0.26 | 2 | 0 | 1 |
| | 1 | 3 | 1.00 | 0.81 | 0.45 | 2 | 2 | 0 |
| | 3 | 0.01 | 1.00 | 0.35 | 0.26 | 2 | 0 | 1 |
| 4 | 3 | 3 | 0.42 | 0.74 | 0.64 | 3 | 1 | 0 |

*Appendix 2—table 6 Continued on next page*

*Appendix 2—table 6 Continued*

| K | α | $\sigma_x^2$ | Test 1 | Test 3 | DI (test 3 vs test 1) | $K_{acq}$ | $K_{ext}$ | $K_{rem}$ |
|---|---|---|---|---|---|---|---|---|
| | 1 | 0.01 | 1.00 | 0.34 | 0.26 | 2 | 0 | 1 |
| | 1 | 3 | 1.00 | 0.34 | 0.26 | 2 | 3 | 2 |
| | 3 | 0.01 | 1.00 | 0.35 | 0.26 | 2 | 0 | 1 |
| 36 | 3 | 3 | 0.42 | 0.19 | 0.31 | 3 | 31 | 2 |

Note. This table is used for the reply to reviewer comment #31. The fixed parameter value: $g = 1$, $\eta = 0.1$, max. no. of iteration = 3, $w_0 = 0$, $\sigma_r^2 = 0.4$, $\theta = 0.01$, $\lambda = 0.01$. Test 1, Test 3: the simulated CR. $K_{acq}$, $K_{ext}$, $K_{rem}$: the number of latent causes inferred during acquisition phases, extinction phases, and the remaining trials.

**Appendix 2—table 7.** Effect of $w_0$ on conditioned response (CR) in test trials and number of latent causes.

| $w_0$ | Test 1 | Test 2 | Test 3 | $K_{acq}$ | $K_{ext}$ | $K_{rem}$ |
|---|---|---|---|---|---|---|
| –0.01 | 1 | 0.22 | 0.13 | 2 | 2 | 1 |
| –0.005 | 1 | 0.27 | 0.24 | 2 | 2 | 1 |
| 0 | 1 | 0.32 | 0.37 | 2 | 2 | 1 |
| 0.005 | 1 | 0.38 | 0.52 | 2 | 2 | 1 |
| 0.01 | 1 | 0.44 | 0.66 | 2 | 2 | 1 |

Note. This table is used for the reply to reviewer comment #33. The fixed parameter value: $\alpha = 1.5$, $g = 1$, $\eta = 0.1$, max. no. of iteration = 3, $\sigma_r^2 = 0.4$, $\sigma_x^2 = 1$, $\theta = 0.01$, $\lambda = 0.01$, $K = 33$. Test 1, Test 2, Test 3: the simulated CR in each test. $K_{acq}$, $K_{ext}$, $K_{rem}$: the number of latent causes inferred during acquisition phases, extinction phases, and the remaining trials.

