## [Editor Report · eLife Assessment]

This **valuable** study employs a formalized computational model of learning to assess memory deficits in Alzheimer's Disease with the goal of developing an early diagnosis tool. Using an established mouse model of the disease, the authors studied multiple behavioral tasks and ages with the goal of showing similarities in behavioral deficits across tasks. Using the model, the authors indicate specific deficits in memory (overgeneralization and over differentiation) in mice with the transgene for the disease. The evidence presented is **solid**, yet certain concerns remain regarding the interpretation of the results of the modeling.

---

## [Referee Report · Reviewer #1 (Public review)]

I applaud the authors' for providing a thorough response to my comments from the first round of review. The authors' have addressed the points I raised on the interpretation of the behavioral results as well as the validation of the model (fit to the data) by conducting new analyses, acknowledging the limitations where required and providing important counterpoints. As a result of this process, the manuscript has considerably improved. I have no further comments and recommend this manuscript for publication.

---

## [Referee Report · Reviewer #2 (Public review)]

Summary:

This manuscript proposes that the use of a latent cause model for assessment of memory-based tasks may provide improved early detection in Alzheimer's Disease as well as more differentiated mapping of behavior to underlying causes. To test the validity of this model, the authors use a previously described knock-in mouse model of AD and subject the mice to several behaviors to determine whether the latent cause model may provide informative predictions regarding changes in the observed behaviors. They include a well-established fear learning paradigm in which distinct memories are believed to compete for control of behavior. More specifically, it's been observed that animals undergoing fear learning and subsequent fear extinction develop two separate memories for the acquisition phase and the extinction phase, such that the extinction does not simply 'erase' the previously acquired memory. Many models of learning require the addition of a separate context or state to be added during the extinction phase and are typically modeled by assuming the existence of a new state at the time of extinction. The Niv research group, Gershman et al. 2017, have shown that the use of a latent cause model applied to this behavior can elegantly predict the formation of latent states based on a Bayesian approach, and that these latent states can facilitate the persistence of the acquisition and extinction memory independently. The authors of this manuscript leverage this approach to test whether deficits in production of the internal states, or the inference and learning of those states, may be disrupted in knock-in mice that show both a build-up of amyloid-beta plaques and a deterioration in memory as the mice age.

Strengths:

I think the authors' proposal to leverage the latent cause model and test whether it can lead to improved assessments in an animal model of AD is a promising approach for bridging the gap between clinical and basic research. The authors use a promising mouse model and apply this to a paradigm in which the behavior and neurobiology are relatively well understood - an ideal situation for assessing how a disease state may impact both the neurobiology and behavior. The latent cause model has the potential to better connect observed behavior to underlying causes and may pave a road for improved mapping of changes in behavior to neurobiological mechanisms in diseases such as AD.

The authors also compare the latent cause model to the Rescorla-Wagner model and a latent state model allowing for better assessment of the latent cause model as a strong model for assessing reinstatement.

Weaknesses:

I have several substantial concerns which I've detailed below. These include important details on how the behavior was analyzed, how the model was used to assess the behavior, and the interpretations that have been made based on the model._A_ (the putative acquisition memory), increases, partially leading to reinstatement. This does not appear to be the case as test 3 (day 36) appears to have similar posterior probabilities for _A_ as well as similar weights for the CS as compared to the last days of extinction. Rather, the model appears to mainly modify the weights in the most recent latent cause, _B_ - the putative the 'extinction state', during reinstatement. The authors suggest that previous experimental data have indicated that spontaneous recovery or reinstatement effects are due to an interaction of the acquisition and extinction memory. These studies have shown that conditioned responding at a later time point after extinction is likely due to a balance between the acquisition memory and the extinction memory, and that this balance can shift towards the acquisition memory naturally during spontaneous recovery, or through artificial activation of the acquisition memory or inhibition of the extinction memory (see Lacagnina et al. for example). Here the authors show that the same latent cause learned during extinction, _B_, appears to dominate during the learning phase of reinstatement, with rapid learning to the context - the weight for the context goes up substantially on day 35 - in _B_. This latent cause, _B_, dominates at the reinstatement test, and due to the increased associative strength between the context and shock, there is a strong CR. For the simulation shown in figure 1, it's not clear why a latent cause model is necessary for this behavior. This leads to the next point.

(1) There is substantial data to suggest that during fear learning in mice separate memories develop for the acquisition and extinction phases, with the acquisition memory becoming more strongly retrieved during spontaneous recovery and reinstatement. The Gershman paper, cited by the authors, shows how the latent causal model can predict this shift in latent causes by allowing for the priors to decay over time, thereby increasing the posterior of the acquisition memory at the time of spontaneous recovery. In this manuscript, the authors suggest a similar mechanism of action for reinstatement, yet the model does not appear to return to the acquisition memory after reinstatement, at least based on the simulation and examples shown in figures 1 and 3. More specifically, in figure 1, the authors indicate that the posterior probability of the latent cause**, z**_A_ (the putative acquisition memory), increases, partially leading to reinstatement. This does not appear to be the case as test 3 (day 36) appears to have similar posterior probabilities for **z**_A_ as well as similar weights for the CS as compared to the last days of extinction. Rather, the model appears to mainly modify the weights in the most recent latent cause, **z**_B_ - the putative the 'extinction state', during reinstatement. The authors suggest that previous experimental data have indicated that spontaneous recovery or reinstatement effects are due to an interaction of the acquisition and extinction memory. These studies have shown that conditioned responding at a later time point after extinction is likely due to a balance between the acquisition memory and the extinction memory, and that this balance can shift towards the acquisition memory naturally during spontaneous recovery, or through artificial activation of the acquisition memory or inhibition of the extinction memory (see Lacagnina et al. for example). Here the authors show that the same latent cause learned during extinction, **z**_B_, appears to dominate during the learning phase of reinstatement, with rapid learning to the context - the weight for the context goes up substantially on day 35 - in **z**_B_. This latent cause, **z**_B_, dominates at the reinstatement test, and due to the increased associative strength between the context and shock, there is a strong CR. For the simulation shown in figure 1, it's not clear why a latent cause model is necessary for this behavior. This leads to the next point.

(2) The authors compared the latent cause model to the Rescorla-Wagner model. This is very commendable, particularly since the latent cause model builds upon the RW model, so it can serve as an ideal test for whether a more simplified model can adequately predict the behavior. The authors show that the RW model cannot successfully predict the increased CR during reinstatement (Appendix figure 1). Yet there are some issues with the way the authors have implemented this comparison:

(2A) The RW model is a simplified version of the latent cause model and so should be treated as a nested model when testing, or at a minimum, the number of parameters should be taken into account when comparing the models using a method such as the Bayesian Information Criterion, BIC.

(2B) The RW model provides the associative strength between stimuli and does not necessarily require a linear relationship between V and the CR. This is the case in the original RW model as well as in the LCM. To allow for better comparison between the models, the authors should be modeling the CR in the same manner (using the same probit function) in both models. In fact, there are many instances in which a sigmoid has been applied to RW associative strengths to predict CRs. I would recommend modeling CRs in the RW as if there is just one latent cause. Or perhaps run the analysis for the LCM with just one latent cause - this would effectively reduce the LCM to RW and keep any other assumptions identical across the models._B_, the extinction memory, it may be possible to replicate the pattern of results without requiring a latent cause model. For example, the 12-month-old App NL-G-F mice behavior may have a deficit in learning about the context. Within the RW model, if the alpha for context is set to zero for those mice, but kept higher for the other groups, say alpha_context = 0.8, the authors could potentially observe the same pattern of discrimination indices in figure 2G and 2H at test. Because the authors don't explicitly state which parameters might be driving the change in the DI, the authors should show in some way that their results cannot simply be due to poor contextual learning in the 12 month old App NL-G-F mice, as this can presumably be predicted by the RW model. The authors' model fits using RW don't show this, but this is because they don't consider this possibility that the alpha for context might be disrupted in the 12-month-old App NL-G-F mice. Of course, using the RW model with these alphas won't lead to as nice of fits of the behavior across acquisition, extinction, and reinstatement as the authors' LCM, the number of parameters are substantially reduced in the RW model. Yet the important pattern of the DI would be replicated with the RW model (if I'm not mistaken), which is the important test for assessment of reinstatement.

(2C) In the paper, the model fits for the alphas in the RW model are the same across the groups. Were the alphas for the two models kept as free variables? This is an important question as it gets back to the first point raised. Because the modeling of the reinstatement behavior with the LCM appears to be mainly driven by latent cause **z**_B_, the extinction memory, it may be possible to replicate the pattern of results without requiring a latent cause model. For example, the 12-month-old App NL-G-F mice behavior may have a deficit in learning about the context. Within the RW model, if the alpha for context is set to zero for those mice, but kept higher for the other groups, say alpha_context = 0.8, the authors could potentially observe the same pattern of discrimination indices in figure 2G and 2H at test. Because the authors don't explicitly state which parameters might be driving the change in the DI, the authors should show in some way that their results cannot simply be due to poor contextual learning in the 12 month old App NL-G-F mice, as this can presumably be predicted by the RW model. The authors' model fits using RW don't show this, but this is because they don't consider this possibility that the alpha for context might be disrupted in the 12-month-old App NL-G-F mice. Of course, using the RW model with these alphas won't lead to as nice of fits of the behavior across acquisition, extinction, and reinstatement as the authors' LCM, the number of parameters are substantially reduced in the RW model. Yet the important pattern of the DI would be replicated with the RW model (if I'm not mistaken), which is the important test for assessment of reinstatement.

(3) As stated by the authors in the introduction, the advantage of the fear learning approach is that the memory is modified across the acquisition-extinction-reinstatement phases. Although perhaps not explicitly stated by the authors, the post-reinstatement test (test 3) is the crucial test for whether there is reactivation of a previously stored memory, with the general argument being that the reinvigorated response to the CS can't simply be explained by relearning the CS-US pairing, because re-exposure the US alone leads to increase response to the CS at test. Of course there are several explanations for why this may occur, particularly when also considering the context as a stimulus. This is what I understood to be the justification for the use of a model, such as the latent cause model, that may better capture and compare these possibilities within a single framework. As such, it is critical to look at the level of responding to both the context alone and to the CS. It appears that the authors only look at the percent freezing during the CS, and it is not clear whether this is due to the contextual-US learning during the US re-exposure or to increased responding to the CS - presumably caused by reactivation of the acquisition memory. The authors do perform a comparison between the preCS and CS period, but it is not clear whether this is taken into account in the LCM. For example, the instance of the model shown in figure 1 indicates that the 'extinction cause', or cause z6, develops a strong weight for the context during the reinstatement phase of presenting the shock alone. This state then leads to increased freezing during the final CS probe test as shown in the figure. If they haven't already, I think the authors must somehow incorporate these different phases (CS vs ITI) into their model, particularly since this type of memory retrieval that depends on assessing latent states is specifically why the authors justified using the latent causal model. In more precise terms, it's not clear whether the authors incorporate a preCS/ITI period each day the cue is presented as a vector of just the context in addition to the CS period in which the vector contains both the context and the CS. Based on the description, it seemed to me that they only model the CRs during the CS period on days when the CS is presented, and thereby the context is only ever modeled on its own (as just the context by itself in the vector) on extinction days when the CS is not presented. If they are modeling both timepoints each day that the CS I presented, then I would recommend explicitly stating this in the methods section.

(4) The authors fit the model using all data points across acquisition and learning. As one of the other reviewers has highlighted, it appears that there is a high chance for overfitting the data with the LCM. Of course, this would result in much better fits than models with substantially fewer free parameters, such as the RW model. As mentioned above, the authors should use a method that takes into account the number of parameters, such as the BIC.

(5) The authors have stated that they do not think the Barnes maze task can be modeled with the LCM. Whether or not this is the case, if the authors do not model this data with the LCM, the Barnes maze data doesn't appear valuable to the main hypothesis. The authors suggest that more sophisticated models such as the LCM may be beneficial for early detection of diseases such as Alzheimer's, so the Barnes maze data is not valuable for providing evidence of this hypothesis. Rather, the authors make an argument that the memory deficits in the Barnes maze mimic the reinstatement effects providing support that memory is disrupted similarly in these mice. Although, the authors state that the deficits in memory retrieval are similar across the two tasks, the authors are not explicit as to the precise deficits in memory retrieval in the reinstatement task - it's a combination of overgeneralizing latent causes during acquisition, poor learning rate, over differentiation of the stimuli.

---

## [Referee Report · Reviewer #3 (Public review)]

Summary:

This paper seeks to identify underlying mechanisms contributing to memory deficits observed in Alzheimer's disease (AD) mouse models. By understanding these mechanisms, they hope to uncover insights into subtle cognitive changes early in AD to inform interventions for early-stage decline.

Strengths:

The paper provides a comprehensive exploration of memory deficits in an AD mouse model, covering early and late stages of the disease. The experimental design was robust, confirming age-dependent increases in Aβ plaque accumulation in the AD model mice and using multiple behavior tasks that collectively highlighted difficulties in maintaining multiple competing memory cues, with deficits most pronounced in older mice.

In the fear acquisition, extinction, and reinstatement task, AD model mice exhibited a significantly higher fear response after acquisition compared to controls, as well as a greater drop in fear response during reinstatement. These findings suggest that AD mice struggle to retain the fear memory associated with the conditioned stimulus, with the group differences being more pronounced in the older mice.

In the reversal Barnes maze task, the AD model mice displayed a tendency to explore the maze perimeter rather than the two potential target holes, indicating a failure to integrate multiple memory cues into their strategy. This contrasted with the control mice, which used the more confirmatory strategy of focusing on the two target holes. Despite this, the AD mice were quicker to reach the target hole, suggesting that their impairments were specific to memory retrieval rather than basic task performance.

The authors strengthened their findings by analyzing their data with a leading computational model, which describes how animals balance competing memories. They found that AD mice showed somewhat of a contradiction: a tendency to both treat trials as more alike than they are (lower α) and similar stimuli as more distinct than they are (lower *σ*_*x*_) compared to controls.

Weaknesses:

While conceptually solid, the model struggles to fit the data and to support the key hypothesis about AD mice's inability to retain competing memories. These issues are evident in Figure 3:

(1) The model misses trends in the data, including the gradual learning of fear in all groups during acquisition, the absence of a fear response at the start of the experiment, and the faster return of fear during reinstatement compared to the gradual learning of fear during acquisition. It also underestimates the increase in fear at the start of day 2 of extinction, particularly in controls.

(2) The model explains the higher fear response in controls during reinstatement largely through a stronger association to the context formed during the unsignaled shock phase, rather than to any memory of the conditioned stimulus from acquisition (as seen in Figure 3C). In the experiment, however, this memory does seem to be important for explaining the higher fear response in controls during reinstatement (as seen in Author Response Figure 3). The model does show a necessary condition for memory retrieval, which is that controls rely more on the latent causes from acquisition. But this alone is not sufficient, since the associations within that cause may have been overwritten during extinction. The Rescorla-Wagner model illustrates this point: it too uses the latent cause from acquisition (as it only ever uses a single cause across phases) but does not retain the original stimulus-shock memory, updating and overwriting it continuously. Similarly, the latent cause model may reuse a cause from acquisition without preserving its original stimulus-shock association.

These issues lead to potential overinterpretation of the model parameters. The differences in α and σx are being used to make claims about cognitive processes (e.g., overgeneralization vs. over differentiation), but the model itself does not appear to capture these processes accurately.

The authors could benefit from a model that better matches the data and captures the retention and retrieval of fear memories across phases. While they explored alternatives, including the Rescorla-Wagner model and a latent state model, these showed no meaningful improvement in fit. This highlights a broader issue: these models are well-motivated but may not fully capture observed behavior.

Conclusion:

Overall, the data support the authors' hypothesis that AD model mice struggle to retain competing memories, with the effect becoming more pronounced with age. While I believe the right computational model could highlight these differences, the current models fall short in doing so.

---

## [Author Response]

The following is the authors’ response to the current reviews.

**Public Reviews:**

**Reviewer #1 (Public review):**
I applaud the authors' for providing a thorough response to my comments from the first round of review. The authors' have addressed the points I raised on the interpretation of the behavioral results as well as the validation of the model (fit to the data) by conducting new analyses, acknowledging the limitations where required and providing important counterpoints. As a result of this process, the manuscript has considerably improved. I have no further comments and recommend this manuscript for publication.

We are pleased that our revisions have addressed all the concerns raised by Reviewer #1.

**Reviewer #2 (Public review):**
Summary:This manuscript proposes that the use of a latent cause model for assessment of memory-based tasks may provide improved early detection in Alzheimer's Disease as well as more differentiated mapping of behavior to underlying causes. To test the validity of this model, the authors use a previously described knock-in mouse model of AD and subject the mice to several behaviors to determine whether the latent cause model may provide informative predictions regarding changes in the observed behaviors. They include a well-established fear learning paradigm in which distinct memories are believed to compete for control of behavior. More specifically, it's been observed that animals undergoing fear learning and subsequent fear extinction develop two separate memories for the acquisition phase and the extinction phase, such that the extinction does not simply 'erase' the previously acquired memory. Many models of learning require the addition of a separate context or state to be added during the extinction phase and are typically modeled by assuming the existence of a new state at the time of extinction. The Niv research group, Gershman et al. 2017, have shown that the use of a latent cause model applied to this behavior can elegantly predict the formation of latent states based on a Bayesian approach, and that these latent states can facilitate the persistence of the acquisition and extinction memory independently. The authors of this manuscript leverage this approach to test whether deficits in production of the internal states, or the inference and learning of those states, may be disrupted in knock-in mice that show both a build-up of amyloid-beta plaques and a deterioration in memory as the mice age.Strengths:I think the authors' proposal to leverage the latent cause model and test whether it can lead to improved assessments in an animal model of AD is a promising approach for bridging the gap between clinical and basic research. The authors use a promising mouse model and apply this to a paradigm in which the behavior and neurobiology are relatively well understood - an ideal situation for assessing how a disease state may impact both the neurobiology and behavior. The latent cause model has the potential to better connect observed behavior to underlying causes and may pave a road for improved mapping of changes in behavior to neurobiological mechanisms in diseases such as AD.The authors also compare the latent cause model to the Rescorla-Wagner model and a latent state model allowing for better assessment of the latent cause model as a strong model for assessing reinstatement.Weaknesses:I have several substantial concerns which I've detailed below. These include important details on how the behavior was analyzed, how the model was used to assess the behavior, and the interpretations that have been made based on the model.(1) There is substantial data to suggest that during fear learning in mice separate memories develop for the acquisition and extinction phases, with the acquisition memory becoming more strongly retrieved during spontaneous recovery and reinstatement. The Gershman paper, cited by the authors, shows how the latent causal model can predict this shift in latent causes by allowing for the priors to decay over time, thereby increasing the posterior of the acquisition memory at the time of spontaneous recovery. In this manuscript, the authors suggest a similar mechanism of action for reinstatement, yet the model does not appear to return to the acquisition memory after reinstatement, at least based on the simulation and examples shown in figures 1 and 3. More specifically, in figure 1, the authors indicate that the posterior probability of the latent cause,**z**_A_ (the putative acquisition memory), increases, partially leading to reinstatement. This does not appear to be the case as test 3 (day 36) appears to have similar posterior probabilities for **z**_A_ as well as similar weights for the CS as compared to the last days of extinction. Rather, the model appears to mainly modify the weights in the most recent latent cause, **z**_B_ - the putative the 'extinction state', during reinstatement. The authors suggest that previous experimental data have indicated that spontaneous recovery or reinstatement effects are due to an interaction of the acquisition and extinction memory. These studies have shown that conditioned responding at a later time point after extinction is likely due to a balance between the acquisition memory and the extinction memory, and that this balance can shift towards the acquisition memory naturally during spontaneous recovery, or through artificial activation of the acquisition memory or inhibition of the extinction memory (see Lacagnina et al. for example). Here the authors show that the same latent cause learned during extinction, **z**_B_, appears to dominate during the learning phase of reinstatement, with rapid learning to the context - the weight for the context goes up substantially on day 35 - in **z**_B_. This latent cause, **z**_B_, dominates at the reinstatement test, and due to the increased associative strength between the context and shock, there is a strong CR. For the simulation shown in figure 1, it's not clear why a latent cause model is necessary for this behavior. This leads to the next point.

We would like to first clarify that our behavioral paradigm did not last for 36 days, as noted by the reviewer. Our reinstatement paradigm contained 7 phases and 36 trials in total: acquisition (3 trials), test 1 (1 trial), extinction 1 (19 trials), extinction 2 (10 trials), test 2 (1 trial), unsignaled shock (1 trial), test 3 (1 trial). The day is labeled under each phase in Figure 2A.

We have provided explanations on how the reinstatement is explained by the latent cause model in the first round of the review. Briefly, both acquisition and extinction latent causes contribute to the reinstatement (test 3). The former retains the acquisition fear memory, and the latter has the updated *w_context_* from unsignaled shock. Although the reviewer is correct that the **z**_B_ in Figure 1D makes a great contribution during the reinstatement, we would like to argue that the elevated CR from test 2 (trial 34) to test 3 (trial 36) is the result of the interaction between **z**_A_ and **z**_B_.

We provided Author response image 1 using the same data in Figure 1D and 1E to further clarify this point. The posterior probability of **z**_A_ increased after an unsignaled shock (trial 35), which may be attributed to the return of acquisition fear memory. The posterior probability of **z**_A_ then decreased again after test 3 (trial 36) because there was no shock in this trial. Along with the weight change, the expected shock change substantially in these three trials, resulting in reinstatement. Note that the mapping of expected shock to CR in the latent cause model is controlled by parameter *θ* and *λ*. Once the expected shock exceeds the threshold *θ*, the CR will increase rapidly if *λ* is smaller.

Lastly, accepting the idea that separate memories are responsible for acquisition and extinction in the memory modification paradigm, the latent cause model (LCM) is a rational candidate modeling this idea. Please see the following reply on why a simple model like the Rescorla-Wagner (RW) model is not sufficient to fully explain the behaviors observed in this study.

**Author response image 1. sa4fig1:** The sum posterior probability (A), the sum of associative weight of CS (B), and the sum of associative weight of context (C) of acquisition and extinction latent causes in Figure 1D and 1E.

(2) The authors compared the latent cause model to the Rescorla-Wagner model. This is very commendable, particularly since the latent cause model builds upon the RW model, so it can serve as an ideal test for whether a more simplified model can adequately predict the behavior. The authors show that the RW model cannot successfully predict the increased CR during reinstatement (Appendix figure 1). Yet there are some issues with the way the authors have implemented this comparison:(2A) The RW model is a simplified version of the latent cause model and so should be treated as a nested model when testing, or at a minimum, the number of parameters should be taken into account when comparing the models using a method such as the Bayesian Information Criterion, BIC.

We acknowledge that the number of parameters was not taken into consideration when we compared the models. We thank the reviewer for the suggestion to use the Bayesian Information Criterion (BIC). However, we did not use BIC in this study for the following reasons. We wanted a model that can explain fear conditioning, extinction and reinstatement, so our first priority is to fit the test phases. Models that simulate CRs well in non-test phases can yield lower BIC values even if they fail to capture reinstatement. When we calculate the BIC by using the half normal distribution (*μ* = 0, *σ* = 0.3) as the likelihood for prediction error in each trial, the BIC of the 12-month-old control is -37.21 for the RW model (Appendix 1–figure 1C) and -11.60 for the LCM (Figure 3C). Based on this result, the RW model would be preferred, yet the LCM was penalized by the number of parameters, even though it fit better in trial 36. Because we did not think this aligned with our purpose to model reinstatement, we chose to rely on the practical criteria to determine whether the estimated parameter set is accepted or not for our purpose (see Materials and Methods). The number of accepted samples can thus roughly be seen as the model's ability to explain the data in this study. These exclusion criteria then created imbalances in accepted samples across models (Appendix 1–figure 2). In the RW model, only one or two samples met the criteria, preventing meaningful statistical comparisons of BIC within each group. Overall, though we agreed that BIC is one of the reasonable metrics in model comparison, we did not think it aligns with our purpose in this study.

(2B) The RW model provides the associative strength between stimuli and does not necessarily require a linear relationship between V and the CR. This is the case in the original RW model as well as in the LCM. To allow for better comparison between the models, the authors should be modeling the CR in the same manner (using the same probit function) in both models. In fact, there are many instances in which a sigmoid has been applied to RW associative strengths to predict CRs. I would recommend modeling CRs in the RW as if there is just one latent cause. Or perhaps run the analysis for the LCM with just one latent cause - this would effectively reduce the LCM to RW and keep any other assumptions identical across the models.

Regarding the suggestion to run the analysis using the LCM with one latent cause, we agree that this method is almost identical to the RW model, which is also mentioned in the original paper (Gershman et al., 2017). Importantly, it would also eliminate the RW model’s advantage of assigning distinct learning rates to different stimuli, highlighted in the next comment (2C).

We thank the reviewer for suggesting applying the transformation of associative strength (*V*) to CR as in the LCM. We examined this possibility by heuristically selecting parameter values to test how such a transformation would influence the RW model (Author response image 2A). Specifically, we set *α_CS_* = 0.5, *α_context_* = 1, *β* = 1, and introduced the additional parameters *θ* and *λ*, as in the LCM. This parameter set is determined heuristically to address the reviewer’s concern about a higher learning rate of context. The dark blue line is the plain associative strength. The remaining lines are CR curves under different combinations of *θ* and *λ*.

Consistent with the reviewer’s comment, under certain parameter settings (*θ* = 0.01, *λ* = 0.01), the extended RW model can reproduce higher CRs at test 3, thereby approximating the discrimination index observed in the 12-month-old control group. However, this modification changes the characteristics of CRs in other phases from those in the plain RW model. In the acquisition phase, the CRs rise more sharply. In the extinction phase, the CRs remain high when *θ* is small. Though changing *λ* can modulate the steepness, the CR curve is flat on the second day of the extinction phase, which does not reproduce the pattern in observed data (Figure 2B). These trade-offs suggest that the RW model with the sigmoid transformation does not improve fit quality and, in fact, sacrifices features that were well captured by simpler RW simulations (Appendix 1–figure 1A to 1D). To further evaluate this extended RW model (RW*), we applied the same parameter estimation method used in the LCM for individual data (see Materials and Methods). For each animal, *α_CS_*, *α_context_*, *β*, *θ*, and *λ* were estimated with their lower and upper bounds set as previously described (see Appendix 1, Materials and Methods). The results showed that the number of accepted samples slightly increased compared to the RW model without sigmoidal transformation of CR (RW* vs. RW in Author response image 2B, 2C). However, this improvement did not surpass the LCM (RW* vs. LCM in Author response image 2B, Author response image 1C). Overall, these results suggest that while using the same method to map the expected shock to CR, the RW model does not outperform the LCM. Practically, further extension, such as adding novel terms, might improve the fitting level. We would like to note that such extensions should be carefully validated if they are reasonable and necessary for an internal model, which is beyond the scope of this study. We hope this addresses the reviewer's concerns about the implementation of the RW model.

**Author response image 2. sa4fig2:** Simulation (A) and parameter estimation (B and C) in the extended Rescorla-Wagner model.

(2C) In the paper, the model fits for the alphas in the RW model are the same across the groups. Were the alphas for the two models kept as free variables? This is an important question as it gets back to the first point raised. Because the modeling of the reinstatement behavior with the LCM appears to be mainly driven by latent cause **z**_B_, the extinction memory, it may be possible to replicate the pattern of results without requiring a latent cause model. For example, the 12-month-old App NL-G-F mice behavior may have a deficit in learning about the context. Within the RW model, if the alpha for context is set to zero for those mice, but kept higher for the other groups, say alpha_context = 0.8, the authors could potentially observe the same pattern of discrimination indices in figure 2G and 2H at test. Because the authors don't explicitly state which parameters might be driving the change in the DI, the authors should show in some way that their results cannot simply be due to poor contextual learning in the 12 month old App NL-G-F mice, as this can presumably be predicted by the RW model. The authors' model fits using RW don't show this, but this is because they don't consider this possibility that the alpha for context might be disrupted in the 12-month-old App NL-G-F mice. Of course, using the RW model with these alphas won't lead to as nice of fits of the behavior across acquisition, extinction, and reinstatement as the authors' LCM, the number of parameters are substantially reduced in the RW model. Yet the important pattern of the DI would be replicated with the RW model (if I'm not mistaken), which is the important test for assessment of reinstatement.

We would like to clarify that we estimated three parameters in the RW model for individuals: *α_CS_*, *α_context_*, and *β*. Even if we did so, many samples did not satisfy our criteria (Appendix 1–figure 2). Please refer to the “Evaluation of model fit” in Appendix 1 and the legend of Appendix 1–figure 1A to 1D, where we have written the estimated parameter values.

We did not agree that paralyzing the contextual learning by setting *α_context_* as 0 in the RW model can explain the CR curve of 12-month-old AD mice well. Specifically, the RW model cannot capture the between-day extinction dynamics (i.e., the increase in CR at the beginning of day 2 extinction) and the higher CR at test 3 relative to test 2 (i.e., DI between test 3 and test 2 is greater than 0.5). In addition, because the context input (= 0.2) was relatively lower than the CS input (= 1), and there is only a single unsignaled shock trial, even setting *α_context_* = 1 results in only a limited increase in CR (Appendix 1–figure 1A to 1D; see also Author response image 2). Thus, the RW model cannot replicate the reinstatement effect or the critical pattern of discrimination index, even under conditions of stronger contextual learning.

(3) As stated by the authors in the introduction, the advantage of the fear learning approach is that the memory is modified across the acquisition-extinction-reinstatement phases. Although perhaps not explicitly stated by the authors, the post-reinstatement test (test 3) is the crucial test for whether there is reactivation of a previously stored memory, with the general argument being that the reinvigorated response to the CS can't simply be explained by relearning the CS-US pairing, because re-exposure the US alone leads to increase response to the CS at test. Of course there are several explanations for why this may occur, particularly when also considering the context as a stimulus. This is what I understood to be the justification for the use of a model, such as the latent cause model, that may better capture and compare these possibilities within a single framework. As such, it is critical to look at the level of responding to both the context alone and to the CS. It appears that the authors only look at the percent freezing during the CS, and it is not clear whether this is due to the contextual-US learning during the US re-exposure or to increased responding to the CS - presumably caused by reactivation of the acquisition memory. The authors do perform a comparison between the preCS and CS period, but it is not clear whether this is taken into account in the LCM. For example, the instance of the model shown in figure 1 indicates that the 'extinction cause', or cause z6, develops a strong weight for the context during the reinstatement phase of presenting the shock alone. This state then leads to increased freezing during the final CS probe test as shown in the figure. If they haven't already, I think the authors must somehow incorporate these different phases (CS vs ITI) into their model, particularly since this type of memory retrieval that depends on assessing latent states is specifically why the authors justified using the latent causal model. In more precise terms, it's not clear whether the authors incorporate a preCS/ITI period each day the cue is presented as a vector of just the context in addition to the CS period in which the vector contains both the context and the CS. Based on the description, it seemed to me that they only model the CRs during the CS period on days when the CS is presented, and thereby the context is only ever modeled on its own (as just the context by itself in the vector) on extinction days when the CS is not presented. If they are modeling both timepoints each day that the CS I presented, then I would recommend explicitly stating this in the methods section.

In this study, we did not model the preCS freezing rate, and we thank the reviewer for the suggestion to model preCS periods as separate context-only trials. In our view, however, this approach is not consistent with the assumptions of the LCM. Our rationale is that the available periods of context and the CS are different. We assume that observation of the context lasts from preCS to CS. If we simulate both preCS (context) and CS (context and tone), the weight of context would be updated twice. Instead, we follow the same method as described in the original code from Gershman et al. (2017) to consider the context effect. We agree that explicitly modeling preCS could provide additional insights, but we believe it would require modifying or extending the LCM. We consider this an important direction for future research, but it is outside the scope of this study.

(4) The authors fit the model using all data points across acquisition and learning. As one of the other reviewers has highlighted, it appears that there is a high chance for overfitting the data with the LCM. Of course, this would result in much better fits than models with substantially fewer free parameters, such as the RW model. As mentioned above, the authors should use a method that takes into account the number of parameters, such as the BIC.

Please refer to the reply to public review (2A) for the reason we did not take the suggestion to use BIC. In addition, we feel that we have adequately addressed the concern of overfitting in the first round of the review.

(5) The authors have stated that they do not think the Barnes maze task can be modeled with the LCM. Whether or not this is the case, if the authors do not model this data with the LCM, the Barnes maze data doesn't appear valuable to the main hypothesis. The authors suggest that more sophisticated models such as the LCM may be beneficial for early detection of diseases such as Alzheimer's, so the Barnes maze data is not valuable for providing evidence of this hypothesis. Rather, the authors make an argument that the memory deficits in the Barnes maze mimic the reinstatement effects providing support that memory is disrupted similarly in these mice. Although, the authors state that the deficits in memory retrieval are similar across the two tasks, the authors are not explicit as to the precise deficits in memory retrieval in the reinstatement task - it's a combination of overgeneralizing latent causes during acquisition, poor learning rate, over differentiation of the stimuli.

We would like to clarify that we valued the latent cause model not solely because it is more sophisticated and fits more data points, but it is an internal model that implicates the cognitive process. Please also see the reply to the recommendations to authors (3) about the reason why we did not take the suggestion to remove this data.

**Reviewer #3 (Public review):**
Summary:This paper seeks to identify underlying mechanisms contributing to memory deficits observed in Alzheimer's disease (AD) mouse models. By understanding these mechanisms, they hope to uncover insights into subtle cognitive changes early in AD to inform interventions for early-stage decline.Strengths:The paper provides a comprehensive exploration of memory deficits in an AD mouse model, covering early and late stages of the disease. The experimental design was robust, confirming age-dependent increases in Aβ plaque accumulation in the AD model mice and using multiple behavior tasks that collectively highlighted difficulties in maintaining multiple competing memory cues, with deficits most pronounced in older mice.In the fear acquisition, extinction, and reinstatement task, AD model mice exhibited a significantly higher fear response after acquisition compared to controls, as well as a greater drop in fear response during reinstatement. These findings suggest that AD mice struggle to retain the fear memory associated with the conditioned stimulus, with the group differences being more pronounced in the older mice.In the reversal Barnes maze task, the AD model mice displayed a tendency to explore the maze perimeter rather than the two potential target holes, indicating a failure to integrate multiple memory cues into their strategy. This contrasted with the control mice, which used the more confirmatory strategy of focusing on the two target holes. Despite this, the AD mice were quicker to reach the target hole, suggesting that their impairments were specific to memory retrieval rather than basic task performance.The authors strengthened their findings by analyzing their data with a leading computational model, which describes how animals balance competing memories. They found that AD mice showed somewhat of a contradiction: a tendency to both treat trials as more alike than they are (lower α) and similar stimuli as more distinct than they are (lower σx) compared to controls.Weaknesses:While conceptually solid, the model struggles to fit the data and to support the key hypothesis about AD mice's inability to retain competing memories. These issues are evident in Figure 3:(1) The model misses trends in the data, including the gradual learning of fear in all groups during acquisition, the absence of a fear response at the start of the experiment, and the faster return of fear during reinstatement compared to the gradual learning of fear during acquisition. It also underestimates the increase in fear at the start of day 2 of extinction, particularly in controls.(2) The model explains the higher fear response in controls during reinstatement largely through a stronger association to the context formed during the unsignaled shock phase, rather than to any memory of the conditioned stimulus from acquisition (as seen in Figure 3C). In the experiment, however, this memory does seem to be important for explaining the higher fear response in controls during reinstatement (as seen in Author Response Figure 3). The model does show a necessary condition for memory retrieval, which is that controls rely more on the latent causes from acquisition. But this alone is not sufficient, since the associations within that cause may have been overwritten during extinction. The Rescorla-Wagner model illustrates this point: it too uses the latent cause from acquisition (as it only ever uses a single cause across phases) but does not retain the original stimulus-shock memory, updating and overwriting it continuously. Similarly, the latent cause model may reuse a cause from acquisition without preserving its original stimulus-shock association.These issues lead to potential overinterpretation of the model parameters. The differences in α and σx are being used to make claims about cognitive processes (e.g., overgeneralization vs. over differentiation), but the model itself does not appear to capture these processes accurately.The authors could benefit from a model that better matches the data and captures the retention and retrieval of fear memories across phases. While they explored alternatives, including the Rescorla-Wagner model and a latent state model, these showed no meaningful improvement in fit. This highlights a broader issue: these models are well-motivated but may not fully capture observed behavior.Conclusion:Overall, the data support the authors' hypothesis that AD model mice struggle to retain competing memories, with the effect becoming more pronounced with age. While I believe the right computational model could highlight these differences, the current models fall short in doing so.

We thank the reviewer for the insightful comments. For the comments (1) and (2), please refer to our previous author response to comments #26 and #27. We recognize that the models tested in this study have limitations and, as noted, do not fully capture all aspects of the observed behavioral data. We see this as an important direction for future research and value the reviewer’s suggestions.

**Recommendations for the authors:**

**Reviewer #2 (Recommendations for the authors):**
I have maintained some of the main concerns included in the first round of reviews as I think they remain concerns with the new draft, even though the authors have included substantially more analysis of their data, which is appreciated. I particularly found the inclusion of the comparative modeling valuable, although I think the analysis comparing the models should be improved.(1) This relates to point 1 in the public assessment or #16 in the response to reviewers from the authors. The authors raise the point that even a low posterior can drive behavioral expression (lines 361-365 in the response to authors), and so the acquisition latent cause may partially drive reinstatement. Yet in the stimulation shown in figure 1D, this does not seem to be the case. As I mentioned in the public response, in figure 1, the posteriors for **z**_A_ are similar on day 34 and day 36, yet only on day 36 is there a strong CR. At least in this example, it does not appear that **z**_A_ contributes to the increased responding from day 34 (test 2) to day 36 (test 3). There may be a slight increase in z1 in figure 3C, but the dominant change from day 34 to day 36 appears to be the increase in the posterior of z3 and the substantial increase in w3. The authors then cite several papers which have shown the shift in balance between what it is the putative acquisition memory and extinction memory (i.e. Lacagnina et al.). Yet I do not see how this modeling fits with most of the previous findings. For example, in the Lacagnina et al. paper, activation of the acquisition ensemble or inhibition of the extinction ensemble drives freezing, whereas the opposite pattern reduces freezing. What appears to be the pattern in the modeling in this paper is primarily learning of context in the extinction latent cause to predict the shock. As I mention in point 2C of the public review, it's not clear why this pattern of results would require a latent cause model. Would a high alpha for context and not the CS not give a similar pattern of results in the RW model? At least for giving similar results of the DIs in figure 2?

First, we would like to clarify that the x-axis in Figure 1D is labeled “Trial,” not “Day.” Please refer to the reply to public review (1), where we clarified the posterior probability of the latent cause from trials 34 to 36. Second, although we did not have direct neural circuit evidence in this study, we discussed the similarities between previous findings and the modeling in the first review. Briefly, our main point focuses on the interaction between acquisition and extinction memory. In other words, responses at different times arise from distinct internal states made up of competing memories. We assume that the reviewer expects a modeling result showing nearly full recovery of acquisition memory, which aligns with previous findings where optogenetic activation of the acquisition engram can partially mimic reinstatement (Zaki et al., 2022; see also the response to comment #12 in the first round of review). We acknowledge that such a modeling result cannot be achieved with the latent cause model and see it as a potential future direction for model improvement.

Please also refer to the reply to public review (2) about how a high alpha for context in the RW model cannot explain the pattern we observed in the reinstatement paradigm.

(2) This is related to point 3 in the public comments and #13 in the response to reviewers. I raised the question of comparing the preCS/ITI period with the CS period, but my main point was why not include these periods in the LCM itself as mentioned in more detail in point 3 in the current public review. The inclusion of the comparisons the authors performed helped, but my main point was that the authors could have a better measure of wcontext if they included the preCS period as a stimulus each day (when only the context is included in the stimulus). This would provide better estimates of wcontext. As stated in the public review, perhaps the authors did this, but my understanding of the methods this was not the case, rather, it seems the authors only included the CS period for CRs within the model (at least on days when the CS was present).

Please refer to the reply to public review (3) about the reason why we did not model the preCS freezing rate.

(3) This relates to point 4 in the public review and #15 and #24 in the response to authors. The authors have several points for why the two experiments are similar and how results may be extrapolated - lines 725-733. The first point is that associative learning is fundamental in spatial learning. I'm not sure that this broad connection between the two studies is particularly insightful for why one supports the other as associative learning is putatively involved in most behavioral tasks. In the second point about reversals, why not then use a reversal paradigm that would be easier to model with LCM? This data is certainly valuable and interesting, yet I don't think it's helpful for this paper to state qualitatively the similarities in the potential ways a latent cause framework might predict behavior on the Barnes maze. I would recommend that the authors either model the behavior with LCM, remove the experiment from the paper, or change the framing of the paper that LCM might be an ideal approach for early detection of dementia or Alzheimer's disease.

We would like to clarify that our aim was not to present the LCM as an ideal tool for early detection of AD symptoms. Rather, our focus is on the broader idea of utilizing internal models and estimating individual internal states in early-stage AD. Regarding using a reversal paradigm that would be easier to model with LCM, the most straightforward approach is to use another type of paradigm for fear conditioning, then to examine the extent to which similar behavioral characteristics are observed between paradigms within subjects. However, re-exposing the same mice to such paradigms is constrained by strong carry-over effects, limiting the feasibility of this experiment. Other behavioral tasks relevant to AD that avoid shock generally involve action selection for subsequent observation (Webster et al., 2014), which falls outside the structure of LCM. Our rationale for including the Barnes maze task is that spatial memory deficit is implicated in the early stage of AD, making it relevant for translational research. While we acknowledge that exact modeling of Barnes maze behavior would require a more sophisticated model (as discussed in the first round of review), our intention to use the reversal Barnes maze paradigm is to suggest a presumable memory modification learning in a non-fear conditioning paradigm. We also discussed whether similar deficits in memory modification could be observed across two behavioral tasks.

(4) Reviewer # mentioned that the change in pattern of behavior only shows up in the older mice questioning the clinical relevance of early detection. I do think this is a valid point and maybe should be addressed. There does seem to be a bit of a bump in the controls on day 23 that doesn't appear in the 6-month group. Perhaps this was initially a spontaneous recovery test indicated by the dotted vertical line? This vertical line does not appear to be defined in the figure 1 legend, nor in figures 2 and 3.

We would like to emphasize that the *App^NL-G-F^* knock-in mouse is widely considered a model of early-stage AD, characterized by Aβ accumulation with little to no neurofibrillary tangle pathology or neuronal loss (see Introduction). By examining different ages, we can assess the contribution of both the amount and duration of Aβ accumulation as well as age-related factors. Modeling the deficit in the memory modification process in the older *App^NL-G-F^* knock-in mice, we suggested a diverged internal state in early-stage AD in older age, and this does not diminish the relevance of the model for studying early cognitive changes in AD.

We would also like to clarify again that the x-axis in the figure is “Trial,” not “Day.” The vertical dashed lines in these figures indicate phase boundaries, and they were defined in the figure legend: in Figure 1C, “The vertical dashed lines separate the phases.”; in Figure 2B, “The dashed vertical line separates the extinction 1 and extinction 2 phases.”; in Figure 3, “The vertical dashed lines indicate the boundaries of phases.”

(5) Are the examples in figure 3 good examples? The example for the 12-month-old control shows a substantial increase in weights for the context during test 3, but not for the CS. Yet in the bar plots in Figure 4 G and H, this pattern seems to be different. The weights for the context appear to substantially drop in the "after extinction" period as compared to the "extinction" period. It's hard to tell the change from "extinction" to "after extinction" for the CS weights (the authors change the y-axis for the CS weights but not for the context weights from panels G to H).

We would like to clarify that in Figure 3C, the increase in weights for context is not presented during test 3 (trial 36), noted by the reviewer; rather, it is the unsignaled shock phase (trial 35).

We assumed that the reviewer might misunderstand that the labels on the left in Figure 4, “Acquisition”, “Extinction”, and “After extinction”, indicate the time point. However, the data shown in Figure 4C to 4H are all from the same time point: test 3 (trial 36). The grouping reflects the classification of latent causes based on the trial in which they were inferred. In addition, for Figures 4G and 4H, the y‐axis limits were not set identically because the data range for “Sum of w_CS_” varied. This was done to ensure the visibility of all data points. In Figure 4, each dot represents one animal. Take Figure 3D as an example. The point in Figure 4G is the sum of *w*_3_ and *w*_4_ in trial 36, and the point in Figure 4H is *w_5_* in trial 36, note that the subscript numerals indicate latent cause index. We hope this addresses the reviewer’s question about the difference between the two figures.

The following is the authors’ response to the original reviews

**Reviewer #1 (Public review):**
Summary:The authors show certain memory deficits in a mouse knock-in model of Alzheimer's Disease (AD). They show that the observed memory deficits can be explained by a computational model, the latent cause model of associative memory. The memory tasks used include the fear memory task (CFC) and the 'reverse' Barnes maze. Research on AD is important given its known huge societal burden. Likewise, better characterization of the behavioral phenotypes of genetic mouse models of AD is also imperative to advance our understanding of the disease using these models. In this light, I applaud the authors' efforts.Strengths:(1) Combining computational modelling with animal behavior in genetic knock-in mouse lines is a promising approach, which will be beneficial to the field and potentially explain any discrepancies in results across studies as well as provide new predictions for future work.(2) The authors' usage of multiple tasks and multiple ages is also important to ensure generalization across memory tasks and 'modelling' of the progression of the disease.Weaknesses:[#1] (1) I have some concerns regarding the interpretation of the behavioral results. Since the computational model then rests on the authors' interpretation of the behavioral results, it, in turn, makes judging the model's explanatory power difficult as well. For the CFC data, why do knock-in mice have stronger memory in test 1 (Figure 2C)? Does this mean the knock-in mice have better memory at this time point? Is this explained by the latent cause model? Are there some compensatory changes in these mice leading to better memory? The authors use a discrimination index across tests to infer a deficit in re-instatement, but this indicates a relative deficit in re-instatement from memory strength in test 1. The interpretation of these differential DIs is not straightforward. This is evident when test 1 is compared with test 2, i.e., the time point after extinction, which also shows a significant difference across groups, Figure 2F, in the same direction as the re-instatement. A clarification of all these points will help strengthen the authors' case.

We appreciate the reviewer for the critical comments. According to the latent cause framework, the strength of the memory is influenced by at least 2 parameters: associative weight between CS and US given a latent cause and posterior probability of the latent cause. The modeling results showed that a higher posterior probability of acquisition latent cause, but not higher associative weight, drove the higher test 1 CR in *App^NL-G-F^* mice (Results and Discussion; Figure 4 – figure supplement 3B, 3C). In terms of posterior, we agree that *App^NL-G-F^* mice have strong fear memory. On the other hand, this suggests that *App^NL-G-F^* mice exhibited a tendency toward overgeneralization, favoring modification of old memories, which adversely affected the ability to retain competing memories. The strong memory in test 1 would be a compensatory effect of overgeneralization.

To estimate the magnitude of reinstatement, at least, one would have to compare CRs between test 2 (extinction) and test 3 (reinstatement), as well as those between test 1 (acquisition) and test 3. These comparisons represent the extent to which the memory at the reinstatement is far from that in the extinction, and close to that in the acquisition. Since discrimination index (DI) has been widely used as a normalized measure to evaluate the extent to which the system can distinguish between two conditions, we applied DI consistently to behavioral and simulated data in the reinstatement experiment, and the behavioral data in the reversal Barnes maze experiment, allowing us to evaluate the discriminability of an agent in these experiments. In addition, we used DI to examine its correlation with estimated parameters, enabling us to explore how individual discriminability may relate to the internal state. We have already discussed the differences in DI between test 3 and test 1, as well as CR in test 1 between control and *App^NL-G-F^* in the manuscript and further elaborated on this point in Line 232, 745-748.

[#2] (2) I have some concerns regarding the interpretation of the Barnes maze data as well, where there already seems to be a deficit in the memory at probe test 1 (Figure 6C). Given that there is already a deficit in memory, would not a more parsimonious explanation of the data be that general memory function in this task is impacted in these mice, rather than the authors' preferred interpretation? How does this memory weakening fit with the CFC data showing stronger memories at test 1? While I applaud the authors for using multiple memory tasks, I am left wondering if the authors tried fitting the latent cause model to the Barnes maze data as well.

While we agree that the deficits shown in probe test 1 may imply impaired memory function in *App^NL-G-F^* mice in this task, it would be difficult to explain this solely in terms of impairments in general memory function. The learning curve and the daily strategy changes suggested that *App^NL-G-F^* mice would have virtually intact learning ability in the initial training phase (Figure 6B, 6F, Figure 6 – figure supplement 1 and 3). For the correspondence relationship between the reinstatement and the reversal Barnes maze learning from the aspect of memory modification process, please also see our reply to comment #24. We have explained why we did not fit the latent cause model to the Barnes maze data in the provisional response.

[#3] (3) Since the authors use the behavioral data for each animal to fit the model, it is important to validate that the fits for the control vs. experimental groups are similar to the model (i.e., no significant differences in residuals). If that is the case, one can compare the differences in model results across groups (Figures 4 and 5). Some further estimates of the performance of the model across groups would help.

We have added the residual (i.e., observed CR minus simulated CR) in Figure 3 – figure supplement 1D and 1E. The fit was similar between control and *App^NL-G-F^* mice groups in the test trials, except test 3 in the 12-month-old group. The residual was significantly higher in the 12-month-old control mice than *App^NL-G-F^* mice, suggesting the model underestimated the reinstatement in the control, yet the DI calculated from the simulated CR replicates the behavioral data (Figure 3 – figure supplement 1A to 1C). These results suggest that the latent cause model fits our data with little systematic bias such as an overestimation of CR for the control group in the reinstatement, supporting the validity of the comparisons in estimated parameters between groups. These results and discussion have been added in the manuscript Line 269-276.

One may notice that the latent cause model overestimated the CR in acquisition trials in all groups in Figure 3 – figure supplement 1D and 1E. We have discussed this point in the reply to comment #26, 34 questioned by reviewer 3.

[#4] (4) Is there an alternative model the authors considered, which was outweighed in terms of prediction by this model?

Yes, we have further evaluated two alternative models: the Rescorla-Wagner (RW; Rescorla & Wagner, 1972) model and the latent state model (LSM; Cochran & Cisler, 2019). The RW model serves as a baseline, given its known limitations in explaining fear return after extinction. The LSM is another contemporary model that shares several concepts with the latent cause model (LCM) such as building upon the RW model, assuming a latent variable inferred by Bayes’ rule, and involving a ruminative update for memory modification. We evaluated the three models in terms of the prediction accuracy and reproducibility of key behavioral features. Please refer to the Appendix 1 for detailed methods and results for these two models.

As expected, the RW model fit well to the data till the end of extinction but failed to reproduce reinstatement (Appendix 1 – figure 1A to 1D). Due to a large prediction error in test 3, few samples met the acceptance criteria we set (Appendix 1 – figure 2 and 3A). Conversely, the LSM reproduced reinstatement, as well as gradual learning in acquisition and extinction phases, particularly in the 12-month-old control (Appendix 1 – figure 1G). The number of accepted samples in the LSM was higher than in the RW model but generally lower than in the LCM (Appendix 1 – figure 2). The sum of prediction errors over all trials in the LSM was comparable to that in the LCM in the 6-month-old group (Appendix 1 – figure 4A), it was significantly lower in the 12-month-old group (Appendix 1 – figure 4B). Especially the LSM generated smaller prediction errors during the acquisition trials than in the LCM, suggesting that the LSM might be better at explaining the behaviors of acquisition (Appendix 1 – figure 4A and 4B; but see the reply for comment #34). While the LSM generated smaller prediction errors than the LCM in test 2 of the control group, it failed to replicate the observed DIs, a critical behavioral phenotype difference between control and *App^NL-G-F^* mice (Appendix 1 – figure 6A to 6C; cf. Figure 2F to 2H, Figure 3 – figure supplement 1A to 1C).

Thus, although each model could capture different aspects of reinstatement, standing on the LCM to explain the reinstatement better aligns with our purpose. It should also be noted that we did not explore all parameter spaces of the LSM, hence we cannot rule out the possibility that alternative parameter sets could provide a better fit and explain the memory modification process well. A more comprehensive parameter search in the LSM may be a valuable direction for future research.

[#5] One concern here is also parameter overfitting. Did the authors try leaving out some data (trials/mice) and predicting their responses based on the fit derived from the training data?

Following the reviewer’s suggestion, we confirmed if overfitting occurred using all trials to estimate parameters. Estimating parameters while actually leaving out trials would disorder the time lapse across trials, and thereby the prior of latent causes in each trial. Instead, we removed the constraint of prediction error by setting the error threshold to 1 for certain trials to virtually leave these trials out. We treated these trials as a virtual “training” dataset, while the rest of the trials were a “test” dataset. For the median CR data of each group (Figure 3), we estimated parameters under 6 conditions with unique training and test trials, then evaluated the prediction error for the training and test trials. Note that training and test trials were arbitrarily decided. Also, the error threshold for the acquisition trial was set to 1 as described in Materials and Methods, which we have further discussed the reason in the reply to comment #34 and treated acquisition trials separately from the test trials. We expect that the contribution of the data from the acquisition and test trials for parameter estimation could be discounted compared to those from the training trials with the constraint, and if overfitting occurred, the prediction error in the test data would be worse than that in the training trials.

Author response image 1A to 1F showed the simulated and observed CR under each condition, where acquisition trials were in light-shaded areas, test trials were in dark-shaded areas, and the rest of the trials were training trials. Author response image 1G showed mean squared prediction error across the acquisition, training and test trials under each condition. The dashed gray line showed the mean squared prediction error of training trials in Figure 3 as a baseline.

In conditions i and ii, where two or four trials in the extinction were used for training (Author response image 1A and 1B), the prediction error was generally higher in test trials than in training trials. In conditions iii and iv where ten trials in the extinction were used for training (Author response image 1C and 1D), the difference in prediction error between testing and training trials became smaller. These results suggest that providing more extinction trial data would reduce overfitting. In condition v (Author response image 1E), the results showed that using trials until extinction can predict reinstatement in control mice but not *App^NL-G-F^* mice. Similarly, in condition vi (Author response image 1F), where test phase trials were left out, the prediction error differences were greater in *App^NL-G-F^* mice. These results suggest that the test trials should be used for the parameter estimation to minimize prediction error for all groups. Overall, this analysis suggests that using all trials would reduce prediction error with few overfitting.

**Author response image 3. sa4fig3:** Leaving trials out in parameter estimation in the latent cause model. (A – F) The observed CR (colored line) is the median freezing rate during the CS presentation over the mice within each group, which is the same as that in Figure 3. The colors indicate different groups: orange represents 6-month-old control, light blue represents 6-month-old *App^NL-G-F^* mice, pink represents 12-month-old control, and dark blue represents 12-month-old *App^NL-G-F^* mice. Under six different leave-out conditions (i – vi), parameters were estimated and used for generating simulated CR (gray line). In each condition, trials were categorized as acquisition (light-shaded area), training data (white area), and test data (dark-shaded area) based on the error threshold during parameter estimation. Only the error threshold of the test data trial was different from the original method (see Material and Method) and set to 1. In conditions i to vi, the number of test data trials is 27, 25, 19, and 19 in extinction phases. In condition v, the number of test data trials is 2 (trials 35 and 36). In condition vi, test data trials were the 3 test phases (trials 4, 34, and 36). (G) Each subplot shows the mean squared prediction error for the test data trial (gray circles), training data trial (white squares), and acquisition trial (gray triangles) in each group. The left y-axis corresponds to data from test and training trials, and the right y-axis corresponds to data from acquisition trials. The dashed line indicates the results calculated from Figure 3 as a baseline.

**Reviewer #1 (Recommendations for the authors):**
Minor:[#6] (1) I would like the authors to further clarify why 'explaining' the reinstatement deficit in the AD mouse model is important in working towards the understanding of AD i.e., which aspect of AD this could explain etc.

In this study, we utilized the reinstatement paradigm with the latent cause model as an internal model to illustrate how estimating internal states can improve understanding of cognitive alteration associated with extensive Aβ accumulation in the brain. Our findings suggest that misclassification in the memory modification process, manifesting as overgeneralization and overdifferentiation, underlies the memory deficit in the *App^NL-G-F^* knock-in model mice.

The parameters in the internal model associated with AD pathology (e.g., *α* and *σ_x_*^2^ in this study) can be viewed as computational phenotypes, filling the explanatory gap between neurobiological abnormalities and cognitive dysfunction in AD. This would advance the understanding of cognitive symptoms in the early stages of AD beyond conventional behavioral endpoints alone.

We further propose that altered internal states in *App^NL-G-F^* knock-in mice may underlie a wide range of memory-related symptoms in AD as we observed that *App^NL-G-F^* knock-in mice failed to retain competing memories in the reversal Barnes maze task. We speculate on how overgeneralization and overdifferentiation may explain some AD symptoms in the manuscript:

- Line 565-569: overgeneralization may explain deficits in discriminating highly similar visual stimuli reported in early-stage AD patients as they misclassify the lure as previously learned object

- Line 576-579: overdifferentiation may explain impaired ability to transfer previously learned association rules in early-stage AD patients as they misclassify them as separated knowledge.

- Line 579-582: overdifferentiation may explain delusions in AD patients as an extended latent cause model could simulate the emergence of delusional thinking

We provide one more example here that overgeneralization may explain that early-stage AD patients are more susceptible to proactive interference than cognitively normal elders in semantic memory tests (Curiel Cid et al., 2024; Loewenstein et al., 2015, 2016; Valles-Salgado et al., 2024), as they are more likely to infer previously learned material. Lastly, we expect that explaining memory-related symptoms within a unified framework may facilitate future hypothesis generation and contribute to the development of strategies for detecting the earliest cognitive alteration in AD.

[#7] (2) The authors state in the abstract/introduction that such computational modelling could be most beneficial for the early detection of memory disorders. The deficits observed here are pronounced in the older animals. It will help to further clarify if these older animals model the early stages of the disease. Do the authors expect severe deficits in this mouse model at even later time points?

The early stage of the disease is marked by abnormal biomarkers associated with Aβ accumulation and neuroinflammation, while cognitive symptoms are mild or absent. This stage can persist for several years during which the level of Aβ may reach a plateau. As the disease progresses, tau pathology and neurodegeneration emerge and drive the transition into the late stage and the onset of dementia. The *App^NL-G-F^* knock-in mice recapitulate the features present in the early stage (Saito et al., 2014), where extensive Aβ accumulation and neuroinflammation worsen along with ages (Figure 2 – figure supplement 1). Since *App^NL-G-F^* knock-in mice are central to Aβ pathology without tauopathy and neurodegeneration, it should be noted that it does not represent the full spectrum of the disease even at advanced ages. Therefore, older animals still model the early stages of the diseases and are suitable to study the long-term effect of Aβ accumulation and neuroinflammation.

The age tested in previous reports using *App^NL-G-F^* mice spanned a wide range from 2 months old to 24 months old. Different behavioral tasks have varied sensitivity but overall suggest the dysfunction worsens with aging (Bellio et al., 2024; Mehla et al., 2019; Sakakibara et al., 2018). We have tested the reinstatement experiment with 17-month-old *App^NL-G-F^* mice before (Author response image 2). They showed more advanced deficits with the same trends observed in 12-month-old *App^NL-G-F^* mice, but their freezing rates were overall at a lower level. There is a concern that possible hearing loss may affect the results and interpretation, therefore we decided to focus on 12-month-old data.

**Author response image 4. sa4fig4:** Freezing rate across reinstatement paradigm in the 17-month-old *App^NL-G-F^* mice. Dashed and solid lines indicate the median freezing rate over 34 mice before (preCS) and during (CS) tone presentation, respectively. Red, blue, and yellow backgrounds represent acquisition, extinction, and unsignaled shock in Figure 2A. The dashed vertical line separates the extinction 1 and extinction 2 phases.

[#8] (3) There are quite a few 'marginal' p-values in the paper at p>0.05 but near it. Should we accept them all as statistically significant? The authors need to clarify if all the experimental groups are sufficiently powered.

For our study, we decided *a priori* that *p* < 0.05 would be considered statistically significant, as described in the Materials and Methods. Therefore, in our Results, we did not consider these marginal values as statistically significant but reported the trend, as they may indicate substantive significance.

We described our power analysis method in the manuscript Line 897-898 and have provided the results in Tables S21 and S22.

[#9] (4) The authors emphasize here that such computational modelling enables us to study the underlying 'reasoning' of the patient (in the abstract and introduction), I do not see how this is the case. The model states that there is a latent i.e. another underlying variable that was not previously considered.

Our use of the term “reasoning” was to distinguish the internal model, which describes how an agent makes sense of the world, from other generative models implemented for biomarker and disease progression prediction. However, we agree that using “reasoning” may be misleading and imprecise, so to reduce ambiguity we have removed this word in our manuscript Line 27: Nonetheless, internal models of the patient remain underexplored in AD; Line 85: However, previous approaches did not suppose an internal model of the world to predict future from current observation given prior knowledge.

[#10] (5) The authors combine knock-in mice with controls to compute correlations of parameters of the model with behavior of animals (e.g. Figure 4B and Figure 5B). They run the risk of spurious correlations due to differences across groups, which they have indeed shown to exist (Figure 4A and 5A). It would help to show within-group correlations between DI and parameter fit, at least for the control group (which has a large spread of data).

We agree that genotype (control, *App^NL-G-F^*) could be a confounder between the estimated parameters and DI, thereby generating spurious correlations. To address this concern, we have provided withingroup correlation in Figure 4 – figure supplement 2 for the 12-month-old group and Figure 5 – figure supplement 2 for the 6-month-old group.

In the 12-month-old group, the significant positive correlation between *σ_x_*^2^ and DI remained in both control and *App^NL-G-F^* mice even if we adjusted the genotype effect, suggesting that it is very unlikely that the correlations in Figure 4B are due to the genotype-related confounding. On the other hand, the positive correlation between *α* and DI was found to be significant in the control mice but not in the *App^NL-G-F^* mice. Most of *α* were distributed around the lower bound in *App^NL-G-F^* mice, which possibly reduced the variance and correlation coefficient. These results support our original conclusion that *α* and *σ_x_*^2^ are parameters associated with a lower magnitude of reinstatement in aged *App^NL-G-F^* mice.

In the 6-month-old group, the correlations shown in Figure 5B were not preserved within subgroups, suggesting genotype would be a confounder for *α*, *σ_x_*^2^, and DI. We recognized that significant correlations in Figure 5B may arise from group differences, increased sample size, or greater variance after combining control and *App^NL-G-F^* mice.

Therefore, we concluded that *α* and *σ_x_*^2^ are associated with the magnitude of reinstatement but modulated by the genotype effect depending on the age.

We have added interpretations of within-group correlation in the manuscript Line 307-308, 375-378.

[#11] (6) It is unclear to me why overgeneralization of internal states will lead to the animals having trouble recalling a memory. Would this not lead to overgeneralization of memory recall instead?

We assume that the reviewer is referring to “overgeneralization of internal states,” a case in which the animal’s internal state remained the same regardless of the observation, thereby leading to “overgeneralization of memory recall.” We agree that this could be one possible situation and appears less problematic than the case in which this memory is no longer retrievable.

However, in our manuscript, we did not deal with the case of “overgeneralization of internal states”. Rather, our findings illustrated how the memory modification process falls into overgeneralization or overdifferentiation and how it adversely affects the retention of competing memories, thereby causing *App^NL-G-F^* mice to have trouble recalling the same memory as the control mice.

According to the latent cause model, retrieval failure is explained by a mismatch of internal states, namely when an agent perceives that the current cue does not match a previously experienced one, the old latent cause is less likely to be inferred due to its low likelihood (Gershman et al., 2017). For example, if a mouse exhibited higher CR in test 2, it would be interpreted as a successful fear memory retrieval due to overgeneralization of the fear memory. However, it reflects a failure of extinction memory retrieval due to the mismatch between the internal states at extinction and test 2. This is an example that overgeneralization of memory induces the failure of memory retrieval.

On the other hand, *App^NL-G-F^* mice exhibited higher CR in test 1, which is conventionally interpreted as a successful fear memory retrieval. When estimating their internal states, they would infer that their observation in test 1 well matches those under the acquisition latent causes, that is the overgeneralization of fear memory as shown by a higher posterior probability in acquisition latent causes in test 1 (Figure 4 – figure supplement 3). This is an example that over-generalization of memory does not always induce retrieval failure as we explained in the reply to comment #1.

**Reviewer #2 (Public review):**
Summary:This manuscript proposes that the use of a latent cause model for the assessment of memory-based tasks may provide improved early detection of Alzheimer's Disease as well as more differentiated mapping of behavior to underlying causes. To test the validity of this model, the authors use a previously described knock-in mouse model of AD and subject the mice to several behaviors to determine whether the latent cause model may provide informative predictions regarding changes in the observed behaviors. They include a well-established fear learning paradigm in which distinct memories are believed to compete for control of behavior. More specifically, it's been observed that animals undergoing fear learning and subsequent fear extinction develop two separate memories for the acquisition phase and the extinction phase, such that the extinction does not simply 'erase' the previously acquired memory. Many models of learning require the addition of a separate context or state to be added during the extinction phase and are typically modeled by assuming the existence of a new state at the time of extinction. The Niv research group, Gershman et al. 2017, have shown that the use of a latent cause model applied to this behavior can elegantly predict the formation of latent states based on a Bayesian approach, and that these latent states can facilitate the persistence of the acquisition and extinction memory independently. The authors of this manuscript leverage this approach to test whether deficits in the production of the internal states, or the inference and learning of those states, may be disrupted in knock-in mice that show both a build-up of amyloid-beta plaques and a deterioration in memory as the mice age.Strengths:I think the authors' proposal to leverage the latent cause model and test whether it can lead to improved assessments in an animal model of AD is a promising approach for bridging the gap between clinical and basic research. The authors use a promising mouse model and apply this to a paradigm in which the behavior and neurobiology are relatively well understood - an ideal situation for assessing how a disease state may impact both the neurobiology and behavior. The latent cause model has the potential to better connect observed behavior to underlying causes and may pave a road for improved mapping of changes in behavior to neurobiological mechanisms in diseases such as AD.Weaknesses:I have several substantial concerns which I've detailed below. These include important details on how the behavior was analyzed, how the model was used to assess the behavior, and the interpretations that have been made based on the model.[#12] (1) There is substantial data to suggest that during fear learning in mice separate memories develop for the acquisition and extinction phases, with the acquisition memory becoming more strongly retrieved during spontaneous recovery and reinstatement. The Gershman paper, cited by the authors, shows how the latent causal model can predict this shift in latent states by allowing for the priors to decay over time, thereby increasing the posterior of the acquisition memory at the time of spontaneous recovery. In this manuscript, the authors suggest a similar mechanism of action for reinstatement, yet the model does not appear to return to the acquisition memory state after reinstatement, at least based on the examples shown in Figures 1 and 3. Rather, the model appears to mainly modify the weights in the most recent state, putatively the 'extinction state', during reinstatement. Of course, the authors must rely on how the model fits the data, but this seems problematic based on prior research indicating that reinstatement is most likely due to the reactivation of the acquisition memory. This may call into question whether the model is successfully modeling the underlying processes or states that lead to behavior and whether this is a valid approach for AD.

We thank the reviewer for insightful comments.

We agree that, as demonstrated in Gershman et al. (2017), the latent cause model accounts for spontaneous recovery via the inference of new latent causes during extinction and the temporal compression property provided by the prior. Moreover, it was also demonstrated that even a relatively low posterior can drive behavioral expression if the weight in the acquisition latent cause is preserved. For example, when the interval between retrieval and extinction was long enough that acquisition latent cause was not dominant during extinction, spontaneous recovery was observed despite the posterior probability of acquisition latent cause (C1) remaining below 0.1 in Figure 11D of Gershman et al. (2017).

In our study, a high response in test 3 (reinstatement) is explained by both acquisition and extinction latent cause. The former preserves the associative weight of the initial fear memory, while the latter has w_context_ learned in the unsignaled shock phase. These positive w were weighted by their posterior probability and together contributed to increased expected shock in test 3. Though the posterior probability of acquisition latent cause was lower than extinction latent cause in test 3 due to time passage, this would be a parallel instance mentioned above. To clarify their contributions to reinstatement, we have conducted additional simulations and the discussion in reply to the reviewer’s next comment (see the reply to comment #13).

We recognize that our results might appear to deviate from the notion that reinstatement results from the strong reactivation of acquisition memory, where one would expect a high posterior probability of the acquisition latent cause. However, we would like to emphasize that the return of fear emerges from the interplay of competing memories. Previous studies have shown that contextual or cued fear reinstatement involves a neural activity switch back to fear state in the medial prefrontal cortex (mPFC), including the prelimbic cortex and infralimbic cortex, and the amygdala, including ventral intercalated amygdala neurons (ITCv), medial subdivision of central nucleus of the amygdala (CeM), and the basolateral amygdala (BLA) (Giustino et al., 2019; Hitora-Imamura et al., 2015; Zaki et al., 2022). We speculate that such transition is parallel to the internal states change in the latent cause model in terms of posterior probability and associative weight change.

Optogenetic manipulation experiments have further revealed how fear and extinction engrams contribute to extinction retrieval and reinstatement. For instance, Gu et al. (2022) used a cued fear conditioning paradigm and found that inhibition of extinction engrams in the BLA, ventral hippocampus (vHPC), and mPFC after extinction learning artificially increased freezing to the tone cue. Similar results were observed in contextual fear conditioning, where silencing extinction engrams in the hippocampus dentate gyrus (DG) impaired extinction retrieval (Lacagnina et al., 2019). These results suggest that the weakening extinction memory can induce a return of fear response even without a reminder shock. On the other hand, Zaki et al. (2022) showed that inhibition of fear engrams in the BLA, DG, or hippocampus CA1 attenuated contextual fear reinstatement. However, they also reported that stimulation of these fear engrams was not sufficient to induce reinstatement, suggesting these fear engram only partially account for reinstatement.

In summary, reinstatement likely results from bidirectional changes in the fear and extinction circuits, supporting our interpretation that both acquisition and extinction latent causes contribute to the reinstatement. Although it remains unclear whether these memory engrams represent latent causes, one possible interpretation is that *w_context_* update in extinction latent causes during unsignaled shock indicates weakening of the extinction memory, while preservation of *w* in acquisition latent causes and their posterior probability suggests reactivation of previous fear memory.

[#13] (2) As stated by the authors in the introduction, the advantage of the fear learning approach is that the memory is modified across the acquisition-extinction-reinstatement phases. Although perhaps not explicitly stated by the authors, the post-reinstatement test (test 3) is the crucial test for whether there is reactivation of a previously stored memory, with the general argument being that the reinvigorated response to the CS can't simply be explained by relearning the CS-US pairing, because re-exposure the US alone leads to increase response to the CS at test. Of course there are several explanations for why this may occur, particularly when also considering the context as a stimulus. This is what I understood to be the justification for the use of a model, such as the latent cause model, that may better capture and compare these possibilities within a single framework. As such, it is critical to look at the level of responding to both the context alone and to the CS. It appears that the authors only look at the percent freezing during the CS, and it is not clear whether this is due to the contextual US learning during the US re-exposure or to increased response to the CS - presumably caused by reactivation of the acquisition memory. For example, the instance of the model shown in Figure 1 indicates that the 'extinction state', or state z6, develops a strong weight for the context during the reinstatement phase of presenting the shock alone. This state then leads to increased freezing during the final CS probe test as shown in the figure. By not comparing the difference in the evoked freezing CR at the test (ITI vs CS period), the purpose of the reinstatement test is lost in the sense of whether a previous memory was reactivated - was the response to the CS restored above and beyond the freezing to the context? I think the authors must somehow incorporate these different phases (CS vs ITI) into their model, particularly since this type of memory retrieval that depends on assessing latent states is specifically why the authors justified using the latent causal model.

To clarify the contribution of context, we have provided preCS freezing rate across trials in Figure 2 – figure supplement 2. As the reviewer pointed out, the preCS freezing rate did not remain at the same level across trials, especially within the 12-month-old control and *App^NL-G-F^* group (Figure 2 – figure supplement 2A and 2B), suggesting the effect context. A paired samples *t*-test comparing preCS freezing (Figure 2 – figure supplement 2E) and CS freezing (Figure 2E) in test 3 revealed significant differences in all groups: 6-month-old control, *t*(23) = -6.344, *p* < 0.001, *d* = -1.295; 6-month-old *App^NL-G-F^*, *t*(24) = -4.679, *p* < 0.001, *d* = -0.936; 12-month-old control, *t*(23) = -4.512, *p* < 0.001, *d* = 0.921; 12-month-old *App^NL-G-F^*, *t*(24) = -2.408, *p* = 0.024, *d* = -0.482. These results indicate that the response to CS was above and beyond the response to context only. We also compared the change in freezing rate (CS freezing rate minus preCS freezing rate) in test 2 and test 3 to examine the net response to the tone. The significant difference was found in the control group, but not in the *App^NL-G-F^* group (Author response image 3). The increased net response to the tone in the control group suggested that the reinstatement was partially driven by reactivation of acquisition memory, not solely by the contextual US learning during the unsignaled shock phase. We have added these results and discussion in the manuscript Line 220-231.

**Author response image 5. sa4fig5:** Net freezing rate in test 2 and test 3. Net freezing rate is defined as the CS freezing rate (i.e., freezing rate during 1 min CS presentation) minus the preCS freezing rate (i.e., 1 min before CS presentation). The dashed horizontal line indicates no freezing rate change from the preCS period to the CS presentation. **p* < 0.05 by paired-sample Student’s *t*-test, and the alternative hypothesis specifies that test 2 freezing rate change is less than test 3. Colors indicate different groups: orange represents 6-month-old control (*n* = 24), light blue represents 6-month-old *App^NL-G-F^* mice (*n* = 25), pink represents 12-month-old control (*n* = 24), and dark blue represents 12-month-old *App^NL-G-F^* mice (*n* = 25). Each black dot represents one animal. Statistical results were as follows: *t*(23) = -1.927, *p* = 0.033, Cohen’s *d* = -0.393 in 6-month-old control; *t*(24) = -1.534, *p* = 0.069, Cohen’s *d* = -0.307 in 6-month-old *App^NL-G-F^*; *t*(23) = -1.775, *p* = 0.045, Cohen’s *d* = -0.362 in 12-month-old control; *t*(24) = 0.86, *p* = 0.801, Cohen’s *d* = 0.172 in 12-monthold *App^NL-G-F^*.

According to the latent cause model, if the reinstatement is merely induced by an association between the context and the US in the unsignaled shock phase, the CR given context only and that given context and CS in test 3 should be equal. However, the simulation conducted for each mouse using their estimated parameters confirmed that this was not the case in this study. The results showed that simulated CR was significantly higher in the context+CS condition than in the context only condition (Author response image 4). This trend is consistent with the behavioral results we mentioned above.

**Author response image 6. sa4fig6:** Simulation of context effect in test 3. Estimated parameter sets of each sample were used to run the simulation that only context or context with CS was present in test 3 (trial 36). The data are shown as median with interquartile range, where white bars with colored lines represent CR for context only and colored bars represent CR for context with CS. Colors indicate different groups: orange represents 6-month-old control (*n* = 15), light blue represents 6-month-old *App^NL-G-F^* mice (*n* = 12), pink represents 12-month-old control (*n* = 20), and dark blue represents 12-month-old *App^NL-G-F^* mice (*n* = 18). Each black dot represents one animal. ***p* < 0.01, and ****p* < 0.001 by Wilcoxon signed-rank test comparing context only and context + CS in each group, and the alternative hypothesis specifies that CR in context is not equal to CR in context with CS. Statistical results were as follows: *W* = 15, *p* = 0.008, effect size *r* = -0.66 in 6-month-old control; *W* = 0, *p* < 0.001, effect size *r* = -0.88 in 6-month-old *App^NL-G-F^*; *W* = 25, *p* = 0.002, effect size *r* = -0.67 in 12-month-old control; *W* = 9, *p* = 0.002 , effect size *r* = -0.75 in 12-month-old *App^NL-G-F^*.

[#14] (3) This is related to the second point above. If the question is about the memory processes underlying memory retrieval at the test following reinstatement, then I would argue that the model parameters that are not involved in testing this hypothesis be fixed prior to the test. Unlike the Gershman paper that the authors cited, the authors fit all parameters for each animal. Perhaps the authors should fit certain parameters on the acquisition and extinction phase, and then leave those parameters fixed for the reinstatement phase. To give a more concrete example, if the hypothesis is that AD mice have deficits in differentiating or retrieving latent states during reinstatement which results in the low response to the CS following reinstatement, then perhaps parameters such as the learning rate should be fixed at this point. The authors state that the 12-month-old AD mice have substantially lower learning rate measures (almost a 20-fold reduction!), which can be clearly seen in the very low weights attributed to the AD mouse in Figure 3D. Based on the example in Figure 3D, it seems that the reduced learning rate in these mice is most likely caused by the failure to respond at test. This is based on comparing the behavior in Figures 3C to 3D. The acquisition and extinction curves appear extremely similar across the two groups. It seems that this lower learning rate may indirectly be causing most of the other effects that the authors highlight, such as the low σx, and the changes to the parameters for the CR. It may even explain the extremely high K. Because the weights are so low, this would presumably lead to extremely low likelihoods in the posterior estimation, which I guess would lead to more latent states being considered as the posterior would be more influenced by the prior.

We thank the reviewer for the suggestion about fitting and fixing certain parameters in different phases.

However, this strategy may not be optimal for our study for the following scientific reasons.

Our primary purpose is to explore internal states in the memory modification process that are associated with the deficit found in *App^NL-G-F^* mice in the reinstatement paradigm. We did not restrict the question to memory retrieval, nor did we have a particular hypothesis such that only a few parameters of interest account for the impaired associative learning or structure learning in *App^NL-G-F^* mice while all other parameters are comparable between groups. We are concerned that restricting questions to memory retrieval at the test is too parsimonious and might lead to misinterpretation of the results. As we explain in reply to comment #5, removing trials in extinction during parameter estimation reduces the model fit performance and runs the risk of overfitting within the individual. Therefore, we estimated all parameters for each animal, with the assumption that the estimated parameter set represents individual internal state (i.e., learning and memory characteristics) and should be fixed within the animal across all trials.

Figure 3 is the parameter estimation and simulation results using the median data of each group as an individual. The estimated parameter value is one of the possible cases in that group to demonstrate how a typical learning curve fits the latent cause model. The reviewer mentioned “20-fold reduction in learning rate” is the comparison of two data points, not the actual comparison between groups. The comparison between control and *App^NL-G-F^* mice in the 12-month-old group for all parameters was provided in Table S7. The Mann-Whitney *U* test did not reveal a significant difference in learning rate (*η*): 12-month-old control (Mdn = 0.09, IQR = 0.23) vs. 12-month-old *App^NL-G-F^* (Mdn = 0.12, IQR = 0.23), *U* = 199, *p* = 0.587.

We agree that lower learning rate could bias the learning toward inferring a new latent cause. However, this tendency may depend on the value of other parameters and varied in different phases in the reinstatement paradigm. Here, we used *α* as an example and demonstrate their interaction in Appendix 2 – table 2 with relatively extreme values: *α* = {1, 3} and *η* = {0.01, 0.5} while the rest of the parameters fixed at the initial guess value.

When *α* = 1, the number of latent causes across phases (K_acq_, K_ext_, K_rem_) remain unchanged and their posterior probability in test 3 were comparable even if *η* increased from 0.01 to 0.5. This is an example that lower *η* does not lead to inferring new latent causes because of low *α*. The effect of low learning rate manifests in test 3 CR due to low *w_context, acq_* and *w_context, ext_*.

When *α* = 3, the number of acquisition latent causes (K_acq_) was higher in the case of *η* = 0.01 than that of *η* = 0.5, showing the effect mentioned by the reviewer. However, test 1 CR is much lower when *η* = 0.01, indicating unsuccessful learning even after inferring a new latent cause. This is none of the cases observed in this study. During extinction phases, the effect of *η* is surpassed by the effect of high *α*, where the number of extinction latent causes (K_ext_) is high and not affected by *η*. After the extinction phases, the effect of *K* kicks in as the total number of latent causes reaches its value (*K* = 33 in this example), especially in the case of *η* = 0.01. A new latent cause is inferred after extinction in the condition of *η* = 0.5, but the CR 3 is still high as the *w_context, acq_* and *w_context, ext_* are high. This is an example that a new latent cause is inferred in spite of higher *η*.

Overall, the learning rate would not have a prominent effect alone throughout the reinstatement paradigm, and it has a joint effect with other parameters. Note that the example here did not cover our estimated results, as the estimated learning rate was not significantly different between control and *App^NL-G-F^* mice (see above). Please refer to the reply to comment #31 for more discussion about the interaction among parameters when the learning rate is fixed. We hope this clarifies the reviewer’s concern.

[#15] (4) Why didn't the authors use the latent causal model on the Barnes maze task? The authors mention in the discussion that different cognitive processes may be at play across the two tasks, yet reversal tasks have been suggested to be solved using latent states to be able to flip between the two different task states. In this way, it seems very fitting to use the latent cause model. Indeed, it may even be a better way to assess changes in σx as there are presumably 12 observable stimuli/locations.

Please refer to our provisional response about the application of the latent cause model to the reversal Barnes maze task. Briefly, it would be difficult to directly apply the latent cause model to the Barnes maze data because this task involves operant learning, and thereby almost all conditions in the latent cause model are not satisfied. Please also see our reply to comment #24 for the discussion of the link between the latent cause model and Barnes maze task.

**Reviewer #2 (Recommendations for the authors):**
[#16] (1) I had a bit of difficulty finding all the details of the model. First, I had to mainly rely on the Gershman 2017 paper to understand the model. Even then, there were certain aspects of the model that were not clear. For instance, it's not quite clear to me when the new internal states are created and how the maximum number of states is determined. After reading the authors' methods and the Gershman paper, it seems that a new internal state is generated at each time point, aka zt, and that the prior for that state decays onwards from alpha. Yet because most 'new' internal states don't ever take on much of a portion of the posterior, most of these states can be ignored. Is that a correct understanding? To state this another way, I interpret the equation on line 129 to indicate that the prior is determined by the power law for all existing internal states and that each new state starts with a value of alpha, yet I don't see the rule for creating a new state, or for iterating k other than that k iterates at each timestep. Yet this seems to not be consistent with the fact that the max number of states K is also a parameter fit. Please clarify this, or point me to where this is better defined.I find this to be an important question for the current paper as it is unclear to me when the states were created. Most notably, in Figure 3, it's important to understand why there's an increase in the posterior of *z5* in the AD 12-month mice at test. Is state *z5* generated at trial 5? If so, the prior would be extremely small by trial 36, making it even more perplexing why *z5* has such a high posterior. If its weights are similar to *z3* and *z4*, and they have been much more active recently, why would *z5* come into play?

We assume that the “new internal state" the reviewer is referring to is the “new latent cause." We would like to clarify that “internal state" in our study refers to all the latent causes at a given time point and observation. As this manuscript is submitted as a Research Advance article in *eLife*, we did not rephrase all the model details. Here, we explain when a new latent cause is created (i.e., the prior probability of a new latent cause is greater than 0) with the example of the 12-month-old group (Figure 3C and 3D).

Suppose that before the start of each trial, an agent inferred the most likely latent cause with maximum posterior, and it inferred *k* latent causes so far. A new latent cause can be inferred at the computation of the prior of latent causes at the beginning of each trial.

In the latent cause model, it follows a distance-dependent Chinese Restaurant Process (CRP; Blei and Frazier, 2011). The prior of each old latent cause is its posterior probability, which is the final count of the EM update before the current. In addition, the prior of old latent causes is sensitive to the time passage so that it exponentially decreases as a forgetting function modulated by *g* (see Figure 2 in Gershman et al., 2017). Simultaneously, the prior of a new cause is assigned *α*. The new latent cause is inferred at this moment. Hence, the prior of latent causes is jointly determined by *α*, *g* and its posterior probability. The maximum number of latent causes *K* is set *a priori* and does not affect the prior while *k* < *K* (see also reply to comment #30 for the discussion of boundary set for *K* and comment #31 for the discussion of the interaction between *α* and *K*). Note that only one new latent cause can be inferred in each trial, and (*k*+1)^th^ latent cause, which has never been inferred so far, is chosen as the new latent cause.

In our manuscript, the subscript number in *zₖ* denotes the order in which they were inferred, not the trial number. In Figures 3C and 3D, *z*_3_ and *z*_4_ were inferred in trials 5 and 6 during extinction; *z*_*5*_ is a new latent cause inferred in trial 36. Therefore, the prior of *z*_5_ is not extremely small compared to *z*_4_ and *z*_3_.

In both control and *App^NL-G-F^* mice in the 12-month-old (Figures 3C and 3D), *z*_3_ is dominant until trial 35. The unsignaled shock at trial 35 generates a large prediction error as only context is presented and followed by the US. This prediction error reduces posterior of z_3_, while increasing the posterior of *z*_4_ and *w*_*context*_ in *z*_3_ and *z*_4_. This decrease of posterior of *z*_3_ is more obvious in the *App^NL-G-F^* than in the control group, prompting them to infer a new latent cause *z*_5_ (Figure 3C and 3D). Although Figure 3C and 3D are illustrative examples as we explained in the reply to comment #14, this interpretation would be plausible as the *App^NL-G-F^* group inferred a significantly larger number of latent causes after the extinction with slightly higher posteriors of them than those in the control group (Figure 4E).

[#17] (2) Related to the above, Are the states **z**_A_ and **z**_B_ defined by the authors to help the reader group the states into acquisition and extinction states, or are they somehow grouped by the model? If the latter is true, I don't understand how this would occur based on the model. If the former, could the authors state that these states were grouped together by the author?

We used **z**_A_ and **z**_B_ annotations to assist with the explanation, so this is not grouped by the model. We have stated this in the manuscript Line 181-182.

[#18] (3) This expands on the third point above. In Figure 3D, internal states *z3*, *z4*, and *z5* appear to be pretty much identical in weights in the App group. It's not clear to me why then the posterior of *z5* would all of a sudden jump up. If I understand correctly, the posterior is the likelihood of the observations given the internal state (presumably this should be similar across *z3*, *z4*, and *z5*), multiplied by the prior of the state. Z3 and Z4 are the dominant inferred states up to state 36. Why would *z5* become more likely if there doesn't appear to be any error? I'm inferring no error because there are little or no changes in weights on trial 36, most prominently no changes in *z3* which is the dominant internal state in step 36. If there's little change in weights, or no errors, shouldn't the prior dominate the calculation of the posterior which would lead to *z3* and *z4* being most prominent at trial 36?

We have explained how *z*_5_ of the 12-month-old *App^NL-G-F^* was inferred in the reply to comment #16. Here, we explain the process underlying the rapid changes of the posterior of *z*_3_, *z*_4_, and *z*_5_ from trial 35 to 36.

During the extinction, the mice inferred *z*_3_ given the CS and the context in the absence of US. In trial 35, they observed the context and the unsignaled shock in the absence of the CS. This reduced the likelihood for the CS under *z*_3_ and thereby the posterior of *z*_3_, while relatively increasing the posterior of *z_4_*. The associative weight between the context and the US , *w*_*context*_, indeed increased in both *z*_3_ and *z*_4_, but *w*_*context*_ of *z*_4_ was updated more than that of *z_3_* due to its higher posterior probability. At the beginning of trial 36, a new latent cause *z*_5_ was inferred with a certain prior (see also the reply for comment #16), and *w*_5_ = *w*_0_, where *w*_0_ is the initial value of weight. After normalizing the prior over latent causes, the emergence of *z*_5_ reduced the prior probability of other latent causes compared to the case where the prior of *z*_5_ is 0. Since the CS was presented while the US was absent in trial 36, the likelihood of the CS and that of the US under *z*_3_, and especially *z*_4_, given the cues and w became lower than the case in which *z*_5_ has not been inferred yet. Consequently, the posterior of *z*_5_ became salient (Figure 3D).

To maintain consistency across panels, we used a uniform y-axis range. However, we acknowledge that this may make it harder to notice the changes of associative weights in Figure 3D. We have provided the subpanel in Figure 3D with a smaller y-axis limit to reveal the weight changes at trial 35 in Author response image 5.

**Author response image 7. sa4fig7:** Magnified view of *w_context_* and *w_CS_* in the last 3 trials in Figure 3D. The graph format is the same as in Figure 3D. The weight for CS (*w_CS_*) and that for context (*w*_*context*_) in each latent cause across trial 34 (test 2), 35 (unsignaled shock), and 36 (test 3) in 12-month-old *App^NL-G-F^* in Figure 3D was magnified in the upper and lower magenta box, respectively.

[#19] (8) In Figure 4B - The figure legend didn't appear to indicate at which time points the DIs are plotted.

We have amended the figure legend to indicate that DI between test 3 and test 1 is plotted.

[#20] (9) Lines 301-303 state that the posterior probabilities of the acquisition internal states in the 12month AD mice were much higher at test 1 and that this resulted in different levels of CR across the control and 12-month App group. This is shown in the Figure 4A supplement, but this is not apparent in Figure 3 panels C and D. Is the example shown in panel D not representative of the group? The CRs across the two examples in Figure 3 C and D look extremely similar at test 1. Furthermore, the posteriors of the internal states look pretty similar across the two groups for the first 4 trials. Both the App and control have substantial posterior probabilities for the acquisition period, I don't see any additional states at test 1. The pattern of states during acquisition looks strikingly similar across the two groups, whereas the weights of the stimuli are considerably different. I think it would help the authors to use an example that better represents what the authors are referring to, or provide data to illustrate the difference. Figure 4C partly shows this, but it's not very clear how strong the posteriors are for the 3rd state in the controls.

Figure 3 serves as an example to explain the internal states in each group (see also the third paragraph in the reply to comment #14). Figure 4C to H showed the results from each sample for between-group comparison in selected features. Therefore, the results of direct comparisons of the parameter values and internal states between genotypes in Figure 3 are not necessarily the same as those in Figure 4. Both examples in Figure 3C and 3D inferred 2 latent causes during the acquisition. In terms of posterior till test 1 (trial 4), the two could be the same. However, such examples were not rare, as the proportion of the mice that inferred 2 latent causes during the acquisition was slightly lower than 50% in the control, and around 90% in the *App^NL-G-F^* mice (Figure 4C). The posterior probability of acquisition latent cause in test 1 showed a similar pattern (Figure 4 – figure supplement 3), with values near 1 in around 50% of the control mice and around 90% of the *App^NL-G-F^* mice.

[#21] (10) Line 320: This is a confusing sentence. I think the authors are saying that because the App group inferred a new state during test 3, this would protect the weights of the 'extinction' state as compared to the controls since the strength of the weight updates depends on the probability of the posterior.

In order to address this, we have revised this sentence to “Such internal states in *App^NL-G-F^* mice would diverge the associative weight update from those in the control mice after extinction.” in the manuscript Line 349-351.

[#22] (11) In lines 517-519 the authors address the difference in generalizing the occurrence of stimuli across the App and control groups. It states that App mice with lower alpha generalized observations to an old cause rather than attributing it as a new state. Going back to statement 3 above, I think it's important to show that the model fit of a reduction in alpha does not go hand-in-hand with a reduction in the learning rates and hence the weights. Again, if the likelihoods are diminished due to the low weights, then the fit of alpha might be reduced as well. To reiterate my point above, if the observations in changes in generalization and differentiation occur because of a reduction in the learning rate, the modeling may not be providing a particularly insightful understanding of AD, other than that poor learning leads to ineffectual generalization and differentiation. Do these findings hold up if the learning rates are more comparable across the control and App group?

These findings were explained on the basis of comparable learning rates between control and *App^NL-G-F^* mice in the 12-month-old group (see the reply to comment #14). In addition, we have conducted simulation for different *α* and *σ_x_*^2^ values under the condition of the fixed learning rate, where overgeneralization and overdifferentaiton still occurred (see the reply to comment #26).

[#23] (12) Lines 391 - 393. This is a confusing sentence. "These results suggest that App NL-G-F mice could successfully form a spatial memory of the target hole, while the memory was less likely to be retrieved by a novel observation such as the absence of the escape box under the target hole at the probe test 1." The App mice show improved behavior across days of approaching the correct hole. Is this statement suggesting that once they've approached the target hole, the lack of the escape box leads to a reduction in the retention of that memory?

We speculated that when the mice observed the absence of the escape box, a certain prediction error would be generated, which may have driven the memory modification. In *App^NL-G-F^* mice, such modification, either overgeneralization or overdifferentiation, could render the memory of the target hole vulnerable; if overgeneralization occurred, the memory would be quickly overwritten as the goal no longer exists in this position in this maze, while if overdifferentiation occurred, a novel memory such that the goal does not exist in the maze different from previous one would be formed. In either case of misclassification, the probability of retrieving the goal position would be reduced. To reduce ambiguity in this sentence, we have revised the description in the manuscript Line 432-434 as follows: “These results suggest that *App^NL-G-F^* mice could successfully form a spatial memory of the target hole, while they did not retrieve the spatial memory of the target hole as strongly as control mice when they observed the absence of the escape box during the probe test.”

[#24] (13) The connection between the results of Barnes maze and the fear learning paradigm is weak. How can changes in overgeneralization due to a reduction in the creation of inferred states and differentiation due to a reduced σx lead to the observations in the Barnes maze experiment?

We extrapolated our interpretation in the reinstatement modeling to behaviors in a different behavioral task, to explore the explanatory power of the latent cause framework formalizing mechanisms of associative learning and memory modification. Here, we explain the results of the reversal Barnes maze paradigm in terms of the latent cause model, while conferring the reinstatement paradigm.

Whilst we acknowledge that fear conditioning and spatial learning are not fully comparable, the reversal Barnes maze paradigm used in our study shares several key learning components with the reinstatement paradigm.

First, associative learning is fundamental in spatial learning (Leising & Blaisdell, 2009; Pearce, 2009). Although we did not make any specific assumptions of what kind of associations were learned in the Barnes maze, performance improvements in learning phases likely reflect trial-and-error updates of these associations involving sensory preconditioning or secondary conditioning. Second, the reversal training phases could resemble the extinction phase in the reinstatement paradigm, challenge previously established memory. In terms of the latent cause model, both the reversal learning phase in the reversal Barnes maze paradigm and the extinction phase in the reinstatement paradigm induce a mismatch of the internal state. This process likely introduces large prediction errors, triggering memory modification to reconcile competing memories.

Under the latent cause framework, we posit that the mice would either infer new memories or modify existing memories for the unexpected observations in the Barnes maze (e.g., changed location or absence of escape box) as in the reinstatement paradigm, but learn a larger number of association rules between stimuli in the maze compared to those in the reinstatement. In the reversal Barnes maze paradigm, the animals would infer that a latent cause generates the stimuli in the maze at certain associative weights in each trial, and would adjust behavior by retaining competing memories.

Both overgeneralization and overdifferentiation could explain the lower exploration time of the target hole in the *App^NL-G-F^* mice in probe test 1. In the case of overgeneralization, the mice would overwrite the existing spatial memory of the target hole with a memory that the escape box is absent. In the case of overdifferentiation, the mice would infer a new memory such that the goal does not exist in the novel field, in addition to the old memory where the goal exists in the previous field. In both cases, the *App^NL-G-F^* mice would not infer that the location of the goal is fixed at a particular point and failed to retain competing spatial memories of the goal, leading to relying on a less precise, non-spatial strategy to solve the task.

Since there is no established way to formalize the Barnes maze learning in the latent cause model, we did not directly apply the latent cause model to the Barnes maze data. Instead, we used the view above to explore common processes in memory modification between the reinstatement and the Barnes maze paradigm.

The above description was added to the manuscript on page 13 (Line 410-414) and page 19-20 (Line 600-602, 626-639).

[#25] (14) In the fear conditioning task, it may be valuable to separate responding to the context and the cue at the time of the final test. The mice can learn about the context during the reinstatement, but there must be an inference to the cue as it's not present during the reinstatement phase. This would provide an opportunity for the model to perhaps access a prior state that was formed during acquisition. This would be more in line with the original proposal by Gershman et al. 2017 with spontaneous recovery.

Please refer to the reply to comment #13 regarding separating the response to context in test 3.

**Reviewer #3 (Public review):**
Summary:This paper seeks to identify underlying mechanisms contributing to memory deficits observed in Alzheimer's disease (AD) mouse models. By understanding these mechanisms, they hope to uncover insights into subtle cognitive changes early in AD to inform interventions for early-stage decline.Strengths:The paper provides a comprehensive exploration of memory deficits in an AD mouse model, covering the early and late stages of the disease. The experimental design was robust, confirming age-dependent increases in Aβ plaque accumulation in the AD model mice and using multiple behavior tasks that collectively highlighted difficulties in maintaining multiple competing memory cues, with deficits most pronounced in older mice.In the fear acquisition, extinction, and reinstatement task, AD model mice exhibited a significantly higher fear response after acquisition compared to controls, as well as a greater drop in fear response during reinstatement. These findings suggest that AD mice struggle to retain the fear memory associated with the conditioned stimulus, with the group differences being more pronounced in the older mice.In the reversal Barnes maze task, the AD model mice displayed a tendency to explore the maze perimeter rather than the two potential target holes, indicating a failure to integrate multiple memory cues into their strategy. This contrasted with the control mice, which used the more confirmatory strategy of focusing on the two target holes. Despite this, the AD mice were quicker to reach the target hole, suggesting that their impairments were specific to memory retrieval rather than basic task performance.The authors strengthened their findings by analyzing their data with a leading computational model, which describes how animals balance competing memories. They found that AD mice showed somewhat of a contradiction: a tendency to both treat trials as more alike than they are (lower α) and similar stimuli as more distinct than they are (lower σx) compared to controls.Weaknesses:While conceptually solid, the model struggles to fit the data and to support the key hypothesis about AD mice's ability to retain competing memories. These issues are evident in Figure 3:[#26] (1) The model misses key trends in the data, including the gradual learning of fear in all groups during acquisition, the absence of a fear response at the start of the experiment, the increase in fear at the start of day 2 of extinction (especially in controls), and the more rapid reinstatement of fear observed in older controls compared to acquisition.

We acknowledge these limitations and explained why they arise in the latent cause model as follows.

a. Absence of a fear response at the start of the experiment and the gradual learning of fear during acquisition

In the latent cause model, the CR is derived from a sigmoidal transformation from the predicted outcome with the assumption that its mapping to behavioral response may be nonlinear (see Equation 10 and section “Conditioned responding” in Gershman et al., 2017).

The magnitude of the unconditioned response (trial 1) is determined by *w*_0_, *θ,* and *λ*. An example was given in Appendix 2 – table 3. In general, a higher *w*_0_ and a lower *θ* produce a higher trial 1 CR when other parameters are fixed. During the acquisition phase, once the expected shock exceeds *θ*, CR rapidly approaches 1, and further increases in expected shock produce few changes in CR. This rapid increase was also evident in the spontaneous recovery simulation (Figure 11) in Gershman et al. (2017). The steepness of this rapid increase is modulated by *λ* such that a higher value produces a shallower slope. This is a characteristic of the latent cause model, assuming CR follows a sigmoid function of expected shock, while the ordinal relationship over CRs is maintained with or without the sigmoid function, as Gershman et al. (2017) mentioned. If one assumes that the CR should be proportional to the expected shock, the model can reproduce the gradual response as a linear combination of **w** and posteriors of latent causes while omitting the sigmoid transformation (Figure 3).

b. Increase in fear at the start of day 2 extinction

This point is partially reproduced by the latent cause model. As shown in Figure 3, trial 24 (the first trial of day 2 extinction) showed an increase in both posterior probability of latent cause retaining fear memory and the simulated CRs in all groups except the 6-month-old control group, though the increase in CR was small due to the sigmoid transformation (see above). This can be explained by the latent cause model as 24 h time lapse between extinction 1 and 2 decreases the prior of the previously inferred latent cause, leading to an increase of those of other latent causes.

Unlike other groups, the 6-month-old control did not exhibit increased observed CR at trial 24

but at trial 25 (Figure 3A). The latent cause model failed to reproduce it, as there was no increase in posterior probability in trial 24 (Figure 3A). This could be partially explained by the low value of *g*, which counteracts the effect of the time interval between days: lower *g* keeps prior of the latent causes at the same level as those in the previous trial. Despite some failures in capturing this effect, our fitting policy was set to optimize prediction among the test trials given our primary purpose of explaining reinstatement.

c. more rapid reinstatement of fear observed in older controls compared to acquisition

We would like to point out that this was replicated by the latent cause model as shown in Figure 3 – figure supplement 1C. The DI between test 3 and test 1 calculated from the simulated CR was significantly higher in 12-month-old control than in *App^NL-G-F^* mice (cf. Figure 2C to E).

[#27] (2) The model attributes the higher fear response in controls during reinstatement to a stronger association with the context from the unsignaled shock phase, rather than to any memory of the conditioned stimulus from acquisition. These issues lead to potential overinterpretation of the model parameters. The differences in α and σx are being used to make claims about cognitive processes (e.g., overgeneralization vs. overdifferentiation), but the model itself does not appear to capture these processes accurately. The authors could benefit from a model that better matches the data and that can capture the retention and recollection of a fear memory across phases.

First, we would like to clarify that the latent cause model explains the reinstatement not only by the extinction latent cause with increased *w*_*context*_ but also the acquisition latent cause with preserved *w_CS_* and *w*_*context*_ (see also reply to comment #13). Second, the latent cause model primarily attributes the higher fear reinstatement in control to a lower number of latent causes inferred after extinction (Figure 4E) and higher *w*_*context*_ in extinction latent cause (Figure 4G). We noted that there was a trend toward significance in the posterior probability of latent causes inferred after extinction (Figure 4E), which in turn influences those of acquisition latent causes. Although the posterior probability of acquisition latent cause appeared trivial and no significance was detected between control and *App^NL-G-F^* mice (Figure 4C), it was suppressed by new latent causes in *App^NL-G-F^* mice (Author response image 6).

This indicates that *App^NL-G-F^* mice retrieved acquisition memory less strongly than control mice. Therefore, we argue that the latent cause model attributed a higher fear response in control during reinstatement not solely to the stronger association with the context but also to CS fear memory from acquisition. Although we tested whether additional models fit the reinstatement data in individual mice, these models did not satisfy our fitting criteria for many mice compared to the latent cause model (see also reply to comment #4 and #28).

**Author response image 8. sa4fig8:** Posterior probability of acquisition, extinction, and after extinction latent causes in test 3. The values within each bar indicate the mean posterior probability of acquisition latent cause (darkest shade), extinction latent cause (medium shade), and latent causes inferred after extinction (lightest shade) in test 3 over mice within genotype. Source data are the same as those used in Figure 4C–E (posterior of z).

Conclusion:Overall, the data support the authors' hypothesis that AD model mice struggle to retain competing memories, with the effect becoming more pronounced with age. While I believe the right computational model could highlight these differences, the current model falls short in doing so.
**Reviewer #3 (Recommendations for the authors):**
[#28] Other computational models may better capture the data. Ideally, I'd look for a model that can capture the gradual learning during acquisition, and, in some mice, the inferring of a new latent cause during extinction, allowing the fear memory to be retained and referenced at the start of day 2 extinction and during later tests.

We have further evaluated another computational model, the latent state model, and compared it with the latent cause model. The simulation of reinstatement and parameter estimation method of the latent state model were described in the Appendix.

The latent state model proposed by Cochran and Cisler (2019) shares several concepts with the latent cause model, and well replicates empirical data under certain conditions. We expect that it can also explain the reinstatement.

Following the same analysis flow for the latent cause model, we estimated the parameters and simulated reinstatement in the latent state model from individual CRs and median of them. In the median freezing rate data of the 12-month-old control mice, the simulated CR replicated the observed CR well and exhibited the ideal features that the reviewer looked for: gradual learning during acquisition and an increased fear at the start of the second-day extinction (Appendix 1 – figure 1G). However, a lot of samples did not fit well to the latent state model. The number of anomalies was generally higher than that in the latent cause model (Appendix 1 – figure 2). Within the accepted samples, the sum of squared prediction error in all trials was significantly lower in the latent state model, which resulted from lower prediction error in the acquisition trials (Appendix 1 – figure 4A and 4B). In the three test trials, the squared prediction error was comparable between the latent state model and the latent cause model except for the test 2 trials in the control group (Appendix 1 – figure 4A and 4B, rightmost panel). On the other hand, almost all accepted samples continued to infer the acquisition latent states during extinction without inferring new states (Appendix 1 – figure 5B and 5E, left panel), which differed from the ideal internal states the reviewer expected. While the latent state model fit performance seems to be better than the latent cause model, the accepted samples cannot reproduce the lower DI between test 3 and test 1 in aged *App^NL-G-F^* mice (Appendix 1 – figure 6C). These results make the latent state model less suitable for our purpose and therefore we decided to stay with the latent cause model. It should also be noted that we did not explore all parameter spaces of the latent state model hence we cannot rule out the possibility that alternative parameter sets could provide a better fit and explain the memory modification process well. A more comprehensive parameter search in the LSM may be a valuable direction for future research.

If you decide not to go with a new model, my preference would be to drop the current modeling. However, if you wish to stay with the current model, I'd like to see justification or acknowledgment of the following:

[#29] (1) Lower bound on alpha of 1: This forces the model to infer new latent causes, but it seems that some mice, especially younger AD mice, might rely more on classical associative learning (e.g., Rescorla-Wagner) rather than inferring new causes.

We acknowledge that the default value set in Gershman et al. (2017) is 0.1, and the constraint we set is a much higher value. However, *α* = 1 does not always force the model to infer new latent causes.

In the standard form Chinese restaurant process (CRP), the prior that *n*^th^ observation is assigned to a new cluster is given by *α* / (*n* - 1 + *α*) (Blei & Gershman, 2012). When *α* = 1, the prior of the new cluster for the 2nd observation will be 0.5; when *α* = 3, this prior increases to 0.75. Thus, when *α* > 1, the prior of the new cluster is above chance early in the sequence, which may relate to the reviewer’s concern. However, this effect diminishes as the number of observations increases. For instance, the prior of the new cluster drops to 0.1 and 0.25 for the 10th observation when *α* = 1 and 3, respectively. Furthermore, the prior in the latent cause model is governed by not only *α* but also *g*, a scaling parameter for the temporal difference between successive observations (see Results in the manuscript) following “distance-dependent” CRP, then normalized over all latent causes including a new latent cause. Thus, it does not necessarily imply that ⍺ greater than 1 forces agents to infer a new latent cause. As shown in Appendix 2 – table 4, the number of latent causes does not inflate in each trial when *α* = 1. On the other hand, the high number of latent causes due to *α* = 2 can be suppressed when *g* = 0.01. More importantly, the driving force is the prediction error generated in each trial (see also comment #31 about the interaction between *α* and *σ*_*x*_^*2*^). Raising the value of *α per se* can be viewed as increasing the probability to infer a new latent cause, not forcing the model to do so by higher *α* alone.

During parameter exploration using the median behavioral data under a wider range of *α* with a lower boundary at 0.1, the estimated value eventually exceeded 1. Therefore, we set the lower bound of *α* to be 1 is to reduce inefficient sampling.

[#30] (2) Number of latent causes: Some mice infer nearly as many latent causes as trials, which seems unrealistic.

We set the upper boundary for the maximum number of latent causes (*K*) to be 36 to align with the infinite features of CRP. This allowed some mice to infer more than 20 latent causes in total. When we checked the learning curves in these mice, we found that they largely fluctuated or did not show clear decreases during the extinction (Author response image 7, colored lines). The simulated learning curves were almost flat in these trials (Author response image 7, gray lines). It might be difficult to estimate the internal states of such atypical mice if the sampling process tried to fit them by increasing the number of latent causes. Nevertheless, most of the samples have a reasonable total number of latent causes: 12-month-old control mice, Mdn = 5, IQR = 4; 12-month-old *App^NL-G-F^* mice, Mdn = 5, IQR = 1.75; 6-month-old control mice, Mdn = 7, IQR = 12.5; 6-month-old *App^NL-G-F^* mice, Mdn = 5, IQR = 5.25. These data were provided in Tables S9 and S12.

**Author response image 9. sa4fig9:** Samples with a high number of latent causes. Observed CR (colored line) and simulated CR (gray line) for individual samples with a total number of inferred latent causes exceeding 20.

[#31] (3) Parameter estimation: With 10 parameters fitting one-dimensional curves, many parameters (e.g., α and σx) are likely highly correlated and poorly identified. Consider presenting scatter plots of the parameters (e.g., α vs σx) in the Supplement.

We have provided the scatter plots with a correlation matrix in Figure 4 – figure supplement 1 for the 12-month-old group and Figure 5 – figure supplement 1 for the 6-month-old group. As pointed out by the reviewer, there are significant rank correlations between parameters including *α* and *σ*_*x*_^*2*^ in both the 6 and 12-month-old groups. However, we also noted that there are no obvious linear relationships between the parameters.

The correlation above raises a potential problem of non-identifiability among parameters. First, we computed the variance inflation index (VIF) for all parameters to examine the risk of multicollinearity, though we did not consider a linear regression between parameters and DI in this study. All VIF values were below the conventional threshold 10 (Appendix 2 – table 5), suggesting that severe multicollinearity is unlikely to bias our conclusions. Second, we have conducted the simulation with different combinations of *α*, *σ*_*x*_^2^, and *K* to clarify their contribution to overgeneralization and overdifferentiation observed in the 12-month-old group.

In Appendix 2 – table 6, the values of *α* and *σ*_*x*_^2^ were either their upper or lower boundary set in parameter estimation, while the value *K* was selected heuristically to demonstrate its effect. Given the observed positive correlation between *α* and *σ_x_*^2^, and their negative correlation with *K* (Figure 4 - figure supplement 1), we consider the product of *K* = {4, 35}, *α* = {1, 3} and *σ*_*x*_^2^ = {0.01, 3}. Among these combinations, the representative condition for the control group is *α* = 3, *σ*_*x*_^2^ = 3, and that for the *App^NL-G-F^* group is *α* = 1, *σ*_*x*_^2^ = 0.01. In the latter condition, overgeneralization and overdifferentiation, which showed higher test 1 CR, lower number of acquisition latent causes (K_acq_), lower test 3 CR, lower DI between test 3 and test 1, and higher number of latent causes after extinction (K_rem_), was extremely induced.

We found conditions that fall outside of empirical correlation, such as *α* = 3, *σ*_*x*_^2^ = 0.01, also reproduced overgeneralization and overdifferentiation. Similarly, the combination, *α* = 1, *σ*_*x*_^2^ = 3, exhibited control-like behavior when *K* = 4 but shifted toward *App^NL-G-F^*-like behavior when *K* = 36. The effect of *K* was also evident when *α* = 3 and *σ*_*x*_^2^ = 3, where *K* = 36 led to over-differentiation. We note that these conditions were artificially set and likely not representative of biologically plausible. These results underscore the non-identifiability concern raised by the reviewer. Therefore, we acknowledge that merely attributing overgeneralization to lower *α* or overdifferentiation to lower *σ*_*x*_^2^ may be overly reductive. Instead, these patterns likely arise from the joint effect of *α, σ_x_*^2^, and *K*. We have revised the manuscript accordingly in Results and Discussion (page 11-13, 18-19).

[#32] (4) Data normalization: Normalizing the data between 0 and 1 removes the interpretability of % freezing, making mice with large changes in freezing indistinguishable seem similar to mice with small changes.

As we describe in our reply to comment #26, the conditioned response in the latent cause model was scaled between 0 and 1, and we assume 0 and 1 mean the minimal and maximal CR within each mouse, respectively. Furthermore, although we initially tried to fit simulated CRs to raw CRs, we found that the fitting level was low due to the individual difference in the degree of behavioral expression: some mice exhibited a larger range of CR, while others showed a narrower one. Thus, we decided to normalize the data. We agree that this processing will make the mice with high changes in freezing% indistinguishable from those with low changes. However, the freezing% changes within the mouse were preserved and did not affect the discrimination index.

[#33] (5) Overlooking parameter differences: Differences in parameters, like w_0_, that didn't fit the hypothesis may have been ignored.

Our initial hypothesis is that internal states were altered in *App^NL-G-F^* mice, and we did not have a specific hypothesis on which parameter would contribute to such a state. We mainly focus on the parameters (1) that are significantly different between control and *App^NL-G^*^-^*^F^* mice and (2) that are significantly correlated to the empirical behavioral data, DI between test 3 and test 1.

In the 12-month-old group, besides *α* and *σ_x_*^2^, *w*_0_ and *K* showed marginal *p*-value in Mann-Whitney *U* test (Table S7) and moderate correlation with the DI (Table S8). While differences in *K* were already discussed in the manuscript, we did miss the point that *w*_0_ could contribute to the differences in w between control and *App^NL-G-F^* (Figure 4G) in the previous manuscript. We explain the contribution of *w*_0_ on the reinstatement results here. When other parameters are fixed, higher *w*_0_ would lead to higher CR in test 3, because higher *w*_0_ would allow increasing *w*_*context*_ by the unsignaled shock, leading to reinstatement (Appendix 2 – table 7). It is likely that higher *w*_0_ would be sampled through the parameter estimation in the 12-month-old control but not *App^NL-G-F^*. On the other hand, the number of latent causes is not sensitive to *w*_0_ when other parameters were fixed at the initial guess value (Appendix 2 – table 1), suggesting *w*_*0*_ has a small contribution to memory modification process.

Thus, we speculate that although the difference in *w*_*0*_ between control and *App^NL-G-F^* mice may arise from the sampling process, resulting in a positive correlation with DI between test 3 and test 1, its contribution to diverged internal states would be smaller relative to *α* or *σ_x_*^2^ as a wide range of *w*_0_ has no effect on the number of latent causes (Appendix 2 – table 7). We have added the discussion of differences in *w*_*0*_ in the 12-month-old group in manuscript Line 357-359.

In the 6-month-old group, besides *α* and *σ_x_*^2^, θ is significantly higher in the AD mice group (Table S10) but not correlated with the DI (Table S11). We have already discussed this point in the manuscript.

[#34] (6) Initial response: Higher initial responses in the model at the start of the experiment may reflect poor model fit.

Please refer to our reply to comment #26 for our explanation of what contributes to high initial responses in the latent cause model.

In addition, achieving a good fit for the acquisition CRs was not our primary purpose, as the response measured in the acquisition phase includes not only a conditioned response to the CS and context but also an unconditioned response to the novel stimuli (CS and US). This mixed response presumably increased the variance of the measured freezing rate over individuals, therefore we did not cover the results in the discussion.

Rather, we favor models at least replicating the establishment of conditioning, extinction and reinstatement of fear memory in order to explain the memory modification process. As we mentioned in the reply for comment #4, alternative models, the latent state model and the Rescorla-Wagner model, failed to replicate the observation (cf. Figure 3 – figure supplement 1A-1C). Thus, we chose to stand on the latent cause model as it aligns better with the purpose of this study.

[#35] In addition, please be transparent if data is excluded, either during the fitting procedure or when performing one-way ANCOVA. Avoid discarding data when possible, but if necessary, provide clarity on the nature of excluded data (e.g., how many, why were they excluded, which group, etc?).

We clarify the information of excluded data as follows. We had 25 mice for the 6-month-old control group, 26 mice for the 6-month-old *App^NL-G-F^* group, 29 mice for the 12-month-old control group, and 26 mice for the 12-month-old *App^NL-G-F^* group (Table S1).

Our first exclusion procedure was applied to the freezing rate data in the test phase. If the mouse had a freezing rate outside of the 1.5 IQR in any of the test phases, it is regarded as an outlier and removed from the analysis (see Statistical analysis in Materials and Methods). One mouse in the 6-month-old control group, one mouse in the 6-month-old *App^NL-G-F^* group, five mice in the 12-month-old control group, and two mice in the 12-month-old *App^NL-G-F^* group were excluded.

Our second exclusion procedure was applied during the fitting and parameter estimation (see parameter estimation in Materials and Methods). We have provided the number of anomaly samples during parameter estimation in Appendix 1 – figure 2.

Lastly, we would like to state that all the sample sizes written in the figure legends do not include outliers detected through the exclusion procedure mentioned above.

[#36] Finally, since several statistical tests were used and the differences are small, I suggest noting that multiple comparisons were not controlled for, so p-values should be interpreted cautiously.

We have provided power analyses in Tables S21 and S22 with methods described in the manuscript (Line 897-898) and added a note that not all of the multiple comparisons were corrected for in the manuscript (Line 898-899).

References cited in the response letter only

Bellio, T. A., Laguna-Torres, J. Y., Campion, M. S., Chou, J., Yee, S., Blusztajn, J. K., & Mellott, T. J. (2024). Perinatal choline supplementation prevents learning and memory deficits and reduces brain amyloid Aβ42 deposition in *AppNL-G-F* Alzheimer’s disease model mice. PLOS ONE, 19(2), e0297289. https://doi.org/10.1371/journal.pone.0297289

Blei, D. M., & Frazier, P. I. (2011). Distance Dependent Chinese Restaurant Processes. Journal of Machine Learning Research, 12(74), 2461–2488.

Cochran, A. L., & Cisler, J. M. (2019). A flexible and generalizable model of online latent-state learning. PLOS Computational Biology, 15(9), e1007331. https://doi.org/10.1371/journal.pcbi.1007331

Curiel Cid, R. E., Crocco, E. A., Duara, R., Vaillancourt, D., Asken, B., Armstrong, M. J., Adjouadi, M., Georgiou, M., Marsiske, M., Wang, W., Rosselli, M., Barker, W. W., Ortega, A., Hincapie, D., Gallardo, L., Alkharboush, F., DeKosky, S., Smith, G., & Loewenstein, D. A. (2024). Different aspects of failing to recover from proactive semantic interference predicts rate of progression from amnestic mild cognitive impairment to dementia. Frontiers in Aging Neuroscience, 16. https://doi.org/10.3389/fnagi.2024.1336008

Giustino, T. F., Fitzgerald, P. J., Ressler, R. L., & Maren, S. (2019). Locus coeruleus toggles reciprocal prefrontal firing to reinstate fear. Proceedings of the National Academy of Sciences, 116(17), 8570–8575. https://doi.org/10.1073/pnas.1814278116

Gu, X., Wu, Y.-J., Zhang, Z., Zhu, J.-J., Wu, X.-R., Wang, Q., Yi, X., Lin, Z.-J., Jiao, Z.-H., Xu, M., Jiang, Q., Li, Y., Xu, N.-J., Zhu, M. X., Wang, L.-Y., Jiang, F., Xu, T.-L., & Li, W.-G. (2022). Dynamic tripartite construct of interregional engram circuits underlies forgetting of extinction memory. Molecular Psychiatry, 27(10), 4077–4091. https://doi.org/10.1038/s41380-022-01684-7

Lacagnina, A. F., Brockway, E. T., Crovetti, C. R., Shue, F., McCarty, M. J., Sattler, K. P., Lim, S. C., Santos, S. L., Denny, C. A., & Drew, M. R. (2019). Distinct hippocampal engrams control extinction and relapse of fear memory. Nature Neuroscience, 22(5), 753–761. https://doi.org/10.1038/s41593-019-0361-z

Loewenstein, D. A., Curiel, R. E., Greig, M. T., Bauer, R. M., Rosado, M., Bowers, D., Wicklund, M., Crocco, E., Pontecorvo, M., Joshi, A. D., Rodriguez, R., Barker, W. W., Hidalgo, J., & Duara, R. (2016). A Novel Cognitive Stress Test for the Detection of Preclinical Alzheimer’s Disease: Discriminative Properties and Relation to Amyloid Load. The American Journal of Geriatric Psychiatry : Official Journal of the American Association for Geriatric Psychiatry, 24(10), 804–813. https://doi.org/10.1016/j.jagp.2016.02.056

Loewenstein, D. A., Greig, M. T., Curiel, R., Rodriguez, R., Wicklund, M., Barker, W. W., Hidalgo, J., Rosado, M., & Duara, R. (2015). Proactive Semantic Interference Is Associated With Total and Regional Abnormal Amyloid Load in Non-Demented Community-Dwelling Elders: A Preliminary Study. The American Journal of Geriatric Psychiatry : Official Journal of the American Association for Geriatric Psychiatry, 23(12), 1276–1279. https://doi.org/10.1016/j.jagp.2015.07.009

Valles-Salgado, M., Gil-Moreno, M. J., Curiel Cid, R. E., Delgado-Á lvarez, A., Ortega-Madueño, I., Delgado-Alonso, C., Palacios-Sarmiento, M., López-Carbonero, J. I., Cárdenas, M. C., MatíasGuiu, J., Díez-Cirarda, M., Loewenstein, D. A., & Matias-Guiu, J. A. (2024). Detection of cerebrospinal fluid biomarkers changes of Alzheimer’s disease using a cognitive stress test in persons with subjective cognitive decline and mild cognitive impairment. Frontiers in Psychology, 15. https://doi.org/10.3389/fpsyg.2024.1373541

Zaki, Y., Mau, W., Cincotta, C., Monasterio, A., Odom, E., Doucette, E., Grella, S. L., Merfeld, E., Shpokayte, M., & Ramirez, S. (2022). Hippocampus and amygdala fear memory engrams reemerge after contextual fear relapse. Neuropsychopharmacology, 47(11), 1992–2001. https://doi.org/10.1038/s41386-022-01407-0